# LGR signaling mediates muscle-adipose tissue crosstalk and protects against diet-induced insulin resistance

Olga Kubrak[1,4], Anne F. Jørgensen [1,2,4], Takashi Koyama [1,4], Mette Lassen [1,4], Stanislav Nagy[1], Jacob Hald[2], Gianluca Mazzoni[2], Dennis Madsen [2], Jacob B. Hansen [1], Martin Røssel Larsen [3], Michael J. Texada [1], Jakob L. Hansen [2], Kenneth V. Halberg [1] & Kim Rewitz [1] ✉

Obesity impairs tissue insulin sensitivity and signaling, promoting type-2 diabetes. Although improving insulin signaling is key to reversing diabetes, the multi-organ mechanisms regulating this process are poorly defined. Here, we screen the secretome and receptome in *Drosophila* to identify the hormonal crosstalk affecting diet-induced insulin resistance and obesity. We discover a complex interplay between muscle, neuronal, and adipose tissues, mediated by Bone Morphogenetic Protein (BMP) signaling and the hormone Bursicon, that enhances insulin signaling and sugar tolerance. Muscle-derived BMP signaling, induced by sugar, governs neuronal Bursicon signaling. Bursicon, through its receptor Rickets, a Leucine-rich-repeat-containing G-protein coupled receptor (LGR), improves insulin secretion and insulin sensitivity in adipose tissue, mitigating hyperglycemia. In mouse adipocytes, loss of the Rickets ortholog LGR4 blunts insulin responses, showing an essential role of LGR4 in adipocyte insulin sensitivity. Our findings reveal a muscle-neuronal-fat-tissue axis driving metabolic adaptation to high-sugar conditions, identifying LGR4 as a critical mediator in this regulatory network.

Insulin signaling is essential for glucose homeostasis. In peripheral tissues, insulin promotes the uptake of glucose from the blood and stimulates its conversion into glycogen and fat for storage. Insulin-stimulated glucose uptake and metabolism are reduced in insulin-resistant subjects, leading to elevated blood glucose levels[1]. This elevation intensifies the secretory demand on pancreatic β-cells, which is further amplified by the increase in body mass due to obesity. This can result in β-cell loss or damage, ultimately leading to inadequate levels of circulating insulin and the development of diabetes[2]. The adaptive mechanisms of hormonal and cellular physiology effectively manage glucose homeostasis under normal conditions. However, obesity and high-calorie diets can trigger insulin resistance in adipose tissue and other organs[3]. Despite the great medical importance of this condition,

the underlying hormonal pathways and mechanisms that become disrupted due to nutrient excess and subsequently lead to impaired insulin signaling remain poorly defined.

Many in-vivo studies and models of diet-induced insulin resistance have linked decreased tissue insulin sensitivity with factors such as lipid accumulation, endoplasmic-reticulum stress, and inflammatory stress responses[4,5]. While improving β-cell function and ameliorating insulin resistance in individuals with obesity are key steps towards restoring normal glucose absorption to prevent and treat diabetes, the underlying mechanisms involve multi-organ crosstalk within complex hormonal systems that are still inadequately defined. Animal models of diet-induced obesity display insulin resistance and thus enable a range of experimental

[1]Department of Biology, University of Copenhagen, 2100 Copenhagen O, Denmark. [2]Novo Nordisk, Novo Nordisk Park, 2760 Maaløv, Denmark. [3]Department of Biochemistry and Molecular Biology, University of Southern Denmark, 5230 Odense, Denmark. [4]These authors contributed equally: Olga Kubrak, Anne F. Jørgensen, Takashi Koyama, Mette Lassen. ✉e-mail: Kim.Rewitz@bio.ku.dk

approaches to elucidate these mechanisms and identify targets for diabetes treatment[6,7]. Because of the conservation of most metabolic pathways and many hormonal systems, the fruit fly *Drosophila* has become a valuable model for understanding the contribution of nutrition to metabolic disorders[6,8]. In flies, diet-induced obesity produces many of the same pathophysiological effects observed in humans with obesity, indeed including hyperglycemia, altered insulin secretion, and insulin resistance[9,10]. Given the fly's genetic adaptability, this model of diet-induced insulin resistance can be exploited to provide insight into the complex tissue crosstalk that drives insulin resistance and diabetes progression.

Insulin sensitivity is essential in both muscle and adipose tissue for glucose uptake and storage, and these tissues are critical regulators of systemic glucose homeostasis[11,12]. Signals from organs such as the gut and adipose tissue also relay nutritional information to the brain, which produces factors that regulate systemic energy homeostasis[13]. The crosstalk between these and other organs is mediated by circulating hormones and cytokines. Gut hormones such as glucagon-like peptide 1 (GLP-1) are responsible for incretin effects, which potentiate glucose-induced insulin release[14]. While gut-derived signals such as GLP-1 have become a primary therapeutic target for obesity and diabetes treatment, diverse myokines and adipokines also mediate crosstalk between the muscles, adipose tissue, and other organs, including the brain, to modulate systemic energy metabolism and insulin sensitivity[15,16]. The adipokine leptin can improve hyperglycemia and diabetes in animal models[17], and myokines can enhance insulin sensitivity[18]. Like those of mammals, the tissues of the fly produce a range of adipokines, myokines, and gut-derived hormones that govern development, metabolism, and food choice[19–25]. Mammalian leptin is secreted from adipose tissue and stimulates insulin secretion and sensitivity in other organs via the nervous system. Similarly, the *Drosophila* cytokine Unpaired 2, related to mammalian leptin, is secreted from the adipose tissue and promotes insulin secretion[24]. While leptin enables fat tissue to modulate insulin signaling and sensitivity in the liver and muscles, reciprocal communication from muscles to adipose tissue via the nervous system has not been described.

Here, we present the results of an extensive in-vivo knockdown screen in *Drosophila* that examines the effects of the secretome and receptome on sucrose toxicity. Our findings show that, on a high-sugar diet that induces obesity-like phenotypes, including insulin resistance, interorgan communication between muscle, neuronal, and fat tissue maintains insulin production and adipose insulin sensitivity, which mitigates sugar-induced hyperglycemia. In response to elevated sugar levels, muscle-derived Bone Morphogenetic Protein (BMP) signals to neurons expressing the hormone Bursicon. This factor acts through a Leucine-rich-repeat-containing GPCR (LGR) type-B receptor and directly stimulates insulin production and indirectly increases insulin signaling by enhancing the sensitivity of adipose tissue to insulin. Moreover, in mouse adipocytes, loss of *Lgr4*, orthologous with the *Drosophila* Bursicon receptor, leads to a significantly blunted insulin response, demonstrating that LGR4 is essential for maintaining normal adipocyte insulin sensitivity and signaling.

## Results

### In-vivo RNA-interference screen of the secretome and receptome for hormonal signals affecting sugar tolerance

The maintenance of sugar homeostasis in response to nutritional challenges requires the coordinated adaptive functions of multiple organs. Secreted molecules such as hormones and cytokines mediate signaling between organs to orchestrate these coordinated responses. To identify potential interorgan signaling factors that mediate the effects of nutrient excess on insulin signaling and sugar tolerance, we performed an in-vivo RNAi-mediated screen of the secretome and receptome in a fly model of diet-induced obesity and

diabetes. In this system, a high-sugar diet (HSD) induces phenotypes that mirror some effects of human obesity, including hyperglycemia and insulin resistance[9,10]. The fly genome encodes only a single insulin receptor, and thus this pathway combines the effects of mammalian insulin itself (sugar balance) and IGFs (tissue growth). Because of this common signaling channel, insulin-perturbation effects in the larva manifest as a strong slowing of growth that leads to a sugar-dose-dependent developmental delay (Fig. 1a, b). We rationalized that knockdown of genes with critical roles in sugar tolerance would impair survival or delay pupariation on HSD (5x sugar in the diet) while having no effect on development on a normal diet (ND, 1x sugar). We confirmed that loss of Hexokinase-A (Hex-A), the enzyme mediating the first step in the glycolytic breakdown of glucose, produces a sugar-dose-dependent developmental-delay phenotype, validating our screening approach for identifying genes important for sugar metabolism (Fig. 1c). To ensure that our approach could detect differences in timing despite variation in the number of offspring among the different fly crosses, we assessed whether crowding would affect HSD-induced phenotypes. Examining mean pupariation timing in 25-mm diameter vials of 1x- or 5x-sugar medium containing between 5 and 150 larvae, we observed no difference in pupariation time between populations (Supplementary Fig. 1), indicating that larval crowding does not affect timing within this range, which covers the distribution of populations produced within our screen.

Next, we screened for genetically induced alterations in sugar-induced developmental delay – indicating a role in sugar tolerance or insulin signaling – by targeting 2,256 genes encoding potential secreted factors and receptors. Hits were identified by comparing the HSD-induced delay of knockdown animals to effects on development of controls. To identify sugar-dependent phenotypes, we compared 5x-sugar (HSD) phenotypes to 1x-sugar (ND) phenotypes. Global knockdown driven by *daughterless-GAL4* (*da* >) identified 119 genes (a hit rate of 5.3%) whose reduction exacerbated the sugar-induced delay phenotypes or impaired survival, or both, on HSD compared to controls (*da* > +), while giving no or comparatively minor effects on ND (Fig. 1d, e and Supplementary Data 1). We note that we did not identify any hits whose knockdown led to a significant reversal of the sugar-induced delay phenotypes. This outcome might be attributed to limitations in the screening approach, such as sensitivity or the severity of the HSD challenge, which could make it difficult to mitigate the induced delay. It might also require simultaneously targeting multiple partially redundant genes or pathways, where a single-gene approach is inefficient. The significant delay phenotypes ranged from full developmental arrest on HSD to partial arrest or prolongation of the HSD-induced developmental delay. Among the strongest 79 hits that produced only sugar-dependent phenotypes (*i.e.*, developmental arrest or lengthened delay on HSD paired with no delay on ND; Fig. 1d), 81% have at least one human ortholog (predicted by DIOPT[26]) (Fig. 1d). While many of these sugar-dependent gene hits (39%) have been associated with diabetes, obesity, or lipid and glucose metabolism (DIOPT-DIST based on OMIM and GWAS disease terms[26]), only 5% (2 genes) of the genes showing partially sugar-dependent phenotypes were associated with these terms (Fig. 1e). Among the sugar-dependent screen hits were several genes known to be important for sugar tolerance and metabolic homeostasis, including those encoding the insulin receptor (*InR*) itself; the transcription factor Seven-up (*svp*), a positive regulator of insulin signaling in adipose tissue required for maintaining sugar homeostasis under high-dietary-sugar conditions[27]; Smad on X (*Smox*), a TGF-β-pathway transcription factor required for glycemic control on a high-sugar diet via regulation of glucagon-like signaling in the fat tissue[28]; and FGF signaling components, which regulate insulin in response to fat-tissue oxygen levels[29]. These observations validate our approach and suggest that it is a powerful strategy to identify the mechanisms and hormonal pathways that

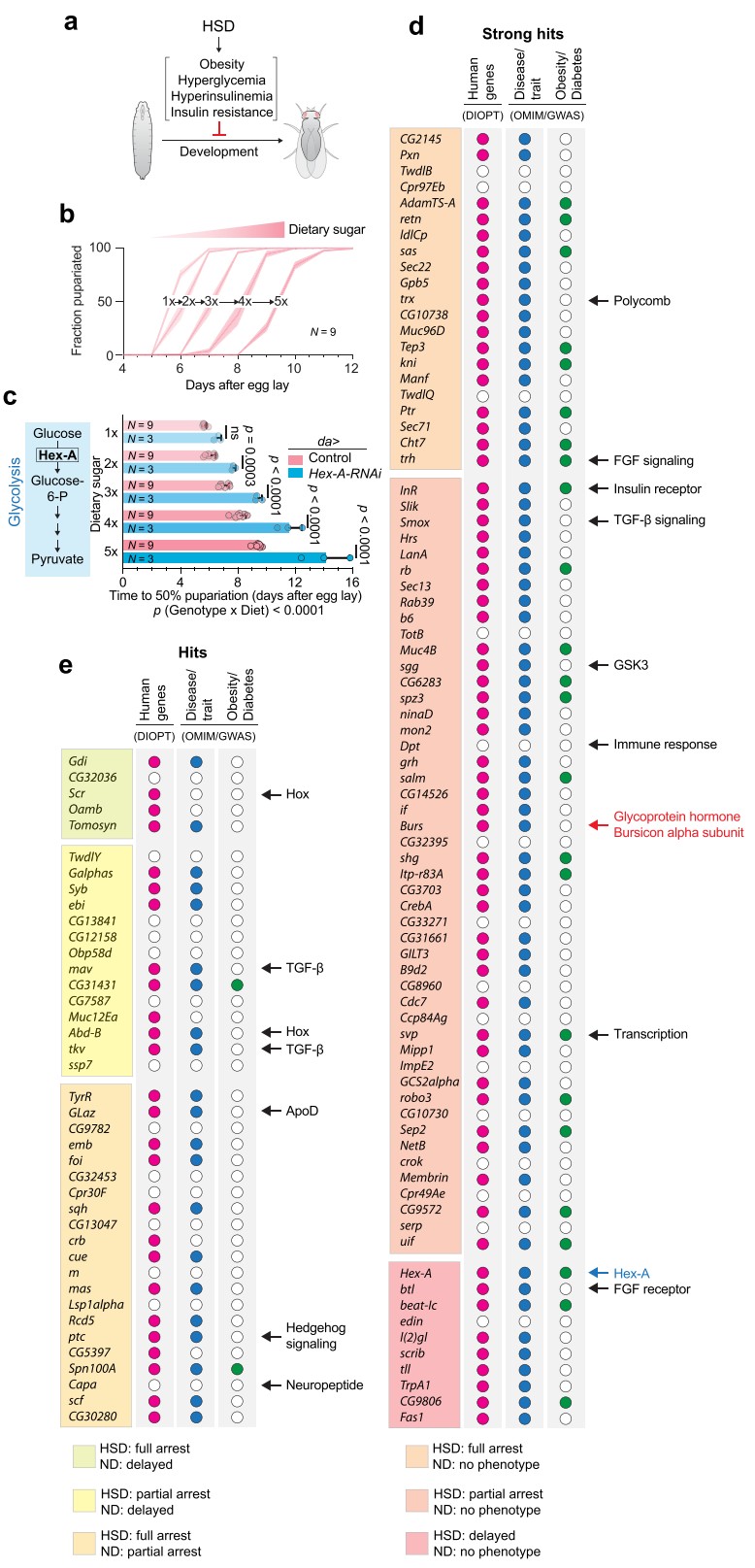

control sugar homeostasis. The Hedgehog and Capa-peptide signaling pathways that have been linked with metabolism were also identified[30,31], along with other interesting pathways that have not previously been linked with sugar homeostasis, such as Hox genes (*Scr* and *Abd-B*), Polycomb chromatin remodeling, and immune responses. Among our strongest hits, we identified α-Bursicon, which with its

paralog β-Bursicon [encoded by *Burs* and *Partner of bursicon* (*Pburs*), respectively] makes up the heterodimeric cystine-knot glycoprotein Bursicon. This hormone acts through the conserved type-B leucine-rich repeat-containing G-protein coupled receptor (LGR) dLgr2/Rick-ets (orthologous with mammalian type-B receptors Lgr4/−5/−6)[32], as an important factor for sugar tolerance.

**Fig. 1 | A screen for genes affecting adaptation to high-sugar diet in *Drosophila*.** **a** The basis of the screen. **b** Pupariation timing on media containing a range of sugar concentrations. **c** The time until 50% of animals had pupariated, in controls and *Hex-A* knockdown animals, on a range of dietary sugar levels, with the difference in 50%-pupariation times between controls and knockdowns on 1x (normal diet, ND) and 5x sugar (high sugar diet, HSD). **d** Screen hits that displayed no phenotype on 1x sugar while exhibiting full (top) or partial (middle) failure to pupate or a delay in pupariation (bottom) on 5x-sugar medium. Among these hits was the Burs (Bursalpha) subunit of the Bursicon glycoprotein heterodimer. Other notable hits are indicated as well. Genes marked with a red dot have at least one ortholog in humans; if the human gene(s) have been associated with a disease or metabolic trait, the *Drosophila* gene is marked with a blue dot; and if that human disease or trait is related to obesity or diabetes, the *Drosophila* gene is marked with a green dot. **e** Screen hits that displayed a stronger phenotype on 5x sugar than on 1x sugar – those that permit development, albeit with delay, on 1x sugar while fully (top) or partially (middle) blocking development on 5x, and those that exhibit a partial arrest on normal diet and a full arrest on 5x sugar (bottom). Genes are flagged as in (**d**), and notable hits are indicated. All animals were raised at 25 °C. Statistics: b, Data are presented as means, with shaded areas representing the SEM from biologically independent replicates. **c** Data are presented as means, with error bars representing SEM from biologically independent replicates. Sample sizes (*N*) are indicated on graphs. ns, not significant ($p > 0.05$); ND normal diet; HSD high-sugar diet. C twoway ANOVA for interaction and one-way Kruskal-Wallis nonparametric ANOVA with Dunn's correction for multiple comparisons. Source data are provided as a Source Data file.

## Neuronal Bursicon signaling is required for metabolic adaptations that promote fat storage and reduce hyperglycemia on diets rich in sugar

To investigate the role of Bursicon signaling in sugar tolerance, we first assessed the effects of a variety of *Burs* loss-of-function genotypes on the HSD phenotype. We found that animals homozygous for a *Burs* null mutation (*Burs^Z4410*) affecting the entire animal exhibited a strong prolongation of the HSD-dependent delay, as did tissue-specific RNAi-mediated knockdown or CRISPR-mediated somatic deletion of *Burs* in Burs⁺ cells (driven by *Burs-GAL4*, *Burs >* ) (Fig. 2a,b). The phenotypes were not attributable to the genetic background of the UAS-RNAi and UAS-CRISPR lines, as animals carrying these constructs without the GAL4 driver did not exhibit delay compared to the control (Supplementary Fig. 2a). Burs, the α-subunit of the heterodimeric Bursicon hormone, was previously shown to be expressed without the β-subunit in the adult midgut and to control systemic metabolism in response to sugar feeding[33]. However, during development, the active Bursicon hormone is believed to be a heterodimer produced by neurons[32]. Consistent with this, Burs and Pburs are predominantly expressed in the nervous system during development[34]. The α-subunit encoded by *Burs* is expressed at very low levels in the larval midgut, and the β-subunit encoded by *Pburs* is not detectably expressed. To address whether neuronal Burs:Pburs signaling is required for sugar tolerance, we disrupted *Burs* and *Pburs* specifically in the nervous system and assessed sugar-dependent effects. Animals with pan-neuronal knockdown or knockout of either *Burs* or *Pburs* showed a sugar-dependent developmental delay (Supplementary Fig. 2b–e) without contributing phenotypic effects of the RNAi or CRISPR constructs alone (Supplementary Fig. 2f). We also ruled out the contribution of gut Bursicon to the sugar-intolerance phenotype (Supplementary Fig. 2g). These data suggest that it is the heterodimeric Burs:Pburs hormone, produced by a subset of Burs⁺ neurons that express both subunits, that is important for metabolic adaptation to high-sugar conditions. We further corroborated our findings by feeding animals a diet with a high sugar-to-yeast ratio but with normal sugar concentration and low yeast concentration (6% sugar and 0.34% yeast, *i.e.*, 0.1x yeast, the latter being the primary source of protein and sterols). In comparison to the controls, animals with neuronal knockdown or knockout of *Burs* demonstrated a substantial delay on the low-yeast diet (Supplementary Fig. 2h). This indicates importance of neuronal Bursicon signaling for adaptation to diets with a high ratio of sugar to yeast.

The capacity to adapt to conditions of high sugar intake directly depends on the ability to store excess energy by converting it into fat deposits in adipose tissues. Animals raised on HSD, therefore, exhibit increased body fat stored as triacylglycerides (TAGs)[9,10]. We, therefore, investigated whether Bursicon signaling mediates metabolic adaptation to nutrient excess by assessing animals' fat storage. We observed a significant increase in whole-body TAG levels in animals fed a high-sugar diet for 10 h, an effect that was abolished by RNAi targeting of *Burs* in Burs⁺ cells (Fig. 2c). This increase was also attenuated in animals expressing pan-neuronal RNAi or CRISPR targeting *Burs* or *Pburs*,

whereas on ND these animals' TAG levels were unchanged from those of controls (Fig. 2d, e). This indicates that neuronal Bursicon signaling is required for excess glucose to be converted to fat for storage on high-sugar diets.

Insulin resistance impairs the cellular uptake of glucose and its conversion to glycogen and fat for storage. We therefore assessed the possibility that the lipid-storage defect observed in animals with reduced neuronal Burs expression might arise from impaired insulin signaling. Inhibition of Bursicon signaling through neuronal *Burs* knockdown or knockout led to increased whole-body expression of *4EBP*, a target gene of FOXO whose expression is repressed by insulin signaling[35], on HSD but not on ND (Fig. 2f), suggesting decreased peripheral insulinsignaling. Consistent with this observation, loss of *Burs* was also associated with decreased phosphorylation of the insulin-signaling component AKT in whole-animal samples only on high-sugar diet (Fig. 2g). Collectively, these findings suggest that loss of *Burs* expression impairs sugar tolerance, metabolic adaptation, and insulin signaling in animals consuming a high-sugar diet. Therefore, we focused our subsequent investigations on the effects of Bursicon signaling under high-sugar diet conditions. Impairments in insulin signaling reduce tissue glucose uptake from circulation, leading to hyperglycemia[9]. Consistent with reduced insulin signaling, loss of *Burs* (through either RNAi or CRISPR) led to elevated glycemic levels after chronic exposure to a high-sugar diet (Fig. 2h). While the same trend was observed following short-term (10-hour) exposure (Fig. 2i), the effect of *Burs* CRISPR-targeting under this condition did not reach statistical significance. Together, our results indicate that neuronal Bursicon signaling is required to modulate insulin signaling or sensitivity to adapt systemic metabolism and maintain glycemic homeostasis under high-sugar-diet conditions.

## Burs⁺ neurons respond to dietary sugar and control insulin production

Insulin is the key regulatory hormone that drives the reduction of blood glucose levels after sugar intake. Secretion of insulin from the insulin-producing cells (IPCs) – the main source of circulating insulin in *Drosophila* and analogous with the pancreatic β-cells – is governed by many nutritional cues from peripheral organs[35,36]. To test whether Bursicon signaling regulates insulin production and release on HSD, we assessed the effect of *Burs* loss on the levels of insulin-like peptides 2, 3, and 5 (Ilp2, −3, and −5), the three main insulins produced by the IPCs. Consistent with the increased *4EBP* expression and lower levels of phosphorylated-AKT observed in animals with reduced Burs signaling on HSD, which suggest reduced systemic insulin signaling (Fig. 2f, g), we observed a strong reduction in *Ilp2* and *Ilp3* transcript levels, albeit without any change in *Ilp5* expression, in the central nervous system (CNS) of animals with loss of *Burs* function, indicating reduced insulin production (Fig. 3a). Consistent with this lower expression, the levels of Ilp2 and Ilp3 peptides, but not of Ilp5, were reduced in the IPCs (Fig. 3b and Supplementary Fig. 3a) in animals with loss of *Burs*. Taken together, these findings suggest that, in high-dietary-sugar conditions,

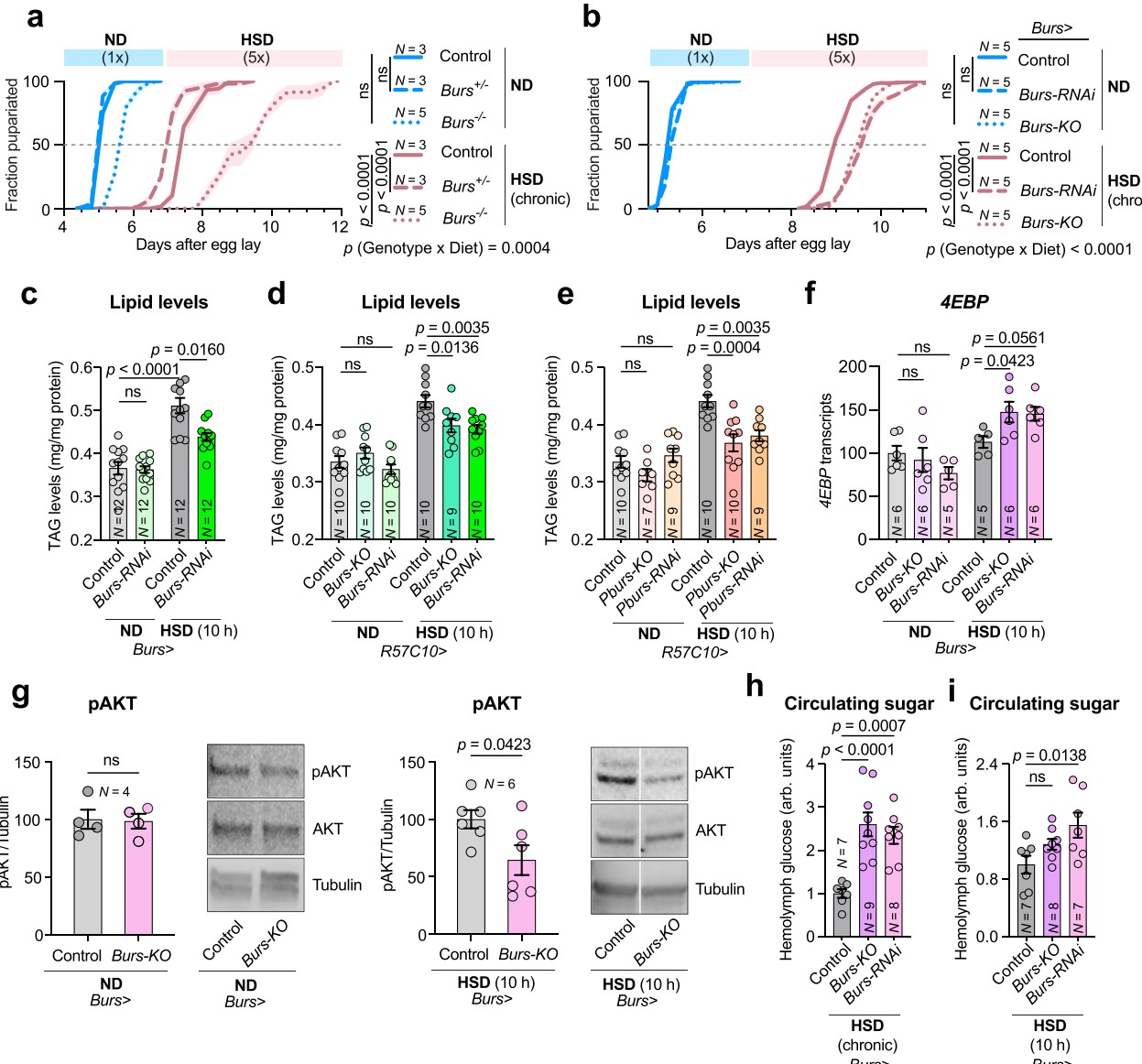

**Fig. 2 | Burs and PBurs are required for metabolic and developmental adaptation to a high-sugar diet in *Drosophila*. a, b** Pupariation timing on normal diet (ND) and high-sugar diet (HSD) for controls and *Burs^Z4410* heterozygotes and homozygous mutants (**a**), and for animals with loss of Burs in *Burs*-expressing cells (**b**). **c–e** Stored lipid levels in animals raised for 90 h on ND and then fed a 10-hour pulse of ND or HSD, comparing controls and animals with *Burs* knockdown in *Burs*-expressing cells (**c**) or pan-neuronal *Burs* (**d**) or *Pburs* (**e**) loss. Experiments measuring TAG levels in animals lacking *Burs* and *Pburs* were performed concurrently and share the control but are presented in separate panels (Fig. 2d, e) for clarity. **f** Whole-body *4EBP* transcript levels measured by qPCR in controls and animals with *Burs* loss in the *Burs*-expressing cells, raised on ND for 90 h and then fed with a 10-hour pulse of ND or HSD. **g** Whole-body pAkt levels, normalized to Tubulin, measured by Western blot, in controls and *Burs*-cell *Burs*-deletion animals fed ND or HSD for ten h after 90 h on ND. **h, i** Hemolymph glucose levels of control and *Burs*-cell *Burs*-loss animals raised on HSD (**h**), and raised on ND then exposed to a 10-hour pulse of HSD (**i**). Animals in (**a**) were raised at 25 °C, while all others were raised at 29 °C to enhance RNAi and CRISPR efficiencies. Data are presented as means; SEM is shown with shaded areas in (**a, b**) and error bars in other figures. All data are from biologically independent replicates, with sample sizes indicated on graphs. ns, not significant ($p > 0.05$). **a** one-way Kruskal-Wallis nonparametric ANOVA with Dunn's multiple-comparison test between 50%-pupariation times and two-way ANOVA for interaction. 50% pupariation time was determined via linear interpolation between adjacent observations below/before and above/after 50%. **b**, one-way ANOVA with Tukey multiple comparisons between 50%-pupariation times and two-way ANOVA for interaction. **c**, two-sided Mann-Whitney nonparametric U test. **d, e, f, h, i**: one-way ANOVAs with Dunnett's multiple-comparison test. **g**, two-sided unpaired parametric t-test. Source data are provided as a Source Data file.

Bursicon signaling acts on the IPCs to promote the production of Ilp2 and Ilp3 to increase insulin signaling and enhance tissue uptake of circulating glucose.

We noticed that Burs⁺ neurons send projections to the subesophageal zone (SEZ), a region in which the IPCs also arborize. To examine the proximity of arbors of Burs⁺ neurons and the IPCs, we co-stained brains for both neuronal populations and found that neurites of the Burs⁺ neurons project in close proximity to the arbors of the IPCs

(Fig. 3c), suggesting that the IPCs might receive input from Burs⁺ neurons. We, therefore, sought to test for synaptic connectivity between the two neuronal populations using the anterograde transsynaptic tracing technique trans-Tango, in which a tethered ligand expressed in pre-synaptic cells activates its cognate receptor in postsynaptic target neurons, leading to *tdTomato* expression in these cells[37]. We observed a trans-Tango-dependent tdTomato signal in the IPCs when the presynaptic ligand was expressed in Burs⁺ neurons

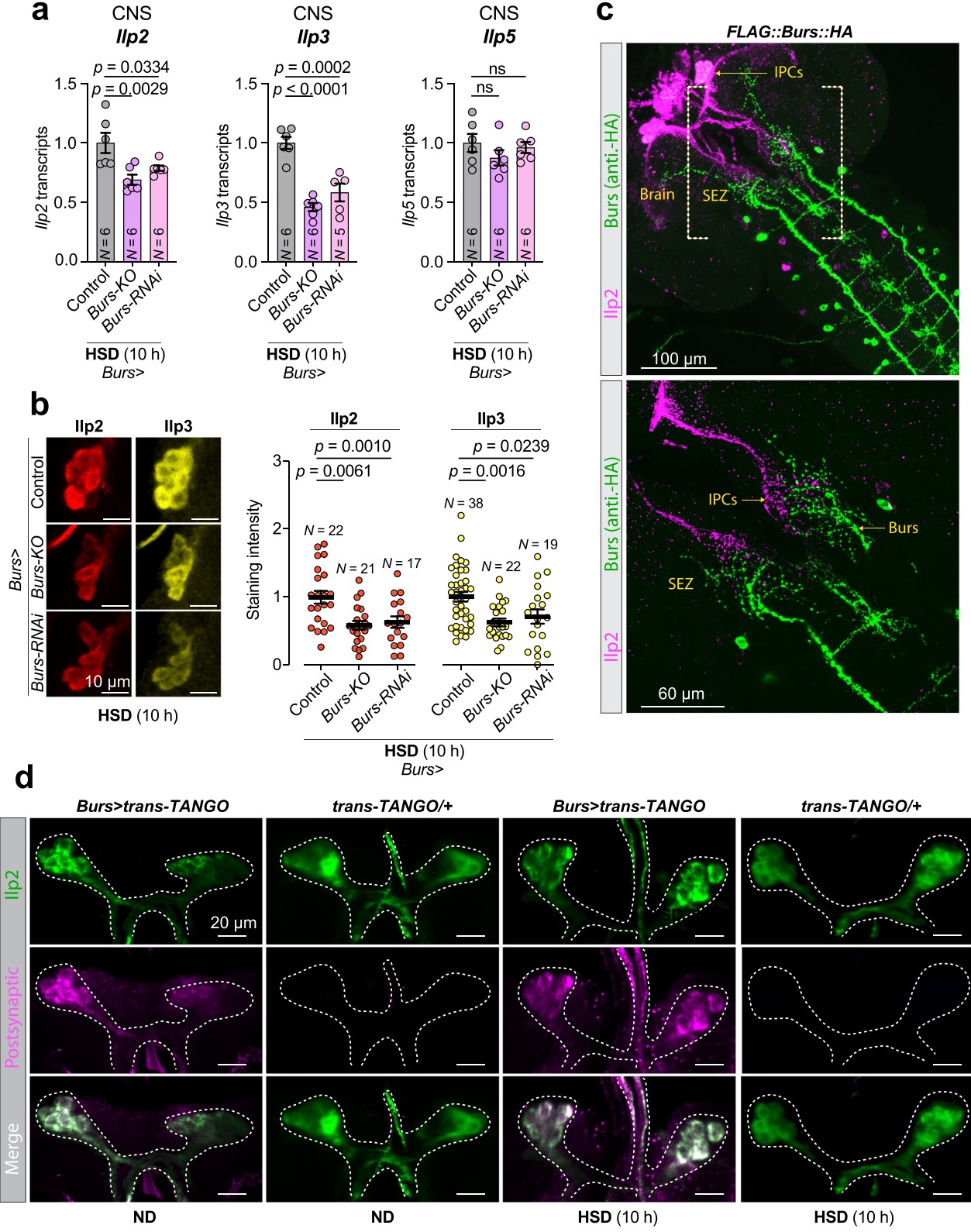

(Fig. 3d), indicating that the IPCs are directly postsynaptic to one or more Burs⁺ neurons. To assess whether Bursicon itself acts directly on the IPCs, we examined whether the larval IPCs express the Burs:Pburs receptor Rickets (Rk) using a T2A::GAL4 knock-in into the endogenous *rk* locus to drive expression of *UAS-mCD8::GFP*. We observed GFP reporter expression in the IPCs (Fig. 4a), indicating that *rk* is expressed in these cells. This is further supported by single-nucleus

transcriptomics data (Supplementary Fig. 3b)[38], together suggesting that the IPCs are receptive for Bursicon signaling. Next, we investigated whether loss of Bursicon signaling in the IPCs might be responsible for the HSD-induced developmental delay observed in Burs-/Pburs-loss animals. We found that knockdown or knockout of *rk* in the IPCs resulted in a prolonged HSD-induced developmental delay (Fig. 4b) without contributing effects from the RNAi or CRISPR

**Fig. 3 | Burs regulates the IPCs, likely through direct neuronal communication in *Drosophila*. a** Transcript levels of *Ilp2, -3*, and *−5* in dissected CNS samples of controls and animals with *Burs⁺*-cell *Burs* loss, exposed to high-sugar diet (HSD) for 10 h after 90 h' feeding on normal diet. **b** Left: Representative images showing levels of Ilp2 and Ilp3 retained within the IPCs in animals with Burs⁺-cell *Burs* loss. Right: Quantified levels of these peptides in IPCs from multiple animals. **c** Larval CNS stained for Ilp2 (purple) and FLAG::Burs::HA (anti-HA, green). Scale bars: top, 100 microns; bottom, 60 microns. Images are representative of three independent experiments with similar results. **d** *Burs*-cell-originating trans-Tango signal (purple) in the IPCs, marked with anti-Ilp2 (green) in animals exposed to normal diet (ND) or high-sugar diet (HSD) for 10 h after 90 h' feeding on normal diet. Scale bars, 20 microns. *Trans-TANGO* without the *Burs-GAL4* (*Burs >* ) serves as a negative control for leaky expression (*trans-TANGO/+*). Anti-Ilp2, green, and anti-tdTomato, magenta. Images are representative of five independent experiments with similar results. All animals were raised at 29 °C. Statistics: Data are presented as means, with error bars representing SEM from biologically independent replicates, and sample sizes (*N*) are indicated on graphs. ns, not significant (*p* > 0.05); HSD, high-sugar diet. a,b one-way ANOVAs with Dunnett's multiple-comparisons test. Source data are provided as a Source Data file.

constructs alone to thís phenotype (Supplementary Fig. 3c), along with reduced *Ilp2* and *Ilp3* transcript levels, with unchanged expression of *Ilp5* (Fig. 4c), phenocopying *Burs* loss. The reduced *Ilp* transcript levels were also reflected by reduced levels of Ilp3 peptides in the IPCs (Fig. 4d). Consistent with this, loss of Rk activity in the IPCs led to reduced levels of circulating levels of Ilp2 peptide in the hemolymph in high-sugar conditions (Fig. 4e; RNAi reaches significance, while *p* = 0.1115 for the knockout) and to increased whole-body *4EBP* expression suggesting reduced peripheral insulin signaling (Supplementary Fig. 3d). Our results indicate that Bursicon signaling via the Rk receptor stimulates insulin production and release from IPCs and suggest that the developmental delay induced by IPC-specific *rk* loss on a high-sugar diet is attributable to diminished IPC activity and reduced circulating insulin levels. To directly test this hypothesis, we employed the thermosensitive cation channel Transient Receptor Potential A1 (TrpA1)[39] to artificially trigger IPC activity. The developmental delay caused by *rk* knockdown in the IPCs was completely reversed by the ectopic activation of these cells through TrpA1 (Fig. 4f). This rescue demonstrates that it is indeed the attenuated IPC activity, due to impaired Bursicon-Rk signaling, that is responsible for the observed phenotypic effects.

The activation of Rk by the Burs:Pburs heterodimer leads to increased production of cAMP[32], a second messenger that in mammalian beta cells promotes glucose-stimulated insulin secretion[40]. Since many effects of cAMP are mediated by cAMP-dependent protein kinase (PKA), we investigated the impact of blocking PKA signaling in the IPCs. Inhibition of PKA through expression of a dominant-negative (cAMP-insensitive) form of the regulatory subunit R1 in the IPCs led to an HSD-dependent delay (Fig. 4g), similar to that observed with loss of neuronal *Burs/Pburs* or IPC-specific loss of *rk*. This indicates that the insulinotropic effects of signaling from Burs neurons to the IPCs through Rk could possibly be mediated, at least in part, by cAMP/PKA signaling. Similar to the effects of neuronal *Burs* or *Pburs* loss, IPC-specific knockdown or knockout of *rk* led to reduction in TAG levels under HSD conditions (Fig. 4h), and to increased circulating sugar (Fig. 4i), indicating that neuronal Burs signaling regulates fat storage and glycemic control via the IPCs.

## Systemic Bursicon signaling to fat cells protects against insulin resistance to maintain glucose homeostasis on high sugar diets

In addition to its signaling role within the CNS, Bursicon (Burs:Pburs) hormone is released into the circulation from neurohemal release sites[41], suggesting a possible humoral role of Bursicon in metabolic adaptations to high sugar. To identify the target tissues mediating any effect of humoral Bursicon signaling on fat storage, we assessed the tissue expression of *rk* in the FlyAtlas transcriptomic database[34] and found expression in the fat body, an adipose-like organ that stores excess energy as TAG and glycogen. This receptor expression suggested that direct action of Bursicon on the adipose cells might cause the observed lipid-storage effects of *Burs/Pburs* loss. Therefore, we investigated whether Rk in the fat body is involved in Bursicon-mediated regulation of sugar tolerance. Knockdown or knockout of *rk* in the larval fat body, using the fat-body-specific drivers *Cg-GAL4* (*Cg >* ) or *ppl-GAL4* (*ppl >* ), indeed led to an HSD-induced delay (Fig. 5a

and Supplementary Fig. 4a), with RNAi-and CRISPR-line effects on this phenotype ruled out (Supplementary Fig. 4b), suggesting that loss of Bursicon signaling in the fat body impairs glucose tolerance. We then assessed whether this sugar intolerance was associated with an inability to store excess energy as fat. Loss of *rk* in the fat body had no effect on lipid levels on a normal diet, but this manipulation significantly attenuated the increase in TAG associated with high-sugar feeding (Fig. 5b). In mammalian and *Drosophila* adipose tissue, stored fat is deposited within specialized lipid droplet organelles. We, therefore, investigated the effect of fat-body Bursicon/Rickets signaling on lipid storage droplets in response to high-sugar feeding. Consistent with the elevated TAG levels observed in these conditions, we found that high-sugar feeding increased the area of fat-body cells that was occupied by lipid droplets in control animals (Fig. 5c, d), indicating increased fat storage. In animals with *rk* knockdown, however, the lipid-droplet area remained unchanged in response to high-sugar feeding, indicating that loss of Bursicon signaling in the adipose tissue is associated with an inability of the tissue to process the excess energy and store it as fat. We next targeted the activation of Burs-Rk signaling specifically to the fat body by expressing a membrane-tethered Bursicon heterodimer (*tet-Burs*), consisting of a tandem Pburs::Burs fusion anchored in the cell membrane via an N-terminal transmembrane domain, to activate Rk signaling within this tissue[42]. Consistent with the role of Bursicon in promoting lipid storage, we found that fat-body-specifically activating Rickets with this construct resulted in an increased lipid-droplet storage area, even under normal sugar conditions (Fig. 5c, d). This mirrors the effects observed in control animals under high-sugar-diet (HSD) feeding, under which conditions Bursicon signaling is required for increasing fat storage. These results suggest that Bursicon signaling through its cognate LGR receptor Rickets is important for the absorption of excess circulating glucose by the fat body or for the conversion of intracellular carbohydrates into triglycerides for storage. We therefore examined whether fat-body Bursicon/Rickets signaling might affect circulating sugar concentrations and found that *rk* loss in the fat body exacerbates HSD-induced hyperglycemia (Fig. 5e), consistent with impaired glucose uptake into the fat tissue. Such hyperglycemic effects are often caused by insulin resistance, which reduces insulin-stimulated glucose uptake into tissues. To examine whether loss of Bursicon signaling in adipocytes impairs insulin signaling, we measured the responsiveness of fat-body tissues to insulin using tGPH, a fluorescent indicator that comprises a GFP moiety fused to a PI(3)P-binding Pleckstrin-homology domain and thus reflects insulin/PI3K activity by increased membrane association[43]. Fat-body cells with *rk* loss showed a stronger reduction in cell-membrane localization of GFP in response to high-sugar feeding compared to controls, indicating that inhibition of Bursicon signaling in adipose tissue reduces insulin/PI3K activity on high-sugar diet (Fig. 5f, g). These results indicate that loss of Bursicon signaling in fat cells leads to reduced basal insulin signaling – either central production or peripheral sensitivity – in HSD conditions. To identify whether insulin sensitivity was reduced in adipose tissue with loss of Bursicon signaling, we measured membrane GFP localization in adipocytes in response to ex-vivo stimulation with human insulin. Fat tissue from control animals fed HSD responded to exogenous insulin with an

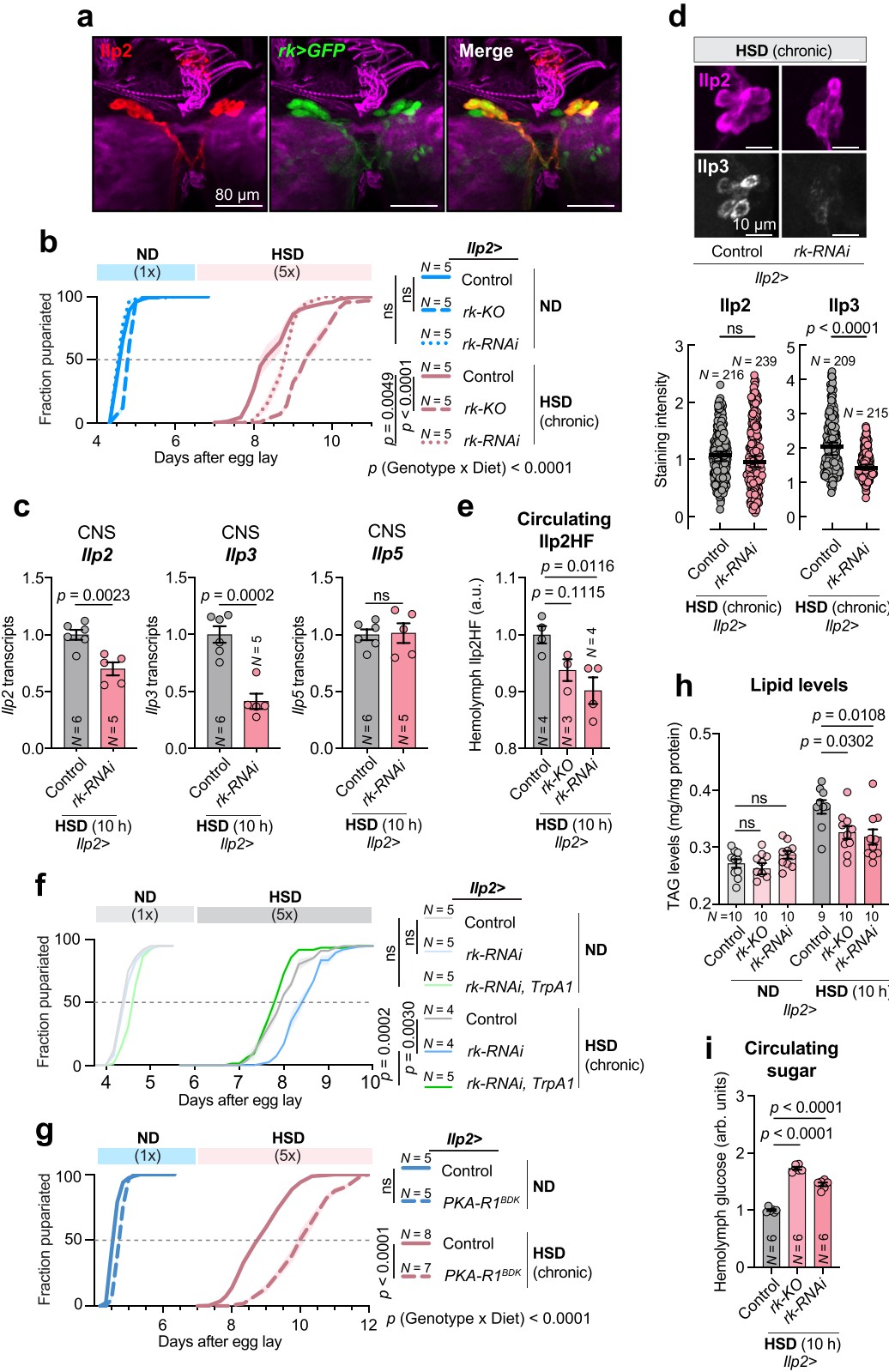

increase in insulin/PI3K activity and thus membrane GFP localization, whereas fat bodies with *rk* knockdown showed no response to this insulin stimulation (Fig. 5h and Supplementary Fig. 4c). Taken together, these observations suggest that fat-body loss of Bursicon signaling results in systemic glucose intolerance and lack of glycemic control, coupled with adipocyte insulin resistance, under high-sugar-diet conditions.

Insulin resistance is a main component of the pathogenesis of diabetes, and the mechanisms involved include cellular stress and inflammatory responses. The upregulation of expression of the lipocalin Neural Lazarillo (NLaz, mammalian ApoD) through the Jun-N-terminal kinase (JNK) pathway is proposed to be a part of the cellular stress-response pathway that contributes to the development of insulin resistance in HSD – indeed, NLaz is necessary and sufficient for

**Fig. 4 | Rickets-mediated signaling regulates the IPCs and phenocopies Burs manipulations in *Drosophila*. a** Larval brain stained for Ilp2, *rickets-GAL4 > GFP*, and actin. Scale bars, 80 microns. Images are representative of five independent experiments with similar results. **b** Pupariation timing of controls and animals with IPC-specific *rickets* loss, on normal diet (ND) and high-sugar diet (HSD). **c** Transcript levels of *Ilp2*, −*3*, and −*5* in CNS preps from animals with IPC-specific *rickets* knockdown raised on ND for 90 h and then fed HSD for 10 h. **d** Representative images of IPC Ilp levels and quantification in controls and animals with IPC-specific *rickets* knockdown, raised on HSD. Scale bars: 10 microns. **e** ELISA against circulating Ilp2 in controls and animals with IPC-specific *rickets* loss, raised on ND for 90 h and fed HSD for 10 h. **f** Pupariation timing of *Ilp2>* controls, *Ilp2>* with *rk-RNAi*, and *Ilp2>* with *rk-RNAi* and concurrent *TrpA1* expression under ND and HSD conditions. **g** Pupariation timing of controls and animals with IPC-specific expression of dominant-negative PKA regulatory subunit (*PKA-R1^BDK*), raised on ND or HSD.

**h, i** Whole-larval lipid-storage levels (**h**) and hemolymph glucose levels (**i**) of controls and IPC-specific *rickets* knockdown animals, raised for 90 h on ND and then fed a 10-hour pulse of ND or HSD. Animals in **a**, **c**, and **d** were raised at 25 °C, while all others were raised at 29 °C to enhance RNAi and CRISPR efficiencies. Data are presented as means; SEM is shown with shaded areas in (**b**, **f**, **g**) and error bars in other figures. Data are from biologically independent replicates, and sample sizes (*N*) are indicated on graphs. ns, not significant ($p > 0.05$). **b** and **f**, one-way ANOVA with Tukey's correction for multiple comparisons between 50%-pupariation times and two-way ANOVA for interaction. 50% pupariation time was determined via linear interpolation between adjacent observations below/before and above/after 50%. **c**, two-sided unpaired t-tests. **d**, **e**, two-sided Mann–Whitney nonparametric U tests. **g** two-sided unpaired t-test between 50% times and two-way ANOVA for interaction. **h**, **i**, one-way ANOVA with Tukey's correction for multiple comparisons. Source data are provided as a Source Data file.

this effect in the fly[10]. We, therefore, assessed whether Bursicon signaling affects *NLaz* expression in the fat tissue. Knockdown of *rk* in the fat body led to *NLaz* upregulation in whole-animal assays (Fig. 5i), consistent with insulin resistance. We then tested the ability of *NLaz* RNAi to rescue the defect in glycemic control induced by *rk* loss in HSD conditions and found that reducing *NLaz* expression in the fat body alone completely alleviated the hyperglycemia caused by *rk* knockdown in that tissue under high-sugar conditions (Fig. 5j). This indicates that reducing *NLaz* expression in fat tissue lacking *rk* is sufficient to restore glycemic control. Consequently, we investigated whether *NLaz* loss could also reverse other high-sugar-induced phenotypes observed in animals with fat-specific *rk* knockdown. We noted a significant increase in *InR* expression in animals expressing fat-body knockout of *rk* when *NLaz* was concurrently silenced, a trend observed even when RNAi was induced for only 10 h (Fig. 5k and Supplementary Fig. 4d). This suggests that *NLaz* knockdown may enhance the sensitivity of cells to insulin when Bursicon signaling is diminished, potentially explaining the reversal of hyperglycemia, since enhanced insulin signaling facilitates increased cellular glucose uptake from circulation. Speculating further, we questioned whether this would also manifest in the reversal of the ability to store excess energy as fat, given that glucose can be converted into fat for storage. Consistent with the potential role of NLaz inhibition in enhancing insulin signaling, we found that *NLaz* knockdown increased the TAG levels of animals with fat-body-specific *rk* loss (Fig. 5l and Supplementary Fig. 4e), which have reduced ability to store fat in high-sugar conditions (Fig. 5b). Insulin signaling regulates glucose uptake and lipogenesis through pathways distinct from FOXO-mediated regulation of *4EBP*, which primarily controls growth. Regulation of glucose metabolism involves post-translational changes to glucose transporters and glycogen synthase kinase 3, while the regulation of growth is mediated by FOXO transcriptional changes including regulation of *4EBP*, a negative regulator of growth[35]. Consequently, depending on the point of interaction of NLaz with the insulin pathway, loss of *NLaz* might only improve certain aspects of the insulin-resistance phenotype. Examining the impact of *NLaz* inhibition on the growth regulator *4EBP* revealed that in animals with fat-specific *rk* knockout, the simultaneous silencing of *NLaz* had no effect or even increased *4EBP* expression in the short term (Fig. 5m and Supplementary Fig. 4f). Thus, while *NLaz* inhibition may ameliorate some metabolic phenotypes downstream of Rk loss, it might not resolve the impairment in the growth-regulatory branch of the insulin pathway that involves 4EBP. Fat-body knockout of *rk* induced late in the second larval instar resulted in significant developmental delay when animals were transferred to a high-sugar diet (Fig. 5n), consistent with our earlier observations. This delay was not strongly attenuated in *rk*-knockout animals simultaneously expressing *NLaz* knockdown, indicating that NLaz loss does not efficiently rescue the growth defects caused by fat-body *rk* knockdown under high-sugar conditions, in line with the *4EBP*-transcription result. These findings position NLaz downstream of Burs/Rickets with respect to at least some aspects of

insulin signaling, suggesting that Burs/Rickets activity normally suppresses NLaz expression under high-sugar diet conditions and this inhibition is important for maintaining glycemic and metabolic control in such environments.

## Mammalian Rickets ortholog LGR4 promotes insulin sensitivity and signaling in mouse adipocytes

These results elucidate the role of the *Drosophila* LGR type B receptor Rickets/dLgr2 in modulating insulin sensitivity in fat-body cells. To extend these findings to a mammalian model, we probed the importance of a mammalian Rickets ortholog, LGR4, in mouse adipocyte insulin sensitivity. We knocked down *Lgr4* expression in mature mouse white adipocytes using two specific siRNAs (Lgr4 KD1 and KD2), both of which were effective in reducing *Lgr4* transcript abundance (Fig. 6a). The knockdown of *Lgr4* did not significantly alter *Insulin receptor* (*Insr*) transcript levels (Fig. 6b). However, knockdown adipocytes exhibited reduced TAG levels (Fig. 6c), mirroring the effect of *rickets* knockdown in *Drosophila* fat-body tissue (Fig. 5b). Knockdown adipocytes also displayed a pronounced loss of insulin-pathway activation after stimulation with insulin: control adipocytes (treated with siRNA that has no endogenous target) showed robustly increased pAKT levels following insulin stimulation, whereas adipocytes treated with siRNA targeting *Lgr4* did not exhibit any change in this readout in response to insulin stimulation (Fig. 6d). This lack of effect indicates that *Lgr4* loss impairs the cellular insulin response, leading to insulin resistance in knockdown adipocytes, similar to the effects observed in *Drosophila* fat-body cells expressing *rickets* knockdown (Fig. 5h). We further expanded our investigation to assess the effects of stimulation with both insulin and R-spondin 1 (RSPO1), one of the endogenous ligands of LGR4[44], as well as with RSPO1 alone. The results of these treatments were comparable to those elicited by insulin stimulation alone, suggesting that simultaneous activation of Lgr4 along with insulin stimulation does not acutely potentiate the insulin-signaling response (Fig. 6d). This observation suggests that the timing of LGR4 activation may be critical for its functional impact on insulin-mediated pathways. Sequential or chronic LGR4 activation, rather than concurrent short-term stimulation, may be necessary to observe a potentiating effect on insulin sensitivity in mouse adipocytes.

Next, we adopted a quantitative mass-spectrometric phosphoproteomics approach, leveraging 18-plex protein labeling technology to allow simultaneous analysis of multiple conditions in triplicate, to further unravel the LGR4 dependency of the global phosphoprotein response to insulin in mammalian adipocytes (schematized in Fig. 6e). Our phosphoproteomics data show a significantly blunted insulin response in adipocytes lacking *Lgr4*: cells treated with no-target control siRNA exhibited a robust number of phosphorylation changes after a 30-minute insulin stimulation ("changes" here defined as a > 30% difference in phosphorylation in either direction between insulin-stimulated cells and mock-stimulated controls), whereas far fewer insulin-induced changes were detected in *Lgr4*-knockdown cells

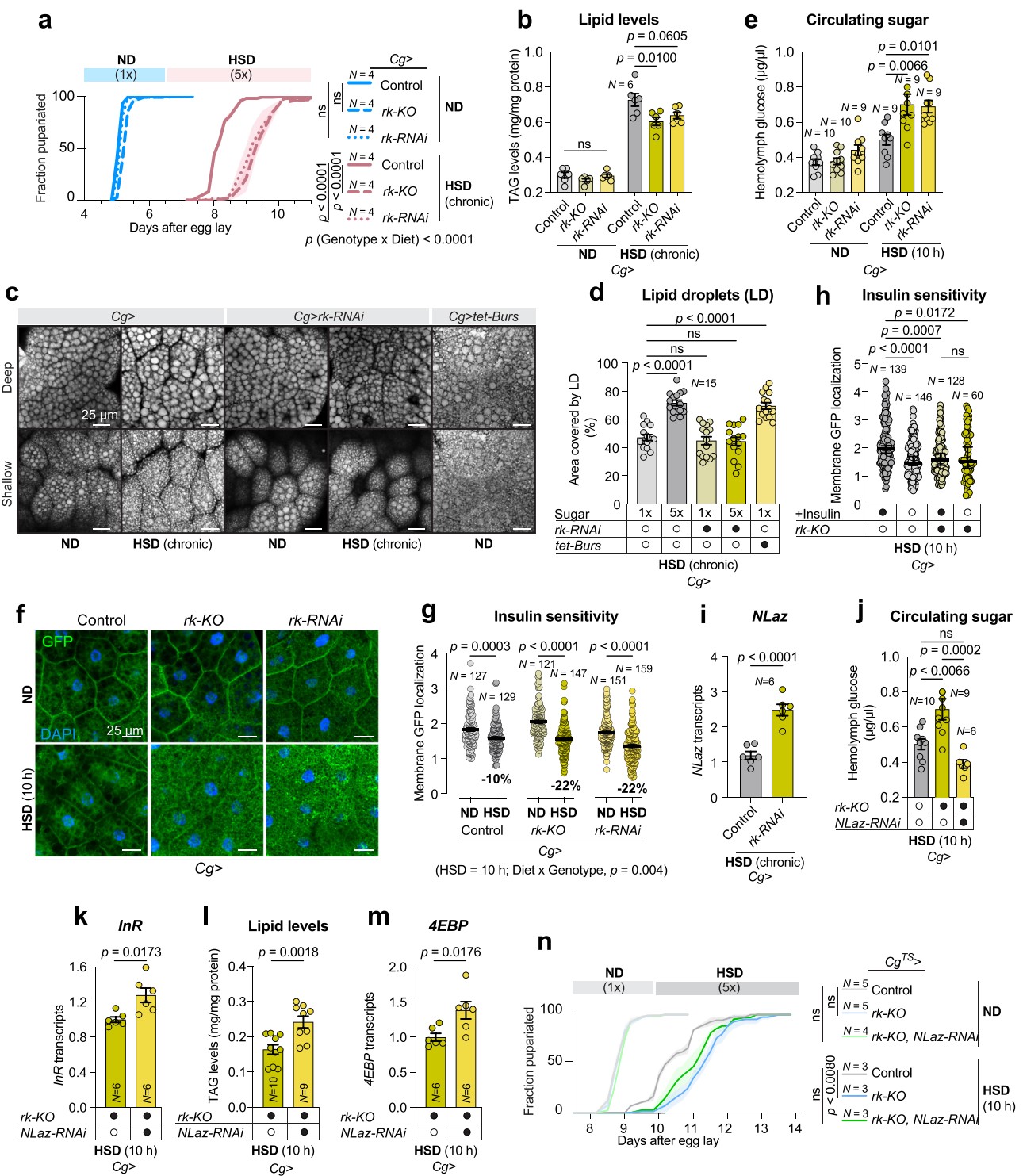

(Fig. 6f, g and Supplementary Data 2). Insulin induced 1974 significant phosphorylation changes (967 upregulated phosphorylations and 1007 down-regulated) in control adipocytes, whereas stimulated *Lgr4-KD1*-treated knockdown adipocytes exhibited only 743 such changes (381 up-regulations and 362 down-regulations), and cells treated with siRNA *Lgr4-KD2* showed just 557 insulin-induced changes (367 up-regulated and 190 down-regulated). The response to insulin was strongly dependent on LGR4, as 1,273 of the regulated protein phosphorylations (592 upregulated and 681 downregulated), which correspond to 64% of the total response to insulin stimulation in the control group, exhibited no response to this treatment in *Lgr4*-deficient mouse adipocytes. Comparisons at the level of individual phosphorylation events between control and *Lgr4*-knockdown adipocytes revealed a substantially reduced insulin signaling response in the absence of LGR4 (Fig. 6h–k). Most proteins that showed altered phosphorylation in response to insulin in controls responded much less strongly in cells lacking *Lgr4*, suggesting that marked insulin resistance arises in these adipocytes following loss of *Lgr4* activity.

Reactome pathway analysis of the proteins phosphorylated in response to insulin in control adipocytes highlighted the expected significant enrichment of proteins involved in insulin signaling and pathways regulating cellular metabolism (Fig. 6l and Supplementary

**Fig. 5 | Interplay of Rickets and lipocalin NLaz in the fat body maintains metabolic control on high-sugar diets in *Drosophila*. a** Pupariation timing of animals raised on normal diet (ND) or high-sugar diet (HSD). **b–d** Lipid-storage levels of whole-body triacylglyceride (TAG) (**b**) and lipid droplets in fat-body tissues (c,d) of animals raised on ND or HSD. Scale bars, 25 microns. **e** Circulating glucose levels of animals after 10-hour HSD feeding. **f–h** tGPH in-vivo insulin-signaling indicator images (**f**) and ratio of membrane GFP signal (**g, h**) in animals after 10-hour HSD feeding. Scale bars, 25 microns. **i** *NLaz* transcript levels in whole animals raised on HSD. **j** Hemolymph glucose levels in animals after 10-hour HSD feeding. Datasets in Figs. 5**e** and **j** were measured concurrently but are presented in separate panels for clarity; the controls and *rk-KO* genotype are shared between them. **k** *InR* transcript levels in whole larvae after 10-hour HSD feeding. **l** TAG levels measured in whole animals under HSD conditions. **m** *4EBP* transcript levels in whole larvae after HSD exposure. **n** Pupariation timing in animals with fat-body-specific expression

activated at the time of transfer to HSD (144 h after egg lay) by shifting animals from 18 °C to 29 °C. All animals were raised at 25 °C, except for those in (**n**). Data are presented as means; SEM is shown with shaded areas in (**a, n**) and error bars in other figures. Data are from biologically independent replicates, and sample sizes (*N*) are indicated on graphs. ns, not significant ($p > 0.05$); **a, n**: one-way Kruskal-Wallis nonparametric ANOVA with Dunn's multiple comparison test between 50%-pupariation times for multiple comparisons and two-way ANOVA for interaction. 50% pupariation time was determined via linear interpolation between adjacent observations below/before and above/after 50%. **b, d, e, h**: one-way ANOVA with Dunnett's multiple-comparisons test. **g**, two-way ANOVA and one-way ANOVA with Tukey's multiple-comparisons test. **h**, one-way Kruskal-Wallis nonparametric ANOVA. **i, k, l, m**: two-sided unpaired t-test. **j**, one-way ANOVA with Tukey's multiple-comparisons test. Source data are provided as a Source Data file.

Data 3). In contrast, insulin stimulation of adipocytes subjected to siRNA treatment targeting *Lgr4* evoked a diminished phosphorylation response in these pathways. Taken together, our findings clearly show that, like its ortholog Rickets/dLgr2 in the *Drosophila* fat body, LGR4 is critical for maintaining normal insulin sensitivity and signaling flux through the insulin-signaling cascade in mouse adipocytes.

## Bursicon mediates a muscle-neuronal-adipose relay that controls insulin signaling and adipocyte insulin sensitivity

Since neuronal Bursicon signaling is required for sugar tolerance, we investigated whether Burs⁺ neurons respond to dietary sugar. Burs is expressed in pairs of neurons in the subesophageal zone (SEZ) and the thoracic and abdominal segments in the larval CNS, whereas the expression of Pburs is confined to a subset of Burs⁺ neurons in the abdominal segments[32]. Consequently, the production of the active heterodimeric Bursicon hormone within the CNS is restricted to those abdominal neurons that express both the Burs and Pburs subunits. We used the CaLexA system[45], including a coexpressed *UAS-tdTomato* as a ratiometric control for *GAL4* expression and protein synthesis, to explore whether high-sugar feeding regulates calcium activity in the bilateral neurons in the thoracic (T3) and abdominal segments (A1-4) that co-express Burs and Pburs[32]. Under HSD conditions, a strong calcium-induced GFP signal (normalized to tdTomato) was exclusively observed in those Burs⁺ neurons that produce the Burs:Pburs heterodimer (Fig. 7a). This indicates that these Burs⁺ neurons are active when dietary sugar levels are high. We then quantified calcium activity of these neurons in response to dietary sugar. Chronic high-sugar feeding increased calcium signaling (normalized to tdTomato) but did not affect intracellular Burs peptide levels in these cells, demonstrating that the subset of neurons producing the Burs:Pburs heterodimer are responsive to dietary sugar (Fig. 7b–d). Conversely, Burs peptide levels were reduced in the SEZ and thoracic pairs of neurons that do not express Pburs (Fig.7e), indicating that neurons that might produce the possible Burs homodimer are affected differently by dietary sugar feeding.

The BMP ligand Glass bottom boat (Gbb) has been shown to activate neuronal expression of Pburs[46]. Muscle-derived Gbb activates the TGF-β receptors Wishful thinking (Wit) and Thickveins (Tkv) in Burs⁺ neurons, providing a retrograde signal from the muscle that upregulates neuronal Pburs. Notably, Tkv and other components of the TGF-β pathway were identified as hits in our screen for genes involved in sugar tolerance (Fig. 1e, d). We therefore examined whether muscle-derived Gbb regulates neuronal Bursicon signaling under high-sugar-feeding conditions. We found that knockdown of the Gbb receptor *tkv* in the Burs⁺ neurons reduced the expression of *Pburs* in the CNS (Supplementary Fig. 4g), confirming previous observations that muscle-derived Gbb acts in a retrograde manner to regulate Bursicon heterodimer signaling in the CNS[46]. To further investigate the role of muscle-to-neuron communication in regulating glucose-dependent Bursicon neuronal activity, we recorded GCaMP6s[47]

fluorescence (again, normalized to coexpressed tdTomato) in the Burs⁺ neurons of larval preparations in which these neurons maintained their neuromuscular connection (Fig. 7f). When these intact preparations were incubated in high-glucose medium, the Bursicon (Burs+Pburs) neurons showed a high level of spontaneous activity, whereas incubation in low-sugar medium elicited a weaker signal. This indicates that these cells are responsive to cues related to sugar availability (Fig. 7f). However, when the neuronal connections between the muscles and the brain were disrupted in preparations incubated at high sugar, this glucose-induced activation was abolished, although depolarization of these cells with KCl at the end of recording indicated that they were still functional. These findings imply that acute retrograde signaling from muscle to the Bursicon neurons is essential for these neurons' activation by high glucose.

We then examined the role of Gbb in the muscle as a potential peripheral regulator of central insulin signaling in the nervous system. Knockdown of *gbb* in the muscles or of *wit* or *tkv* in the Burs⁺ neurons resulted in reduced expression of *Ilp2* and *Ilp3* (Fig. 7g, h: note that the *Ilp2* reduction with *tkv-RNAi* does not reach statistical significance, $p = 0.0609$), mirroring the effects observed in animals with neuronal loss of *Burs* or *Pburs* and in those with loss of *rk* in the IPCs themselves (Figs. 3a, 4c). This suggests that muscles are the source of Gbb that regulates insulin through Bursicon in response to high-sugar feeding. To further delineate the functional relationship between Burs⁺-neuronal activity and IPC physiology, we used TrpA1 to artificially induce release from the Burs⁺ neurons. Ectopic activation of the Burs⁺ neurons by TrpA1 led to a significant depletion of Ilp2 peptide staining levels in the IPCs (Fig. 7i). Notably, this reduction occurred without corresponding changes in *Ilp2* transcript levels (Fig. 7j), suggesting that the decrease in Ilp2 content was due to increased IPC release, rather than diminished synthesis, occurring downstream of Burs⁺-neuron activation. Conversely, when *tkv* was knocked down specifically in the Burs⁺ neurons, there was an observable accumulation of Ilp2 peptide within the IPCs. This was coupled with a concomitant decrease in the expression levels of both *Ilp2* and *Ilp3* in the CNS, indicating a reduction in both the synthesis and the release of insulin-like peptides. Remarkably, TrpA1-mediated activation of these *tkv*-knockdown Burs⁺ neurons was able to fully restore CNS *Ilp3* expression levels (Fig. 7j) and to promote Ilp2 release (Fig. 7i). These findings collectively reinforce the hypothesis that the Burs⁺ neurons positively regulate the IPCs, enhancing the expression and release of ILPs. Furthermore, the data implicate BMP signaling within the Burs⁺ neurons as a critical promoter of this modulatory effect.

We next examined whether Gbb production in the musculature is regulated by high-sugar feeding by measuring *gbb* transcript levels in dissected body-wall muscles included in the carcass. *Gbb* expression was mildly but significantly upregulated in muscles in response to a chronic HSD (Fig. 7k). Dietary sugar can inhibit FOXO activity in muscles through insulin-mediated signaling pathways. We therefore wondered whether reducing FOXO action in the muscle would promote

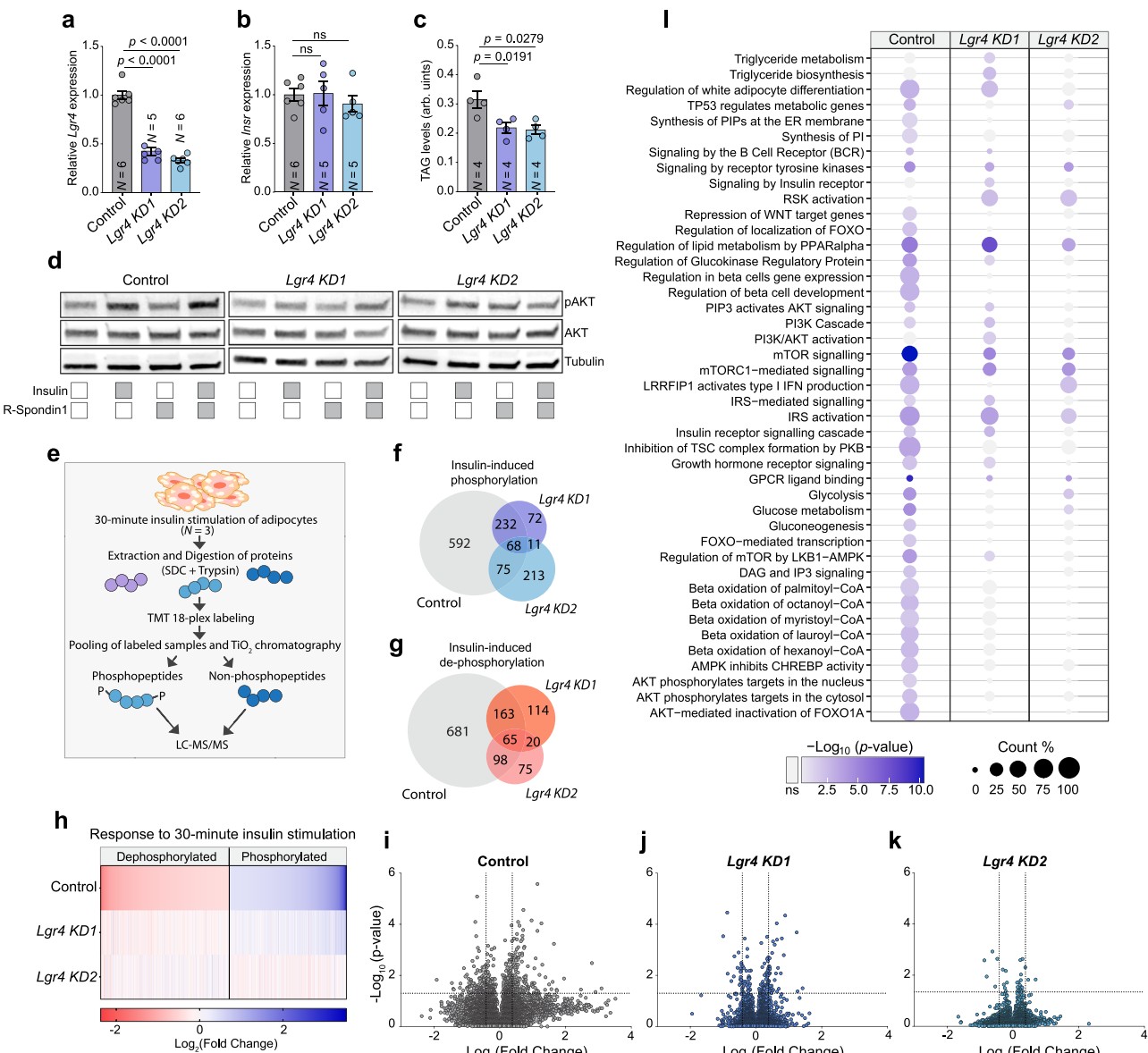

**Fig. 6 | LGR4 modulates insulin sensitivity in cultured mouse adipocytes.**
**a**, **b** *Lgr4* transcript abundance (**a**) in mouse white adipocytes using two specific siRNAs (*Lgr4-KD1* and *Lgr4-KD2*), compared to controls, which were transfected with siRNA that has no endogenous target, without affecting *Insulin receptor* expression levels (**b**). **c** Triacylglyceride (TAG) levels in adipocytes. **d** Western-blot analysis showing impaired phosphorylation of Akt upon 30-minute stimulation with insulin peptide in *Lgr4*-deficient adipocytes compared to control. The lack of potentiation of the insulin-signaling response by concurrent RSPO1 stimulation suggests that the timing of Lgr4 activation may be relevant to its effects. Each image was acquired twice, with similar results. **e** Schematic of mass-spectrometric quantitative phosphoproteomics using 18-plex peptide labeling. **f**, **g** The number of proteins that respond to insulin stimulation with a change in phosphorylation of more than 30% up (**f**) or down (**g**) in each siRNA-treatment group. Knockdown of *Lgr4* significantly diminished the insulin response, with many fewer proteins/sites undergoing phosphorylation changes upon insulin treatment. $N = 3$ parallel cell treatments per condition. **h** Heat map showing fold change in protein

phosphorylation within 30 minutes of insulin stimulation among all phosphosites with a > 30% change (3481 phosphorylation events) compared to unstimulated controls of the same genotype shows a reduced level of response in *Lgr4*-deficient adipocytes. **i**–**k** Volcano plots of phosphoproteomic changes after 30-minute insulin stimulation in control and *Lgr4*-deficient adipocytes. Vertical dashed lines indicate a 30% change in protein phosphorylation, and horizontal dashed lines indicate $p < 0.05$. **l** Reactome pathway analysis showing selected pathways exhibiting phosphorylation-change enrichment in insulin-stimulated adipocytes. Data are presented as means, with error bars representing SEM from biologically independent replicates, and sample sizes ($N$) indicated on graphs. ns, not significant ($p > 0.05$); TAG, triacylglyceride. **a**–**c** One-way ANOVA with Dunnett's multiple comparisons. **i**–**k** The ANOVA test built into the Proteome Discoverer software package was used to generate $p$-values for all the phosphopeptides and proteins identified in database searches. **l** A two-sided Fisher's Exact Test was applied, and only pathways with $p$-value < 0.05 for enrichment were considered significant. Source data are provided as a Source Data file.

the expression of *gbb*. We tested this hypothesis by expressing RNAi against *FOXO* in the muscle during high-sugar feeding, and we found an upregulation of muscle *gbb* expression in animals with muscle-specific *FOXO* knockdown (Fig. 7l), indicating that FOXO inhibits *gbb* expression in the musculature. This suggests that FOXO in the muscle is a dietary-sugar-regulated effector that governs Gbb/BMP signaling.

Therefore, our results indicate that in high-sugar feeding conditions, FOXO-regulated muscle BMP/Gbb signaling remotely regulates insulin production in the IPCs through a neuronal relay mediated by Bursicon signaling. We then investigated whether this muscle-neuronal relay is important for sugar tolerance by examining the effect on HSD-induced phenotypes of silencing *gbb* in the muscles or its receptors in the Burs⁺

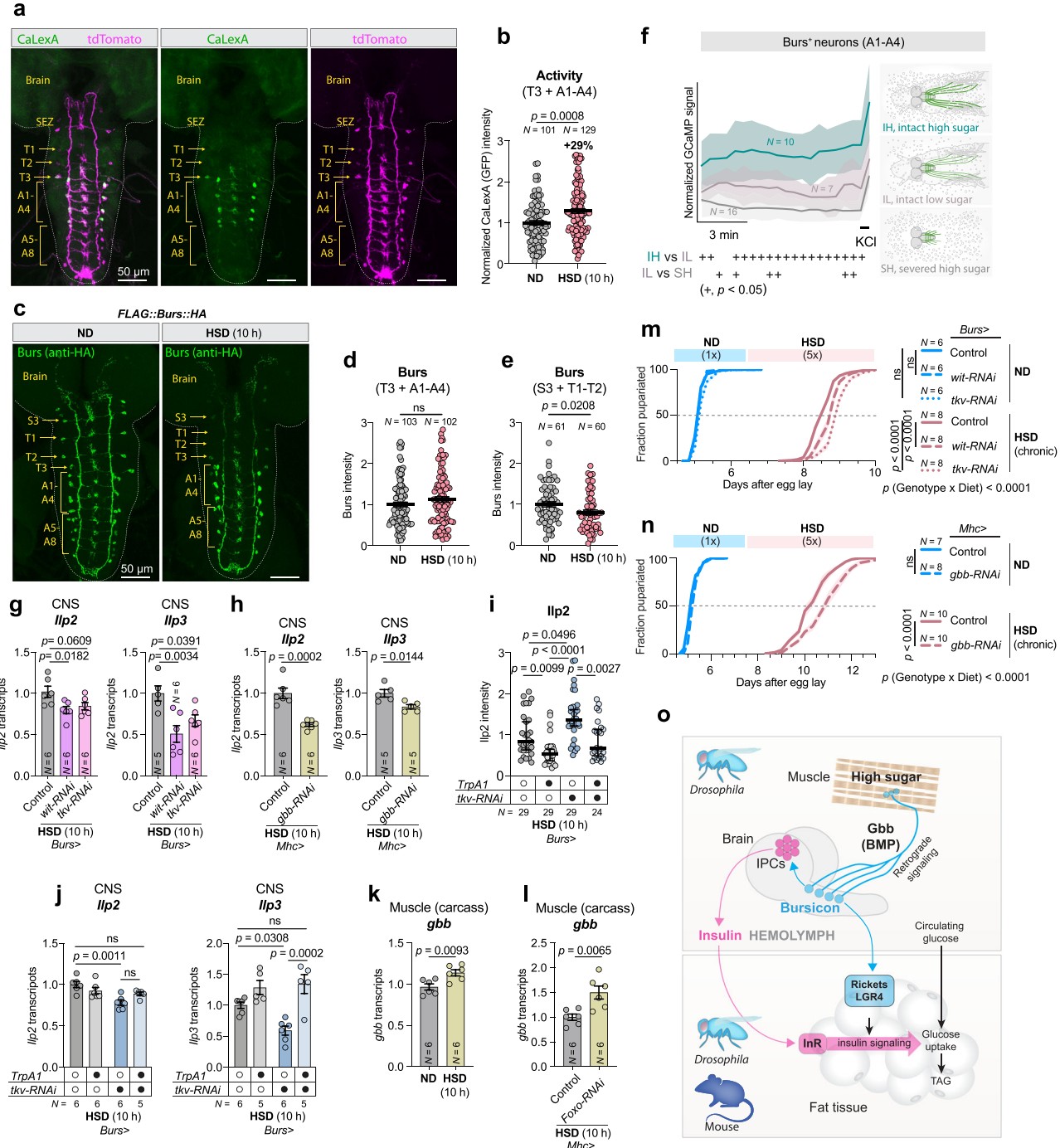

**Fig. 7 | High sugar activates a subset of Burs-expressing cells that project to the larval musculature via retrograde BMP signaling in *Drosophila*. a** Images of larval CNS from an animal expressing CaLexA and tdTomato in the *Burs*-expressing cells, raised on normal diet (ND) for 90 h and transferred to high-sugar diet (HSD) for 10 h. Segments within the CNS are labeled. Scale bars, 50 microns.
**b** Quantification of CaLexA signal in T3 + A1-A4 segment neurons after 10-hour HSD feeding. **c** Images of FLAG::Burs::HA staining in animals after 10-hour HSD feeding. Scale bars, 50 microns. **d**, **e** Quantification of Burs staining intensity after 10-hour HSD feeding in neurons of different neuromeres. **f** Real-time GCaMP6s fluorescence imaging of Burs+ neurons (A1-A4 segments) under varying glucose conditions and neuromuscular connectivity (*p*-values, Supplementary Fig 5c). **g**, **h** CNS transcript levels of *Ilp2* and *Ilp3*. **i** IPC Ilp2 staining intensity. **j** CNS *Ilp2* and *Ilp3* transcript levels. **k**, **l** Carcass *gbb* transcript levels in animals fed ND or HSD (**k**) and with muscle-specific *FOXO* RNAi (**l**). **m**, **n** Pupariation timing. **o** Model showing the regulatory links affecting HSD adaptations via muscle retrograde Gbb/BMP

signaling and Burs+ neuron activation; Burs stimulates the IPCs and promotes adipose insulin sensitivity. Rickets is orthologous with mammalian LGR4. Animals in (**a**–**f**) were raised at 25 °C, while all others were raised at 29 °C to enhance RNAi efficiencies and induce TrpA1 activation. Data are presented as means; SEM is shown with shaded areas in (**a**, **f**, **n**) and error bars in other figures. Data are from biologically independent replicates, and sample sizes (*N*) are indicated on graphs. ns, not significant (*p* > 0.05). **b**, **d**, **e**, two-sided Mann-Whitney nonparametric U tests. **f** Wilcoxon matched-pairs signed rank test (two-tailed). g, one-way ANOVA with Dunnett's multiple-comparisons test. **h**, **j**, **k**, **l**, two-sided unpaired t-tests. i, one-way Kruskal-Wallis nonparametric ANOVA with Dunn's multiple-comparisons test. **m**, **n** one-way ANOVAs with Tukey's multiple-comparisons test between 50%-pupariation times for multiple comparisons and two-way ANOVA for interaction. 50% pupariation time was determined via linear interpolation between adjacent observations below/before and above/after 50%. Source data are provided as a Source Data file.

neurons. We found that either muscle-specific loss of *gbb* or knockdown of its receptors *wit* or *tkv* in the Burs⁺ neurons resulted in prolongation of the HSD-induced delay (Fig. 7m, n), with no contribution from the RNAi-line background (Supplementary Fig. 5a, b), mimicking the phenotype observed with loss of neuronal *Burs*/*Pburs* and IPC- or fat-body-specific loss of *rk* (Figs. 2a, b, 4b, and Supplementary Fig. 2a–c). Although our results do not rule out developmental effects, collectively, they indicate that high-sugar feeding increases muscle-derived Gbb signaling onto Burs⁺/Pburs⁺ neurons of the CNS, leading to increased Bursicon signaling directly to the IPCs, which stimulates insulin production, and via the hemolymph to the fat body, which enhances its insulin sensitivity. Through these effects, Bursicon signaling acts to promote metabolic adaptation to sugar intake and thus to maintain glycemic control (Fig. 7o).

## Discussion

The global rise in obesity is fueling an increase in type-2 diabetes and other metabolic disorders, which are largely driven by resistance to insulin, the key hormone essential for governing glycemic levels and energy use and storage. Despite the importance of insulin resistance in pathogenesis, the molecular mechanisms underlying this phenomenon are not fully understood. In this study we performed a comprehensive in-vivo screen covering the secretome and receptome to identify hormonal mechanisms and pathways modulating sugar tolerance and cellular responses to insulin in a sugar-induced metabolic state characterized by obesity and resistance to insulin. Many of the identified genes have been associated with diabetes and obesity in humans (Fig. 1d, e). Our work provides direct functional evidence for the involvement in sugar metabolism of genes previously identified mostly through genome-wide association studies. Consequently, our findings offer a valuable resource for understanding how these genes and hormonal signaling routes contribute to metabolic disorders, enhancing our understanding of the complex genetic and endocrine crosstalk underlying these conditions. Our findings provide important insight into how interorgan communication between the musculature, the nervous system, and adipose tissue influences sugar tolerance and insulin resistance, broadening our understanding of the intricate metabolic and endocrine networks involved in glucose homeostasis and shedding light on mechanisms relevant to diabetes pathogenesis.

Our research findings on the *Drosophila* LGR-family receptor Rk (dLgr2) may contribute to understanding the role of its mammalian orthologs LGR4, -5, and -6 in human metabolism, particularly in the context of sugar tolerance, adipose-tissue sensitivity to insulin, and insulin secretion. We discovered that Rk is an important factor in metabolic regulation, promoting insulin production and adipose-tissue sensitivity to insulin. The known ligands for these mammalian receptors include the R-spondins and Norrin, a cystine-knot protein orthologous with *Drosophila* Burs and Pburs[44,48]. LGR4 in particular has been implicated in obesity-related metabolic dysfunction, and in humans, polymorphisms and gain-of-function mutations in this gene have been linked to obesity[49,50]. Although substantial genetic evidence links LGR4 variants to metabolic disorders in humans and mice (summarized in Supplementary Table 1), the precise mechanisms by which LGR4 influences metabolism are unclear. Mammalian *Lgr4* is expressed in diverse tissues, including the pancreas, liver, and adipose tissue, with tissue- and cell-type-specific signaling leading to tissue-specific physiological effects and therapeutic potentials[51]. In humans and mice, *Lgr4* mutations and expression levels have been directly associated with obesity and type-2 diabetes traits. For instance, gain-of-function mutations or upregulation of *Lgr4* expression in adipose tissue is linked to increased obesity[52], while loss-of-function mutations are associated with lower body weight[53]. Our findings in *Drosophila* show that the loss of signaling through the LGR4 ortholog Rk in adipocytes leads to reduced lipid levels under conditions of high dietary sugar. This suggests that LGR4 activity promotes fat storage, which

may explain the obesity-related LGR4 phenotypes observed in mammals. While our findings suggest *rk* gene expression in the larval fat body, based on indirect evidence from the FlyAtlas transcriptomic database[34] and corroborated by phenotypic changes observed in both knockdown and knockout models, we acknowledge a limitation of our study, which is the lack of direct demonstration of *rk* expression in the larval fat body.

Our discoveries indicate that LGR signaling may also affect insulin production and secretion. Loss of Bursicon-Rickets signaling in the IPCs, analogous to the mammalian β-cells, reduces insulin production and secretion, and our findings are compatible with a mechanism that involves Rk-mediated promotion of cAMP-PKA signaling. Potentially paralleling the Rk-insulin connection in the fly, *Lgr4* is expressed in β-cells[54], and stimulation of β-cells with R-spondin, a known activating ligand of LGR4, promotes insulin secretion and β-cell growth, indicating a potential regulatory effect of LGR4 on β-cell function and insulin secretion in humans. The effects of LGR4 signaling are hypothesized to be mediated by potentiation of Wnt/β-catenin signaling in β-cells[55,56]. However, LGR4 has also been suggested to induce signaling through the cAMP-PKA pathway, which is also known to stimulate insulin secretion in mammalian β-cells[57]. Based on our findings, it will be interesting for future studies to explore whether Lgr4 promotes insulin secretion from mammalian β-cells via activation of the cAMP-PKA pathway, and if it does, which ligands induce this activity.

Furthermore, we observed that loss of Rk signaling leads to insulin resistance in *Drosophila* adipocytes. If *Lgr4* loss in humans similarly induces insulin resistance, it could exacerbate the risk of developing diabetes, particularly type-2 diabetes, which is characterized by insulin resistance and impaired insulin secretion. This notion is supported by studies showing that *Lgr4* mutations are associated with increased glucose levels and type-2 diabetes occurrence (Supplementary Table 1) and findings that Lgr4 levels are associated with blood glucose control[58]. Our exploration in a mammalian model mirrored our *Drosophila* findings by showing that *Lgr4* reduction in mouse adipocytes leads to a decrease in adipocyte TAG levels and induces strong defects in insulin-signaling pathway activation. These results indicate that the absence of LGR4 cell-autonomously promotes adipocyte insulin resistance and therefore that LGR4 is essential for adipocyte insulin sensitivity. The lack of insulin-induced AKT phosphorylation observed in *Lgr4*-deficient mouse adipocytes suggests that LGR4 exerts its effects on insulin sensitivity at an early stage in the insulin-signaling pathway, possibly at the level of the receptor or a few steps downstream. This hypothesis is strongly supported by phosphoproteomics analysis, which reveals a remarkably blunted cellular response to insulin in *Lgr4*-deficient adipocytes. Although *Lgr4* knockdown did not alter the expression of the insulin receptor, it might influence receptor localization to the plasma membrane. Insights from our work suggest that *Lgr4* loss could elevate diabetes risk through mechanisms that involve effects on insulin resistance, glycemic levels, and the production and secretion of insulin. Future investigations should focus on elucidating the exact mechanisms by which LGR4 signaling regulates insulin sensitivity. This could be key for understanding the underlying causes of insulin resistance that lead to the development of diabetes. A critical question is whether the activation of LGR4 signaling can reverse insulin resistance, which could have significant implications for developing therapies to treat type-2 diabetes. However, several limitations must be considered, including the translational gap from *Drosophila* models and mammalian cell culture to human applications. Additionally, further studies are necessary to fully elucidate the mechanisms by which LGR4 promotes insulin sensitivity. The results from our RSPO1 co-stimulation experiments indicate that LGR4 activation does not instantaneously permit insulin response, suggesting that longer-term changes are required. Taken together, our work provides evidence for the importance of LGR4 signaling in metabolic regulation and suggests the intriguing possibility that modulation of

LGR4 signaling could potentiate insulin secretion and promote sensitivity to insulin in adipose tissue, improving glycemic control in high-sugar diet conditions that cause hyperglycemia. Establishing the significance of LGR4 in these processes could potentially impact the development of therapeutic strategies for metabolic disorders.

Our results also implicate the secreted lipocalin NLaz, homologous with human ApoD, as a possible factor in HSD-induced phenotypes and in their suppression by Bursicon signaling in adipose tissue. Previous reports have shown that NLaz affects longevity, stress resistance, and metabolism in *Drosophila*, and ApoD has been linked to obesity and insulin resistance, but the role of ApoD in diabetes and metabolic disorders remains to be clarified[10,59,60]. Our work suggests that downregulation of NLaz downstream of Rk-mediated Bursicon signaling in the adipose tissue is necessary for the maintenance of glycemic and metabolic control under conditions of high-sugar diet. These findings provide a molecular context for understanding the mechanisms by which NLaz/ApoD and Rk/Lgr4 signaling regulate metabolic signaling in obese-like states and suggest that modulation of NLaz/ApoD function might provide a strategy for treatment of metabolic disorders.

Maintaining metabolic homeostasis under nutritional stress requires a network of inter-organ crosstalk to ensure coordinated adaptive responses of different organs and to effectively balance the uptake, use, and storage of energy. Given this network's complexity, signaling routes that connect and coordinate the functions of organs to maintain metabolic homeostasis have remained difficult to elucidate. Our work here describes complex communication between muscle, neurons, and adipose tissue that is crucial for metabolic adaptations to a high-sugar diet. This supports an emerging paradigm in which metabolic control requires a coordinated effort by multiple organs that each sense different aspects of nutritional intake and metabolic state and relay information to other tissues to balance energy storage and mobilization. In both flies and mammals, fat-derived hormones (leptin or Unpaired-2) convey metabolic information and act via a neuronal relay to regulate insulin secretion[24,61]. Leptin also modulates the insulin sensitivity of skeletal muscles via central relays and thus mediates communication about the metabolic state of the fat tissue via neurons to both insulin-producing β-cells and the muscles to modulate their response to insulin. Our findings add a new axis to the model for regulating glucose homeostasis by demonstrating the existence of a muscle-derived signal that acts via a neuronal relay to modulate both insulin secretion and adipose tissue sensitivity to insulin. Our data suggest that, in flies, Gbb, a conserved BMP5/6/7/8 ortholog, is a sugar-regulated myokine that acts through its receptors (Tkv and Wit) on Bursicon-expressing neurons, which in turn drive both insulin production and adipose insulin sensitivity. All of the insulin- and glucose-homeostasis-regulating genes and pathways investigated in the present work have mammalian orthologs, which is not surprising given the evolutionary conservation of insulin signaling and the central importance of metabolic regulation. This suggests that these overall mechanisms and tissue-crosstalk routes might be conserved across species. The mammalian Gbb ortholog BMP7 has been shown to improve insulin signaling in insulin-resistant cells[62], suggesting that this might be a conserved function of Gbb/BMP7. Although the function of BMP7 in energy metabolism is not well characterized, it seems to act through leptin-independent mechanisms, making it of therapeutic interest in obesity, since the obese state is often characterized by leptin resistance[63]. Identifying the source of insulin-sensitivity-modulating BMP7 will be an interesting avenue of future research, as well examining whether the effects of BMP7 in this regard are mediated by downstream Lgr signaling.

In summary, this work unravels a muscle-neuronal-adipose communication mechanism that involves BMP, LGR, and ApoD components and pathways. This axis regulates glucose homeostasis under conditions that drive the pathological hallmarks of diabetes, including

tissue resistance to insulin, by governing both insulin production and the insulin sensitivity of adipose tissue. Uncovering these mechanisms is not only fundamentally important but may also facilitate the development of targeted interventions for obesity and related metabolic disorders.

## Methods

Our research complies with all relevant ethical regulations. The use of *Drosophila* followed standard procedures and does not require specific ethical approval. The 3T3-L1 mouse pre-adipocytes were differentiated into mature adipocytes using established methods, which also do not require specific ethical approval.

### *Drosophila* husbandry and stocks

Flies were maintained on a standard lab diet (8.2% cornmeal, 6% sucrose, 3.4% baker's yeast, and 0.8% agar, with 0.48% propionic acid and 0.16% methyl-4-hydroxybenzoate[10]) on a 12/12-hour light cycle at 25 °C and 60% relative humidity. This diet, defined as "1x sugar", is the basis for higher-sugar media: *e.g.*, for "5x sugar" diet, sucrose was increased to 30%, with other ingredients left unchanged. Stocks obtained from the University of Iowa Bloomington *Drosophila* Stock Center (BDSC) include *Burs^Z4410*, likely a null allele due to mis-splicing[64], #66432 (*w; arm-GFP; Burs^Z4410/TM6B, Tb^+*). *Arm-GFP* was removed, and the balancer was replaced with *TM6B, Tb. Burs-GAL4*, #40972; CaLexA system[45], #66542, modified by adding *10xUAS-IVS-myr::tdTomato[-su(Hw)attP8]* (#32223)[23]; *Cg-GAL4[65]*, #7011; *daughterless (da)-GAL4*, #55850; *Ilp2-GAL4[66]*, #37516; *Mhc-GAL4/TM3, Sb*, #55133, outcrossed five times to our standard lab genetic background to remove an unlinked homozygous-lethal allele and re-balanced over *TM6B, Hu Tb. R57C10-GAL4[67]*, #39171; *tGPH[43]* (*alphaTub84B-GFP::PH(Grb1))*, #8164; *trans-Tango* system[37], #77124; *Tub-GAL80^TS* (ref[68].), #7018 and 7019; *UAS-Cas9.P2*, #58985; *UAS-gbb-RNAi*, #34898; *UAS-GCaMP[47,69]* #42746; *UAS-PKA-R1^BDK* (ref[70].), #35550; *UAS-TrpA1*, #26263. Stocks obtained from Vienna *Drosophila* Resource Center (VDRC) include *UAS-Burs-RNAi*, # 102204; *UAS-Foxo-RNAi*, #107786; *UAS-NLaz-RNAi*, #101321; *UAS-Pburs-RNAi*, #102690; *UAS-rk-RNAi*, #105360; *UAS-tkv-RNAi*, #105834; and *UAS-wit-RNAi*, #103808. Other RNAi lines are listed in Table S1. *Rickets-GAL4[71]* was a kind gift of Ben White (NIH). *UAS-DILP2HF[72]* was a generous gift of Sangbin Park and Seung Kim (Stanford). *UAS-tet-Burs[42]* was kindly given by Alessandro Scopelliti (University of Edinburgh). A rigorous genetic standardization protocol for all GAL4 and GAL80 lines, as well as for combinations of transgenes, was implemented to ensure they shared a genetic background closely aligned with the *w^1118* strain. This standardization was achieved by initially crossing the balancer lines with our in-house *w^1118* strain to incorporate a genetic background analogous to *w^1118*. To introduce a controlled degree of genetic variation across all lines, we utilized at least 15 *w^1118* individuals for backcrossing purposes. Furthermore, by selecting females for these crosses, we standardized the genetic background of the X chromosome. As a result, transgenic combinations maintained this uniform genetic background, facilitating consistent experimental conditions and outcomes. This approach ensured that the GAL4 driver controls were genetically almost identical to the RNAi or CRISPR animals, with the exception of the RNAi or gRNA construct itself. This was accomplished by crossing the GAL4 drivers into the *w^1118* genetic background in which the UAS-RNAi or UAS-gRNA lines were maintained, thereby aligning the genetic bases of our driver controls closely with those of the RNAi or CRISPR experimental groups. Full genotypes of the animals used in each figure panel are provided in Supplementary Data 4. For all experiments, larvae of mixed sexes were used.

### High-sugar screen

A list of secretome and receptome genes was generated using the online resources GLAD[73] ("Secreted Proteins" and "Receptors") and

MetazSecKB[74] ("Highly likely secreted" and "Plasma Membrane" with probability ≥3). To expand the list of secreted proteins, we also included genes annotated as "Secreted" or associated with the GO terms "Extracellular region" (GO:0005576), "Extracellular space" (GO:0005615), or "Extracellular matrix" (GO:0031012) in FlyBase, UniProt, and Ensembl. These lists were merged to create one common list and cross-referenced with stock availability from Vienna *Drosophila* Resource Center (VDRC)[75]. One RNAi line was chosen for each gene, with lines from the KK collection preferred over those of the GD library. Additional lines from the University of Indiana Bloomington *Drosophila* Stock Center (BDSC) were included for genes from the GLAD Receptome list for which no VDRC RNAi stocks were available. The list of genes and RNAi lines is available as Supplementary Data 1.

Four males of each RNAi line were crossed to six *da-GAL4* virgin females in vials containing 1x- or 5x-sugar medium, and flies were allowed to seed the vials with eggs for 24 h at 25 °C. Flies were transferred to new vials at least twice for additional egg-lays. Adults were removed, and the vials were incubated at 25 degrees. The formation of prepupae or pupae (marked by visible cuticle darkening, Bainbridge and Bownes[76] stage 9) was recorded once each day. Several vials of each cross were scored, and the mean time until 50% pupariation was calculated for replicates that had a minimum of three pupae. Other defects, such as larval arrest, were also recorded. Genotypes that exhibited a phenotype on 5x sugar but not on 1x sugar – those that exhibited a sugar-dose-specific phenotype – were considered to be of interest for follow-up.

## Other pupariation assays

Crosses were set up in egg-laying chambers sealed with a 60-mm Petri dish containing apple-juice agar (1 L water, 340 mL apple juice, 30 g agar, 34 g glucose, 20 mL 10% Tegosept in ethanol). Animals were allowed to lay eggs for four h at 25 °C, and the plates were incubated for 24 h at 25 °C. Hatched larvae were transferred to vials containing 1x or 5x medium, with ~30 larvae per vial, and vials were incubated at 25 °C or 29 °C. Pupae (marked by completion of spiracle eversion and complete immobilization of animals) were counted every 4–8 h. At least 5 vials were scored for each genotype. The time of 50% pupariation was calculated by linearly interpolating between the measurements flanking this point.

## ELISA of circulating tagged Ilp2

A stock of *Ilp2-GAL4, UAS-Ilp2(HA,FLAG)*[72] was created, and this was crossed to animals carrying knockdown/knockout constructs. Four-hour egg-lays were made, and larvae were transferred 24 h later to vials containing 1x-sugar medium ( ~ 30 animals per vial). At 96 h after the midpoint of the egg-laying period, mid-third-instar larvae were recovered from the medium, washed with MilliQ water twice to remove any particles of the medium, and dried on a piece of paper towel; hemolymph was collected by cutting the larval cuticle with iridotomy scissors and collecting the exudate on an ice-cold shallow-welled glass slide. Samples were heat-treated at 60 °C for 5 minutes to denature phenoloxidase (melanization enzyme), centrifuged to remove any hemocytes, debris, or aggregates, and used in a sandwich HA/FLAG ELISA. Anti-FLAG (Sigma-Aldrich #F1804, 5 µg/mL in 200-mM NaHCO₃ buffer, pH 9.4) was adsorbed onto F8 MaxiSorp Nunc-Immuno modules (ThermoScientific #468667) overnight at 4 °C. Modules were washed twice with phosphate-buffered saline (PBS) + 0.1% Triton X-100 (PBST) and then were blocked for 2 h at room temperature with PBST + 4% non-fat dry milk. Modules were washed three times in PBST, and 1 µL of hemolymph or synthetic HA::spacer::FLAG peptide standard (DYKDDDDKGGGGSYPYDVPDYA) was diluted into 50 µL PBST + 25 ng/mL mouse anti-HA peroxidase (Roche, #12013819001) + 1% non-fat dry milk, added to the wells, and incubated overnight at 4 °C. The solution was removed, and the modules were washed six times, five minutes each, with PBST.

One hundred microliters of One-step Ultra TMB ELISA substrate (Thermo Scientific #34028) was added to each well, and the modules were incubated for 15 minutes at room temperature to permit color development. The reaction was terminated by the addition of 100 µL 2 M sulfuric acid, and the absorbance at 450 nm was measured using an EnSight plate reader (PerkinElmer).

## Metabolite analyses

Triacylglycerides (TAG) and glucose levels were measured using standard colorimetric methods[77,78]. Four-hour egg lays were performed, and at 24 h after egg laying (AEL), larvae were transferred to vials of 1x or 5x-sugar medium, ~30 per vial. In chronic-feeding experiments, vials were incubated at 25 °C until 96 h after egg laying for 1x sugar or 144 h AEL for 5x sugar, to equalize developmental progression; for short-term 5x-sugar feeding, animals were raised on 1x diet until 90 h AEL and then transferred to fresh 1x medium or to 5x-sugar diet for ten h (until 100 h AEL). Several samples each containing 4 larvae were collected in 260 µl PBST (PBS + 0.05% Tween-20, Sigma #1379) and homogenized using 5-mm steel beads (Qiagen #69989) in a bead mill (Qiagen TissueLyser LT). For TAG measurements, acylglyceride ester bonds were cleaved using Triglyceride Reagent (Sigma, #T2449) to liberate glycerol, and the concentration of this product was measured using Free Glycerol Reagent (Sigma, #F6428). Due to interruptions in this kit's distribution, we employed the Triglyceride Trigs kit (Randox, #TR210) for the experiments depicted in Fig. 5l and Supplementary Fig. 4e. This alternative kit is less sensitive in fly extracts, resulting in lower TAG measurements. Glucose was measured in a colorimetric assay (Sigma #GAGO20). For hemolymph assays, hemolymph was collected by cutting the cuticle with micro scissors and the exudate was collected with a pipette. Hemolymph was 10-fold diluted with PBS and heat-treated at 70 °C for 5 minutes to prevent melanization and centrifuged to remove pelleted aggregates. In TAG and glucose assays, the resulting absorbance at 540 nm was measured in a 384-well plate using an EnSight multimode plate reader (PerkinElmer). Protein concentrations were determined using a bicinchoninic acid assay (Sigma, #BCA1), and the resulting absorbance was read at 562 nm. Absorbances were converted to concentrations using standard curves for glucose, glycerol, and protein. TAG and glucose levels are reported as the ratio of those metabolites to protein to normalize for slight differences in larval size.

## Transgene construction

**Transgenes for inducible CRISPR-mediated knockout.** UAS-driven guide-RNA constructs targeting two sites in each of *rickets*, *Burs*, and *Pburs* were created in the vector pCFD6[79], obtained from AddGene (#73915; https://addgene.org/). For each gene, gRNA target sequences were identified using the E-CRISP algorithm[80] (https://e-crisp.org/E-CRISP/). A portion of pCFD6 was amplified by standard PCR using pairs of oligos, given in Supplementary Data 5, in which these gene-specific target sequences were inserted (underlined). Each resulting fragment was inserted into BbsI-digested pCFD6 using GeneArt Gibson Assembly HiFi Master Mix (ThermoFisher, #A46628). Clones were sequenced, and correctly assembled clones were midi-prepped using Qiagen kits and integrated into the genome at the *attP2* third-chromosome site by BestGene, Inc (Chino Hills, CA).

**Creation of endogenous FLAG::Burs::HA knock-in allele.** We designed a CRISPR-Cas9-mediated knock-in construct that adds a FLAG tag to the N-terminus and an HA tag to the C-terminus of mature secreted Burs. The FLAG tag sequence was inserted after the putative signal sequence[32,81], and the HA tag sequence was inserted right before the stop codon. Three silent mutations were introduced into the gRNA target sites within the coding sequence to prevent recutting of the insert. The *FLAG::Burs::HA* construct was flanked by 800-base-pair homology arms. The resulting knock-in sequence is given in

Supplementary Data 5. This sequence was synthesized by Thermo-Fisher, transformed into competent cells, and midi-prepped for microinjection. We designed three pairs of primers that contain sense or antisense gRNA sequences (see primer sequences in Supplementary Data 5) and cloned these into plasmid pCFD3[82]. Clones were sequenced, and one correct clone of each gRNA construct was midi-prepped. The knock-in construct and the three gRNA constructs were mixed 3:1:1:1 at a final total concentration of 500 ng/µl, and this plasmid solution was injected into *nos-Cas9* embryos in-house. Successful knock-in fly lines were identified by PCR confirming the presence of the FLAG-tag sequence. *FLAG::Burs::HA* knock-in animals are homozygous-viable and do not exhibit any obvious phenotype.

## Quantitative RT-PCR
For measurement of transcript levels, six samples each containing 5 whole larvae or 5 dissected larval tissues (central neuronal systems or body carcasses) were disrupted in 350 µl lysis buffer from the NucleoSpin RNA kit (Macherey-Nagel, #740955) containing 1% beta-mercaptoethanol, using a Qiagen TissueLyser LT bead mill and 5-mm stainless steel beads (Qiagen, #69989). RNA was isolated from the samples using the NucleoSpin RNA kit, and cDNA was produced using `the High-Capacity cDNA Synthesis Kit (Applied Biosystems, #4368814). Real-time PCR was performed using RealQ Plus 2× Master Mix Green (Ampliqon, #A324402) on a QuantStudio 5 machine (Applied Biosystems). Expression was calculated using the delta-delta-Ct method, with ribosomal-protein gene *Rp49* as control. The oligos are listed in Supplementary Data 5.

## Western blotting
Larvae were raised for 90 h AEL on 1x diet with further transfer onto either 1x or 5x diet for 10 h, until 100 h AEL at collecting, and samples of four larvae each were homogenized in 100 µl 2× SDS sample buffer (Bio-Rad #1610737) containing 5% (355 mM) 2-mercaptoethanol using a Qiagen bead mill (TissueLyser LT) and 5-mm steel beads. Homo-genates were denatured at 95 °C for 5 minutes, and insoluble material was pelleted by five minutes' centrifugation at maximum speed. Samples were loaded into a precast 4–20% polyacrylamide gradient gel (Bio-Rad) and electrophoresed. Separated proteins were transferred to polyvinylidene difluoride membrane (PVDF, Millipore) using a Trans-Blot Turbo Transfer Pack (Bio-Rad, #1704158) on a Trans-Blot Turbo setup (Bio-Rad). The membranes were incubated in Odyssey blocking buffer (LI-COR, #927-40100) for one hour at room temperature. The blocking solution was poured away, and the blots were incubated with rabbit anti-phospho-Akt (Cell Signaling Technology, #4054; 1:1000) and mouse anti-alpha-Tubulin (Sigma #T9026, diluted 1:4000) in Odyssey blocking buffer + 0.2% Tween-20. The membranes were rinsed and washed three times for 15 minutes each with PBST (PBS supplemented with 0.1% Tween 20) and incubated for 45 minutes with IRDye 800CW anti-rabbit (LI-COR #925−32210) and IRDye 680RD anti-mouse (LI-COR #925-68070) secondary antibodies, diluted 1:10,000 in Odyssey blocking buffer + 0.2% Tween-20. Membranes were rinsed and washed three times for 15 minutes each with PBST buffer, avoiding light, and imaged using an Odyssey Fc imaging system (LI-COR).

## Immunostaining
Larvae were raised at 25 °C until 90 h after egg-laying on 1x diet with further transfer onto 1x or 5x diet and kept for further 10 h. Larvae were extracted from the medium and rinsed with PBS. For ex-vivo insulin-stimulation assays, fat-body tissue was dissected in PBS on ice, rinsed twice with cold PBS and transferred into a shallow glass dish containing Schneider's medium. Either pure Schneider's ("mock") or Schneider's containing human insulin (Sigma-Aldrich #I9278, final concentration, 0.5 µM) was added, and tissues were incubated for 20 minutes at room temperature before fixation and further processing as below. For other experiments, tissues were dissected in cold PBS. Tissues were

transferred to 4% paraformaldehyde (PFA) in PBS and incubated at room temperature, with agitation, for one hour. After PFA removal, tissues were rinsed and washed three times, for 15 minutes each time, with PBSTx (PBS + 0.1% Triton X-100). Tissues were blocked for 30 minutes at room temperature in PBSTx with 5% normal goat serum (Sigma). Primary antibodies were diluted as detailed below in blocking buffer, and tissues were incubated in this solution overnight at 4 °C with agitation. Tissues were quickly rinsed with PBSTx and washed three times for 20 minutes each with PBSTx. Secondary antibodies were diluted 1:500 in PBSTx, and tissues were incubated in this solu-tion overnight at 4 °C with agitation. The secondary staining solution was removed, and tissues were rinsed once quickly with PBSTx and washed three times for 20 minutes each with PBSTx. If phalloidin was included in the experiment, the first wash contained Alexa Fluor 647-conjugated phalloidin, diluted 1:1000 from the manufacturer's recommended stock concentration. The last wash contained DAPI (Sigma, #D9542), diluted 1:500 in PBSTx. Tissues were mounted in ProLong Glass anti-fade mountant (Invitrogen, #P36984), using a 0.12-mm-thick spacer (Grace Bio-Lab, #654006) and glass cover slip, on poly-lysine-coated slides (VWR, #631-0107) that had been further coated with poly-L-lysine (Sigma, #P8920). Slides were allowed to cure for at least 24 h at 4 degrees. Tissues were imaged using a Zeiss LSM 900 confocal microscope using 20x and 40x objectives. Z-stacks were captured with a one-micron Z-step. Tiling and stitching using the Zen 3.1 (blue edition) software from Zeiss were in some cases used to cover the entire areas. Samples that were to be compared against one another were prepared at the same time, using the same reagent mixes, and were imaged using the same settings.

The following primary antibodies were used: mouse monoclonal anti-GFP (clone 3E6, ThermoFisher #A11120), 1:500; rat monoclonal anti-HA (clone 3F6, Roche #11867423001), 1:1000; rabbit anti-Ilp2[83], kind gift of Ernst Hafen and Michael Pankratz (University of Bonn), 1:1000; mouse anti-Ilp3[84], kind gift of Jan Veenstra (Bordeaux), 1:1000; rat anti-Ilp5[85], 1:500; and against tdTomato, rat anti-mCherry (Ther-moFisher, no. M11217), 1:1000. The following secondary antibodies were used, all diluted 1:500: Alexa Fluor 488-conjugated goat anti-mouse (ThermoFisher, #A32723); Alexa Fluor 555-conjugated goat anti-rabbit (ThermoFisher, #A32732); Alexa Fluor 555-conjugated goat anti-rat (ThermoFisher, #A21434); and Alexa Fluor 647-conjugated goat anti-rat (ThermoFisher, #A21247).

## Live imaging of Bursicon⁺ neurons in response to glucose
Ninety-six-hour-old *Burs>tdTomato, GCaMP6s*[47] larvae were filleted dorsally to expose the brain in a manner that maintained the functional connections between the body-wall muscles and the Bursicon neurons. To assess the impact of elevated glucose levels on Burs⁺ A1-A4 neurons, larvae were incubated in a modified HL3.1 solution without trehalose[30], containing either high (5 mM, 900 µg/mL) or low (1 mM, 180 µg/mL) glucose concentrations. To prevent imaging interference from body-wall muscle contractions, larvae were incubated in a medium con-taining the calcium-channel blocker isradipine (SigmaAldrich, # I6658) at a concentration of 10 µg/ml for 30 minutes prior to image acquisi-tion to ensure adequate sensing of glucose by the muscles. This con-centration allows selective depression of muscle contractions without affecting cellular calcium oscillations measured with GCaMP6s[86]. TdTomato signal and GCaMP6s fluorescence was then recorded on a Zeiss LSM 900 confocal microscope equipped with AiryScan2 at 30-second intervals over a 10-minute period. At the conclusion of each experiment, KCl was applied to confirm the continued survival of the ex-vivo brain preparation. The open-source image-processing package Fiji[87] was used to manually demarcate the anatomically distinct soma of Burs⁺ A1-A4 neurons, marked with tdTomato. The mean intensity of signal within each neuron was extracted in each frame, and to control for potential variation in GAL4 expression and for specimen-positioning-related artifacts, the mean GCaMP6s intensity was

normalized to the mean tdTomato in each neuron in each frame. The ratios thus obtained for each of the A1-A4 neurons in each frame were averaged to produce a single value for each time point.

## Image analysis

Image analysis was performed using FIJI/ImageJ[87], version 1.53t. For quantification of Ilp staining in the IPCs: Z-stacks were projected using the "sum" method. Each IPC cluster was manually segmented using the freehand drawing tool, and the fluorescence values within this region of interest were summed. The region of interest was moved to an adjacent non-stained region, and the fluorescence within this area was summed as a background value. This background was subtracted from the value obtained for the IPCs. For measuring CaLexA or Bursicon levels: Z-stacks were projected using the "sum" method. Each cell of interest was located using the tdTomato channel and highlighted as a region of interest. The fluorescence of the tdTomato and GFP or anti-HA (Bursicon) channels within this region was summed. The region of interest was offset to a nearby unstained region, and the background intensity was summed and subtracted from the value obtained for the stained region. For quantification of tGPH localization, locally flat regions of fat-body tissue were identified in each Z-stack. For each of several pairs of adjoining cells in this region, using the phalloidin and DAPI channels, a Z section was selected at the depth of the nuclei, where the cell membranes are completely perpendicular to the section. A line (with a width of 15 pixels, to reduce noise) was drawn from one nucleus to the other, placing the midpoint of the line at the membrane, and the fluorescence intensity in the GFP channel at each pixel along this line was recorded. The positions along the line were normalized to a length of 1.0. The fluorescence values from the middle 10% of the line (that is, $0.45 \leq x \leq 0.55$) were averaged to create a "center" intensity, and the fluorescence values within a further 10% on each side of this interval (that is, from $0.35 \leq x < 0.45$ and from $0.55 < x \leq 0.65$) were averaged to produce a "surround" value. The center:surround ratio, reflecting the level of membrane enrichment, is reported in the figures.

## Mouse pre-adipocyte differentiation

Subline Mouse 3T3-L1 white pre-adipocytes (ATCC) were obtained from the American Type Culture Collection (ATCC), item #CL-173, and differentiated into mature adipocytes using established methods[88]. In brief, pre-adipocytes were maintained in DMEM (Life Technologies, #521000) supplemented with 10% calf serum (PPA Laboratories, #B15-004) and 62.5 μg/ml penicillin and 100 μg/ml streptomycin (1% Pen/Strep; Life Technologies, #15140-122). Pre-adipocytes were passaged using 0.05% Trypsin-EDTA for 5 minutes at 37 °C. The cultures were grown at 37 °C under 5% CO₂. Growth medium was changed every 48 h. Differentiation into mature adipocytes was induced at two days post-confluency (designated Day 0) by changing the medium to DMEM supplemented with 10% fetal bovine serum (FBS; Life Technologies, #10270), 1 μM dexamethasone (Sigma-Aldrich, #D1756), 0.5 mM 3-isobutyl-1-methylxanthine (Sigma-Aldrich, #I5879), 5 μg/mL recombinant human Insulin (Roche, #11376497001), the insulin sensitizer rosiglitazone at 1 μM (Cayman Chemicals, #71740), and 1% Pen/Strep. On Day 2, the medium was changed to DMEM containing 10% FBS, 5 μg/mL insulin, and 1% Pen/Strep. Because we intended to test for insulin-sensitivity changes in these cells, the sensitizer rosiglitazone was not added on this day. From Day 4, cells were maintained in DMEM containing 10% FBS and 1% Pen/Strep.

## Knockdown of *Lgr4* using synthetic siRNA

Knockdown was performed on Day-6 mature adipocytes[88] using pre-designed siRNA [Merck; Knockdown #1 (KD1), #SASI_MM02_0034484; KD2, #SASI_MMO2_00344385] or negative control Silencer Negative Control No. 1 siRNA (ThermoFisher, #4404021), Lipofectamine RNAi-MAX Transfection Reagent (Invitrogen, #13778075), and Opti-MEM

Reduced Serum Medium (Gibco, #31985-062). No Pen/Strep was used at this stage. *Lgr4* siRNAs were dissolved as 100 μM stocks in nuclease-free water; negative control siRNA was dissolved at 50 μM. All siRNAs were then diluted into working stocks of 10 μM. For each siRNA, a siRNA solution (solution A) and a Lipofectamine solution (solution B) were prepared; for six-well plates (used in phosphoproteomics experiments), siRNA solutions (solutions A) contained 9.99 μL of 10-μM siRNA working stock and 156.51 μL of Opti-Mem, and Lipofectamine solutions (solution B) contained 9.99 μL of Lipofectamine RNAiMAX and 156.51 μL of Opti-MEM. For 24-well plates, used in the TAG assays, qPCR and immunoblotting experiments, siRNA solutions (solutions A) contained 3.75 μL 10-μM siRNA stock and 58.8 μL Opti-MEM, and Lipofectamine solution (solution B) contained 3.75 μL Lipofectamine RNAiMAX and 58.8 μL Opti-MEM. Culturing plates were coated with gelatin during 30 minutes' incubation at room temperature. The gelatin was fully aspirated away, and the plates were further incubated for 10 minutes at room temperature. For each siRNA, solution A and B were combined before addition to the gelatin-coated wells and incubated for 25 minutes at room temperature. Adipocyte cultures were split using 0.25% Trypsin-EDTA for 15-20 minutes at 37 °C, counted, and added to the wells at defined densities ($2.4 \times 10^6$ or $4.75 \times 10^5$ cells per well for 6- or 24-well plates, respectively). Culture volumes were topped up with DMEM + 10% FBS (no Pen/Strep) to 2 mL or 750 μL (6- or 24-well plates). The final siRNA concentration in each well was 50 nM. On Day 8, *i.e.* two days after transfection with siRNA, media were replaced with fresh DMEM with 10% FBS and 1% Pen/Strep. Stimulation of cells and collection of samples for analysis were conducted on Day 10.

## Stimulation of adipocytes

The cells above were stimulated on Day 10 using recombinant human Insulin (Roche, #11376497001) and recombinant mouse R-Spondin 1 (rmR-Spondin 1, R&D Systems, #3474-RS). Peptides were dissolved in UltraPure distilled water, which was also used as a negative control. Peptides were added to the culture medium to final concentrations of 5 μg/mL insulin and 36 nM R-Spondin. The culture medium was gently mixed upon peptide addition, avoiding loosening of attached cells, and cells were incubated for 30 minutes at 37 °C and 5% CO₂ before harvesting.

## Adipocyte collection and immunoblotting

For cells destined for Western blotting, culture medium was removed, and the cells were quickly washed in warmed DMEM (without serum) before addition of ice-cold lysis buffer. Lysis buffer was prepared as ice-cold RIPA buffer (50 mM Tris-HCl pH 7.5, 150 mM NaCl, 0.5% sodium deoxycholate, 1% Nonidet P-40, 0.1% SDS) supplemented with Roche Complete Protease Inhibitor Cocktail with EDTA (Roche, #11836153001), PhosStop phosphatase inhibitor (used at 2x; Roche, #PHOSS-RO), 11 mg/mL β-glycerophosphate (from a 110-mg/mL 10x stock solution), ~100 mM NaF (a saturated solution in RIPA buffer, ~1 M, was made as a 10x stock solution), and Benzonase endonuclease (Sigma-Aldrich, #G9422). The plates containing lysis buffer were moved to ice for at least 10 minutes, before lysed samples were collected into pre-cooled Eppendorf tubes on ice. The samples were spun down to pellet nuclei and membrane debris at maximum speed (20,817 x *g*) at 4 °C for 15 minutes. Supernatant from the middle of the sample (avoiding pelleted debris at the bottom of the tube and the lipid layer at the top) was removed and centrifuged again, and aliquots for analysis were withdrawn from the middle of the supernatant. These clarified samples were mixed 1:1 with 2x Laemmli Sample Buffer (Bio-Rad, #1620737) containing 5% 2-mercaptoethanol and immediately heat-denatured at 95 °C for 5 minutes. Denatured samples were loaded into precast 4–20% gradient Mini-Protean TGX polyacrylamide gels (Bio-Rad, #4561094), and proteins were electrophoretically separated at 150 V for 30–45 minutes using a BioRad Mini-PROTEAN Tetra Vertical

Electrophoresis Cell system. Running buffer contained, per liter, 3.02 g Tris base, 18.8 g glycine, and 10 g SDS. PageRuler Plus Prestained Ladder (ThermoFisher Scientific, #26615) and Chameleon Duo Prestained Protein Ladder (Li-COR, #928-60000) were used as mass standards. Using procedures similar to those detailed under the *Drosophila* western-blot heading, gels were semi-dry-transferred onto PVDF membrane (Millipore), immunoblotted, and visualized. Antibodies used are: rabbit anti-phospho-Akt (S473) (Cell Signaling Technology, #4060 S; 1:1000), rabbit anti-Akt (pan) (Cell Signaling Technology, #4691 S; diluted 1:1000), and mouse anti-alpha-Tubulin (Sigma #T9026, diluted 1:4000). Stripping of membranes of phospho-Akt antibody was done using Alfa Aesar Stripping Buffer (#J60925) for 30 minutes at 37 °C, after which the membrane were washed 3×15 minutes using PBS + 0.1% Tween20. The membranes are then re-blocked using Intercept blocking buffer (Li-COR, #927-60001) and re-stained for Akt and tubulin.

For adipocyte collection for TAG measurements, cells were treated with 0.25% trypsin to detach them, after which DMEM + 10% FBS was used to de-activate the trypsin. TAG samples were pipetted up and down to ensure loosening of the cells and then collected into Eppendorf tubes on ice. Samples were centrifuged at 153 x *g* for 5 minutes to pellet cells. Supernatant was removed and cell pellet was frozen at -80 °C until further processing. Samples were thawed on ice, and the cells were lysed using manual grinding with a pestle in PBS + 0.05% Tween-20 on ice, followed by the procedure described in the methods section for metabolite analysis.

For qPCR, adipocytes were washed twice with PBS before being lysed with RNA-isolation kit lysis buffer (buffer RA1 + 1% 2-mercaptoethanol; Macherey-Nagel, #740955.250) directly in the plate. Samples were collected and frozen at -80 °C until processing. For RNA extraction, samples were processed as described by the manufacturer, with thorough vortexing prior to extraction. Samples were processed as outlined under the Quantitative PCR method heading.

For sample collection upstream of phosphoproteomics analysis, plated cells were washed with DMEM and gently detached using a cell scraper and collected into 15 mL falcon tubes. Samples were centrifuged at 153 x *g* for 5 minutes at room temperature to pellet cells, after which supernatant was removed before samples were flash-frozen on dry ice and stored at −80 °C.

## Phosphoproteomics analysis of mouse adipocytes

Proteins from mouse adipocyte cells were extracted triplicates per condition as biological replicates in 300 μL 3% sodium deoxycholate in 100 mM HEPES, pH 8.5, by using probe sonication for 3×20 seconds at 60% amplitude. After sonication, the samples were centrifuged at 20,000 x *g* for 20 minutes, and the supernatant was recovered in a new Eppendorf tube. The protein concentration was measured using a spectrophotometer (Nanodrop N60, Implen). From each sample a total of 100 μg of protein was reduced with 10 mM DTT for 20 minutes and then alkylated for 20 minutes using 20 mM iodoacetamide. Unreacted iodoacetamide was quenched with 5 mM DTT for 15 minutes. Proteins were cleaved overnight by 5% trypsin at 37 °C. After this incubation, an additional 1% trypsin was added, and digestion continued for one hour. The peptides were labeled with tandem mass tags (TMTpro 18-plex), following the manufacturer's protocols, and the labeled samples were then pooled. Sodium deoxycholate was removed through acidification and centrifugation at 20,000 x *g* for 20 minutes. The supernatant was then transferred to a low-binding Eppendorf tube and concentrated until only 150 μL remained. Phosphopeptides were enriched using titanium dioxide separation, followed by high-pH reversed-phase (RP) fractionation[89]. The TiO2 flowthrough containing nonphosphorylated proteins was desalted and subjected to high-pH RP fractionation[1].

The phosphopeptide and unmodified-peptide fractions from these separations were analyzed via tandem mass spectrometry, employing an EASY nanoLC system coupled to either an Exploris 480

Orbitrap (phosphopeptides) or an Orbitrap Eclipse Tribrid (non-modified peptides) (ThermoFisher Scientific). Lyophilized peptides from the high-pH RP fractionation (20 concatenated fractions for each) were reconstituted in 3−5 μL of 0.1% formic acid (FA) and applied to a 20-cm analytical column (100-μm inner diameter) packed with ReproSil-Pur C18 AQ 1.9-μm RP material. Peptides were eluted using an organic solvent gradient, starting from 100% phase A (0.1% FA) to 25% phase B (95% ACN, 0.1% FA) over 100 minutes, then increasing from 25% B to 40% B over 20 minutes, before the column was washed with 95% B. The flow rate during elution was maintained at 300 nL/min. On both instruments, the automatic gain control was set to a target value of $1.5 \times 10^6$ ions for MS scans, with a maximum fill time of 50 ms. Each MS scan was performed at a high resolution (120,000 FWHM) at *m/z* 200 in the Orbitrap, covering a mass range of 350−1500/1600 Da. The systems were configured to select the maximum number of precursor ions within the 3-second window of the MS analyses.

For the phosphopeptide analysis on the Exploris 480, peptide ions were selected for higher-energy collision-induced dissociation (HCD) fragmentation (collision energy: 33%), with fragment ions detected in the Orbitrap at high resolution (45,000 FWHM) aiming for a target value of $1.5 \times 10^5$ ions and a maximum injection time of 150 ms, using an isolation window of 0.7 Da and a dynamic exclusion period of 20 seconds.

For LC-MSMS analysis of the non-modified peptides, the Orbitrap Eclipse Tribrid was operated in real-time (RT) searching SPS-MS3 mode[90]. Here the peptides were selected and fragmented using CID in the linear ion trap with isolation window of 0.7 *m/z*, normalized collision energy of 35, and turbo-detection in the linear ion trap using auto settings. Each CID spectrum was subjected to the built-in database search algorithm using the mouse EMBL 2024 FASTA database with fixed TMTpro (N-terminal and lysine) and fixed carbamidomethyl (cysteines). If the CID spectrum resulted in an identification, the peptide ion was reselected and fragmented with CID, and then 10 fragment ions covering the identified sequence were reselected and fragmented using HCD with normalized collision energy of 55 to permit accurate quantitation using the TMT reporter ions[90].

All raw data were analyzed using Xcalibur v3.0 (ThermoFisher Scientific). Phosphopeptide and protein identification and quantification were performed on all LC-MS/MS raw data files using Proteome Discoverer (PD) version 2.5.0.400 (ThermoFisher Scientific). The 20 raw data files from the phosphopeptide analysis were searched in SEQUEST HT using the EMBL mouse-protein database (21,957 entries). The search parameters included: enzyme, trypsin (full); maximum missed cleavages, 2; and fixed modifications of TMTpro (N-terminal), TMTpro (K), and Carbamidomethyl (C) and variable modifications Phosphorylation (S/T/Y). For the non-modified peptide LC-MSMS fractions, the $MS^2$ spectra were used for searching in SEQUEST HT in the same EMBL FASTA protein database. The search parameters included: enzyme, trypsin (full); maximum missed cleavages, 2; and fixed modifications of TMTpro (N-terminal), TMTpro (K), and Carbamidomethyl (C). The MS3 data for each identified peptide was used to extract the TMT quantitative values.

Percolator[91] was used to filter all data for the identified peptides with 1% FDR and a protein FDR below 1%. Quantification of TMTpro reporter ion signals was based on signal-to-noise (S/N) ratios, normalized against the total peptide S/N within Proteome Discoverer (PD). A built-in ANOVA test in PD was utilized to determine *p*-values for all phospho-peptides and proteins identified through these database searches.

## Data filtering and bioinformatics analysis of phosphoproteomics

Phosphoproteomics data for pathway analysis was filtered for differences >30% in magnitude between control and insulin-stimulated samples within each genotype. Pathway analysis was carried out on the phosphoproteomics data using the Panther Classification system

(Protein Analysis Through Evolutionary Relationships), with the statistical overrepresentation test using Reactome database (version 85)[92]. In addition to the statistical analysis already conducted during data filtering, a Fisher's T-test was applied, and only pathways with a $p < 0.05$ for enrichment were considered. Pathway analysis dot-plot illustration was in part made using online platform SRplot[93]. Venn diagram data was based on data with >30% magnitude change in response to insulin stimulation. Repeated proteins (due to multiple significant phosphorylation site changes) were removed. The online platform DeepVenn[94] was used to identify overlapping or unique proteins in each genotype.

## Statistics and reproducibility

Statistical calculations were performed using the Prism software package (GraphPad). All data sets were evaluated for normality before other tests were performed. The statistical tests for significance that were employed in each figure are given in the respective legend. All data were obtained from biologically independent samples. In Supplementary Data 2, the built-in ANOVA test in Proteome Discoverer (PD) version 2.5.0.400 (ThermoFisher Scientific) was used to generate $p$-values for all phosphopeptides and proteins identified in the database searches. In Supplementary Data 3, a two-sided Fisher's Exact Test was applied, and only pathways with a $p$-value < 0.05 for enrichment were considered significant. No statistical method was used to predetermine the sample size. Sample sizes were chosen based on standard practices in *Drosophila* research to ensure sufficient data for statistical analysis. No data were excluded from the analyses. The experiments were not randomized. Allocation into experimental groups was based on standard practices used in *Drosophila* research. The investigators were not blinded to allocation during experiments and outcome assessment.

## Reporting summary

Further information on research design is available in the Nature Portfolio Reporting Summary linked to this article.

## Data availability

All data generated in this study are available within this manuscript, its figures, and its supplementary files. Raw protein mass spectrometry data generated in this study have been deposited in a ProteomeXchange partner repository, under accession code PXD053061 and are accessible via http://www.ebi.ac.uk/pride/archive/projects/PXD053061. The raw data for the larval-CNS RNAseq[95] analysis presented in Supplementary Fig. 3B (top) are available from NIH Gene Expression Omnibus at: https://www.ncbi.nlm.nih.gov/geo/query/acc.cgi?acc=GSM3964166. Adult-IPC RNAseq data[38] presented in Supplementary Fig. 3B (bottom) were analyzed using the SCope interface and the "dilp2IPC" data set at: https://scope.aertslab.org/#/FlyCellAtlas/FlyCellAtlas%2Ffca_biohub_dilp2ipc_ss2.loom/gene and https://www.ebi.ac.uk/biostudies/arrayexpress/studies/E-MTAB-10628?query=E-MTAB-10628. Images are available from the primary contact without restrictions. Source data are provided with this paper.

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

## Acknowledgements

*Rickets-GAL4* (*rickets^Pan^-GAL4*)[71] was a kind gift of Ben White (NIH). Rabbit anti-Ilp2[83] was a gift of Ernst Hafen and Michael Pankratz (University of Bonn). Mouse anti-Ilp3[84] was a gift of Jan Veenstra (University of Bordeaux). Rabbit anti-Foxo was a gift of Pierre Léopold (Institut Curie). Plasmids pCFD3 and pCFD6 were obtained from AddGene. Fly stocks were also obtained from University of Indiana Bloomington *Drosophila* Stock Center and Vienna *Drosophila* Resource Center. This work was supported by the Danish Council for Independent Research Natural Sciences grant 8021-00055B to KR and Novo Nordisk Foundation grant NNF21OC0070402 to KR. AFJ was supported by a Novo STAR co-financed industrial PhD fellowship from Novo Nordisk Foundation and Innovation Foundation. TK and KVH were supported by funding from the Danish Council for Independent Research Natural Sciences (grant 9064-00009B) to KVH. The Zeiss LSM 900 confocal microscope and the PerkinElmer EnSight plate reader were purchased with equipment grants from the Carlsberg Foundation (CF19-0353 and CF17-0615) to KR et al.

## Author contributions

A.F.J., M.J.T., J.L.H., and K.R. conceived and designed the study. O.K., A.F.J., M.L., M.J.T., T.K., S.N., J.B.H., M.R.L., K.V.H., J.L.H., and K.R. designed, performed, and analyzed experiments. J.H. and D.M. contributed to the design of the list of genes. G.M. conducted an in-depth literature search on Lgr4's association with metabolic obesity and diabetes. M.J.T. and K.R. wrote the manuscript, and O.K., A.F.J., T.K., M.L., S.N., J.H., J.B.H., M.R.L., K.V.H., and J.L.H. reviewed and edited the manuscript. M.J.T., J.L.H., and K.V.H. contributed equally.

## Competing interests

The authors declare no competing interests. G.M., J.H., and D.M.M. are Novo Nordisk employees and shareholders. A.F.J., J.L.H. are Novo Nordisk shareholders.
