## [Peer Review File · Nature Communications]

LGR signaling mediates muscle-adipose tissue crosstalk and protects against diet-induced insulin resistanceREVIEWER COMMENTS

Reviewer #1 (Remarks to the Author):

Current report screened the secretome and receptome in *Drosophila* to identify the underlying interorgan hormonal crosstalk affecting diet-induced insulin resistance and obesity. Please conduct the concerns below.

1. The title seems better to add "in *Drosophila*".
2. It has been designed in a good way in experiments.
3. In the abstract, the conserved BMP and LGR signaling pathways need the full name at first time.
4. In the introduction section, the fruit fly *Drosophila* has become a valuable model for understanding the contribution of nutrition to metabolic disorders needs the reference(s) to support.
5. Bursicon signaling is important in metabolic regulation in fruit fly *Drosophila*. Is it same in humans or animals?
6. An interorgan communication between the musculature, the nervous system, and adipose tissue influences sugar tolerance and insulin resistance needs to prove in animals also, or cite the reference(s) to support.
7. Linkage of the obtained findings with human physiology was ignored. Why?

Reviewer #2 (Remarks to the Author):

This interesting study reveals a role of the Bursicon signaling pathway on tolerance to high sugar diets in *Drosophila*. Following an original genetic screen, in which the authors identified the glycoprotein hormone Bursicon alpha subunit, they show that neuronal Bursicon promotes high sugar tolerance by signaling to its known and expected receptor, *Lgr2/Rickets*, an ortholog of mammalian *Lgr4/5/6*, in two locations: to insulin-producing cells in the central nervous system—thereby promoting insulin-like peptide transcription and release—, and to fat body cells, promoting insulin sensitivity in such cells. These activities collectively improve sugar tolerance of the animal by reducing hemolymph sugar levels and storing excess energy as triacylglycerides (TAGs) in the fat body. The authors then show that a muscle BMP ligand, *Gbb*, which had been previously found to be required for peptide expression (including Bursicon) in Bursicon-neurons, responds to high-sugar diet and acts on Bursicon-positive neurons promoting their high-sugar tolerance function. The authors add two further datasets from which they draw the following conclusions: 1) Bursicon-dependent *Rk* activation in the fat body leads to reduced levels of the lipocalin Neural Lazarillo (*NLaz*), and that this is required for proper glycemic control promoted by the Bursicon-*Rk* pathway. 2) *Relish/NFκB* activity is controlled by the insulin pathway in the fat body.

Resolving neuroendocrine circuits is a challenge and the authors provide substantial work towards this end. However, while the initial part is potentially interesting—provided the concerns below are addressed—and it is backed up by such substantial amount of work and experiments, the *NLaz* and *Relish* parts are underdeveloped or are somewhat out of focus, respectively. The *NLaz* finding is directly related to the study narrative, so it should be further substantiated with additional experiments. The *Relish/NFκB* part, in contrast, appears to be floating around and brings little to the discussion of the Burs-pathway role in sugar tolerance. As it is somewhat unrelated to the main narrative, it could be removed from the ms and explored more deeply in another manuscript, or placed in supplementary data.

The manuscript is overall very well written and if the major and minor concerns below can be addressed, the work should be a valuable contribution to different fields, including development, physiology, neurobiology, and metabolism.

Thank you for the opportunity to review this manuscript.

Major concerns

Results Fig 2 and Extended data, Figure 2 a, b, and c. What are the exact genotypes of the controls? Without them it is difficult to evaluate the experiment and results. A file with the full genotypes of the animals used in each figure panel should be provided. Regardless, Figure 2b, for instance (this applies to many other experiments), is lacking many experimental controls. If one considers that the control is Burs>/+ (Burs> crossed to w[1118]?) alone, ideally, the controls UAS-Burs-RNAi alone, Burs>Cas9 alone, UAS-Cas9 UAS-gRNA alone would be assayed for this experiment and for each other similar experiment (eg, Extended Data Fig 2 b,c), or at least in these initial fundamental experiments to establish the responses of each genetic component to the HSD treatment.

Extended data, Figure 2a-c: a rationale justifying the need for these analyses is not presented. The x-axis variable "Delta 50%-pupariation time (%)" is confusing or not very clear. I suggest using simple ratios or % change (of the 50% pupariation time). Regardless, please clarify how the calculations were made in the figure legend and/or methods - the way it is written in the manuscript is neither complete nor precise). Please apply the rationale to the data on Fig 4b and 4f: What does it show? Same to other similar figures of the ms, eg, Fig 5a, 6a, 6d, 7j-k. How do these results affect the conclusions?

Results relative to Figure 3:

What is the effect of Burs-KO and RNAi in Burs cells on ilp2-5 transcript levels in animals raised under a normal (non-HSD) diet? This is relevant as the effect of Burs CRISPR-KO or RNAi on ilp transcript levels is attributed to HSD conditions (as in this sentence "Taken together, these findings suggest that, ***in response to consumption of dietary sugar***, Bursicon signaling acts on the IPCs to promote the production of Ilp2 and Ilp3 to increase insulin signaling and enhance tissue uptake of circulating glucose." , but no data is shown in normal diet, so this effect could be diet independent.

Results, Page 11: "We observed a trans-Tango-dependent tdTomato signal in the IPCs when the presynaptic ligand was expressed in Burs+ neurons (Fig. 3c), "
The technical description of the experimental conditions of the trans-Tango is insufficient and the results depicted are lacking controls (especially the conditions without the GAL4 in normal diet and HSD diet; and images of the tdtomato and myrGFP channels for the IPCs should be included in all conditions). In addition, it would be helpful to show lower magnification projection stacks of whole CNS preparations in supplementary data.
Full conditions of the experiments should be reported somewhere (such as in a dedicated methods section) or for all figures in supplementary materials. What are we looking at? anti-Ilp2 detected with which secondary antibody? Endogenous tdtomato fluorescence of the trans-tango and/or antibody stainings?

Results, Page 11. Relative to Figure 4: The delays in pupariation in Figures 1 and 2, for instance, are linked to compromised sugar tolerance, which is then linked to reduced Ilp levels, lowered TAG levels, and increased circulating sugar. rk-KO/RNAi and PKA-R1[BDK] manipulations in ilp2-GAL4 cells lead to sensitivity to HSD (HSD-dependent delay) and reduced Ilp levels, but have no effect on TAG levels (this is later justified by attributing fat body effects to a second, Burs-dependent role of Rk in the fat body). However, if there is no effect on TAG levels in ilp2>rk-KO/RNAi animals, but there is compromised high-sugar tolerance, how does this happen? What are the levels of circulating sugars in these animals? Why wasn't the ND condition also included in Fig 4g? The TAG levels in Fig 4g (HSD (10h) diet) look more similar to those of ND levels presented in other figures (eg Fig 2d-e and 5b). Is this variation normal? Does the ilp2> background respond differently to HSD? The authors need to address these questions experimentally and discuss how these results fit or not their model.

Results Page 12, first paragraph: "Consistent with this, loss of Rk activity in the IPCs led to reduced levels of circulating levels of Ilp2 peptide in the hemolymph (Fig. 4e)." These results appear to be in the Normal Diet (ND) condition. This is a critical finding, suggesting that the effects of reduced Rk signaling on Ilp levels are independent of the HSD treatment. This further highlights the requirement of ND controls for the qPCR experiments for ilps on Fig 2 and 3. Please adjust the

text to reflect this finding and limitation.

Results Page 13, Figure 5: Please indicate whether data on panels 5e are reused on Panel 5j. Specifically, at least condition CG>rk-KO in HSD looks exactly duplicated in these panels. The controls (CG>?) are nevertheless different. Why are the controls different and the CG>rk-KO condition reused? Data duplication must be explicitly justified and acknowledged. Please revise the whole manuscript for such instances and clearly point them out in the manuscript methods and/or figure legends at least. For instance, Figure 2 panels 2d and 2e have duplicated controls in ND and HSD and this is not acknowledged.

Please acknowledge/discuss the possible limitations for the model and conclusions of using CG-GAL4 as a driver for fat body to study systemic effects, considering that known fact that CG-GAL4 is also expressed in hemocytes. Can this be ignored? If yes, why?

Results Page 15, first paragraph: The evidence leading to the conclusion on the downstream placement of NLaz relative to Burs signalling is relatively weak. Only one readout (circulating sugar) of the rk-KO rescue is shown (what about the HSD-dependent delay, insulin-like peptide transcript levels (there should be no change?), and TAG levels?). The ND diet conditions are not shown, yet the effects under these conditions are required to interpret the results at HSD. Does NLaz RNAi in the fat body rescue the neuronal Burs-KO/RNAi effects and the Gbb-KO/RNAi effects? (see comment below on weaknesses regarding model testing).

Results, Page 17-18. Relative to Fig 7b. This is a particularly weak experiment. The 29% increase albeit statistically significant is a minute effect and the range of the CaLexA signal in the assayed cells is completely overlapping with the control condition. Hence, conclusions on this single readout alone should be taken with great caution, but are not. This part could be stronger if the authors provided an independent measure of neuronal activity, e.g., GCaMP-based live imaging. In addition, parallel quantification of tdTomato fluorescence in the same cells assayed for CaLexA expression could be helpful to control for artefacts such as minute dietary effects on Burs-GAL4 expression levels (if there is no modulation, one would expect to see no difference between ND x HSD). This is critical because a small HSD-dependent Burs-GAL4-expression increase could lead to the observed CaLexA effects. The fact that Burs protein levels in Burs neurons are Gbb-signaling dependent (Veverytza and Allan, Development 2011) and are slightly increased (Fig 7c-d, albeit not statistically significantly) in these cells indicate that this is indeed a possibility. Furthermore, increased pMad immunoreactivity (Veverytza and Allan, Development 2011) could be a more direct alternative reading of increased BMP signalling in Burs neurons via Gbb.

The authors have the challenge of teasing out developmental and nutritional roles of BMP signalling. This could be achieved with GAL80 and temperature shift experiments at least within the context of the 10-h HSD assays. Alternatively, the authors should discuss the limitations of the study and alternative interpretations.

In addition, the wording of the Gbb finding is discussed within the context of a systemic signal (see also summary figure 7L, where a black arrow goes from the muscle to the Burs-positive neurons), whereas the evidence is that Gbb functions as a non-systemic, localised retrograde signal from muscles to the efferent Burs-positive neurons, which project towards the body wall (Veverytza and Allan, Development 2011).

The proposed model implies that Gbb-IR or KO in the fat body will lead to increased circulating sugar levels. Ideally, this would be shown in Fig 7.

Finally, the NLaz and Gbb findings should be substantiated with rescue experiments that test the hypothetical models proposed. The experiments presented for NLaz and Gbb are very limited and the lack of such experiments are a weakness of this paper. For instance, an attempt to rescue the effects of Gbb-IR/KO in fat body could be made in many ways, such as by activation of Burs+Pburs neurons (using TRPA1 or optogenetics) or activation of Rickets-signaling in IPCs (maybe using constitutively active PKA? PKA-CQR?).

Minor concerns

Fig 1. Statistics: "ANOVA with Šidak's correction"

Please verify and clarify. Sidak's correction is applied to a post-hoc multiple comparison test, not to ANOVA per se. Which tests were performed here and corrected with Sidak's correction after the ANOVA? T-tests?

"Statistics" in other figures: "Dunnet's correction" is mentioned many times. Please verify and clarify. Do the authors mean the Dunnet's multiple comparison test? If yes, it is not a correction applied to a test, but rather a post-hoc test itself.

Fig 2a: please confirm that the figure legend indeed corresponds to the plotted lines. (specifically the control and heterozygote in HSD). If it is correct, the effect of heterozygote animals is quite large and to the opposite direction as to the homozygote animals. This should be acknowledged in the text.

Fig 2 (a-b), Extended data Figs 1, 2 (d), Fig 4 b,f - legends say pupation time, but figures say pupariation time. Please clarify.

Results Page 7, 1st paragraph dLgr2/Rickets (orthologous with mammalian Lgr4/-5). Please include -6 (mammalian Lgr6). dLgr2 is equally distantly related to mammalian Lgr4, Lgr5, and Lgr6.

Results, Page 8, 3rd paragraph, last sentence "This further supports to the importance..." maybe revise the wording.

Extended data, Figure 2, Statistics : "Error bars represent mean and SEM." please verify and confirm that the bars in these panels indeed indicate the standard error of the mean. If not, please correct. The error bars seem inconsistent with SEMs considering the distribution of the data points depicted. They look more like SD—which is good, as it is preferable than SEM—or some other measure. Please also verify and correct if necessary Fig 1C.

Results, Fig 2: What was the Cas9 construct used for the somatic KO eg on Fig 2? I did not find it in the methods section. Full genotypes should be reported somewhere.

Results, Fig 2 panels c-e:
please show TAG levels as mg/mgprotein as in panels d and e.

Results, Page 9: "an effect that was abolished by Burs loss (Fig. 2c)." should read "...by RNAi or CRISPR-targeting of Burs in Burs> cells" to avoid confusion with the Burs mutation analyses.

The fact that neuronal Burs knockdown (partial attenuation of TAG increase) only partially mimics Burs knockdown in Burs> cells (complete abrogation of TAG increase) should be discussed.

Results, page 9: "Consistent with reduced insulin signaling, loss of Burs led to elevated glycemic levels after chronic or short-term (10-hour) exposure to a high-sugar diet (Fig. 2h,i)." Please mention that the effect of the Burs CRISPR-targeting in the short-term (10h) treatment condition is not statistically significant, so this statement is only statistically precise for the RNAi condition.

Results Page 12, first paragraph: "This is further supported by single-nucleus transcriptomics³⁷." Please show the specific data/analyses of the data that supports the claim that rk is expressed/enriched in IPCs in a supplementary figure.

Results Page 12: "...IPC led to HSD-dependent delay..." maybe missing an "an" before HSD?

Results Page 12, 2nd paragraph: "To identify the target tissues mediating any effect of humoral Bursicon signaling on fat storage, we assessed the tissue expression of rk and found strong expression in the fat body..." This statement lacks support of data or reference. Please show the data.

Results Page 13, Figure 5: again several panels missing controls. eg, panel 5i missing RNAi alone. Panel 5j missing minimal controls such as UAS transgenes alone and CG>NLaz-IR alone. These controls are critical to interpret the results.

Results Page 14, first paragraph:

"...found that reducing Nlaz expression in the fat alone completely.." maybe missing "body" before alone?

Results Page 15, first paragraph:

"We then tested the ability of NLaz deficiency to..." Please use NLaz RNAi instead of deficiency.

Results, Page 15, 3rd paragraph - "We observed that, in animals fed a high-sugar diet, knockdown of relish in the fat body led to reduced nuclear localization of FOXO (Fig. 6c) and an associated reduction in 4EBP expression (Fig. 6b), indicating increased insulin signaling in the fat tissue ". Again, what occurs in ND? Is the effect of relish on FOXO localisation dependent on HSD?

Results, Page 17, 1st paragraph: "Strong calcium-induced GFP signal was observed solely in Burs+ neurons that produce the Burs:Pburs heterodimer, indicating that these are the active Burs+ neurons (Fig. 7a). " It is not clear if this sentence and Fig 7a refer to a CNS preparation under ND or HSD. Please clarify. Also, the statement "that these are the active Burs+ neurons" is too ample and vague. It can read true or false under different conditions, so please specify which specific condition it applies to."

Results, Page 17: 2nd paragraph: Typo "...pathway were identified as a hits in our screen..."

Results Page 18, Fig 7L: the label "Rickets (Lgr4)" is a bit confusing. Rickets is Lgr2 in Drosophila and, in Drosophila, Lgr4 is another receptor (orthologous to mammalian Lgr7-8). Hence, please indicate the fly equivalent (Lgr2 (or dLgr2)) or include "mammalian Lgr4/5/6".

Discussion, Page 19, 2nd paragraph: "We discovered that Rickets,..." Please include its alternative name "We discovered that Rickets/Leucine-rich repeat- containing G-protein coupled receptor 2 (Lgr2 (or dLgr2, if necessary/preferable)),..." to avoid confusion (see above). Please refer to the orthology to the other mammalian Lgrs. Please mention known ligands to the members of this mammalian Lgr4/5/6 family (R-spondins (Roof plate specific spondins (RSPOs))/Norrin (Norrie Disease Protein, specific for Lgr4)/RANKL (Receptor Activator for Nuclear Factor κ B Ligand)), at least one of which is a cysteine knot protein, Norrin).

Methods, Metabolite assays, Line 13

revise question mark: "collecting the exudate with a pipette?"

Methods, transgene construction:

Is the FLAG::Burs::HA viable or have any Burs-like phenotypes?

Reviewer #3 (Remarks to the Author):

Even though insulin insufficient and insulin resistance have been well established to promote high-caloric diet-induced hyperglycemia, the endocrinal signals mediating different organs to amount systemic responses still remain largely unclear. Integrating RNAi screening against secreted proteins and receptors, Kubrak et al uncovered neuronal Burs governs both insulin synthesis and peripheral insulin sensitivity through its receptor RK in the context of HSD. Further, they found that neuronal Burs production is regulated by muscle-produced Gbb. The findings are pretty interesting and should provide significant impacts in the field of diet-induced hyperglycemia and disease development such as obesity and type 2 diabetes. However, some of the experiments were not designed very carefully. The authors also need to address a few important comments prior to publication in Nature Communications.

Major comments:

1. Previous studies have shown gut-derived Burs dramatically affects systemic energy homeostasis (PMID: 30344016). I recommend authors to check whether gut-derived Burs is involved in the condition in this study using pros-Gal4 and other gut specific Gal4 lines, even though they proposed the predominant roles of neuronal Burs. They also need to confirm the Burs source in brain using specific nSyb-Gal4 that does not target gut cells (PMID: 32917721). No matter Burs mutation or Burs>Burs-RNAi diminishes Burs production in the whole body. Figuring out the real sources of functional Burs would be an important question to be addressed.

2. Only loss-of-function assays were performed in this study. Will gain-of-function of Burs signaling rescues diet-induced insulin resistance and hyperglycemia? How about overexpression of RK or TrpA1-induced activation of Burs cells (neurons and gut cells)?

3. The authors need to repeat most of the development assays on HSD to make results of control flies consistent. For example, pupation occurs between day 6-8 (Fig. 1a), day 8-10 (Fig. 1b, 4f, 6b), day 8-9 (Fig. 4b, 5a, 7j), day 7-10 (Fig. 6a), day 9-12 (Fig. 7k). The differences could be caused the problem of control flies but not manipulations.

Minor issues:

4. For most ILP production assays, the results of decreased ILP2 level in both intracellular accumulation in IPCs and hemolymph do not support the regulation of SECRETION. Please modified the statement as "production/synthesis".

5. The knockdown assays of Burs and signaling worsened the HSD-induced hyperglycemia. The authors should modify some statements like "cause HSD-induced hyperglycemia".

6. Fig 6 should be moved into supplementary data, as the results were irrelevant to the main conclusion and complicated the molecular mechanisms of Burs signaling.

7. Remove the "FoxO" in the regulation of muscle Gbb production in working model in Fig. 7l. the authors did not provide evidence. Moreover, it might lead to a conflict mechanism proposal like "muscle insulin resistance -> Gbb -> Burs -> fat body insulin resistance". Insulin resistance in the muscle is earlier than the fat body?

Response to reviewers' comments

Reviewer #1 (Remarks to the Author):

Reviewer: Extended Current report screened the secretome and receptome in *Drosophila* to identify the underlying interorgan hormonal crosstalk affecting diet-induced insulin resistance and obesity. Please conduct the concerns below.

Response: We thank the reviewer for taking the time to provide feedback on our work. We have significantly revised our manuscript to address the reviewer's concerns and increase the translational value of our work. Our revised work includes 44 new figure panels plus 5 new Supplementary Tables, including new data that translate findings to mammalian adipocytes and emphasize the implications of these findings for human health and disease. We have included a new summarizing table 4 that reviews the available genetic evidence linking LGR4, a mammalian ortholog of the *Drosophila* Bursicon receptor Rickets (also called dLgr2), and obesity and diabetes. The relevant studies are physiological investigations of whole-animal gain or loss of *Lgr4* function, and as such they are informative but do not link the observed phenotypes to LGR4 function in any particular tissue and do not show direct involvement of *Lgr4* in insulin signaling. We have now performed substantial work in cultured mouse adipocytes that directly show that *LGR4* is required for normal insulin signaling in these cells: when *Lgr4* expression is attenuated using either of two independent siRNA treatments, the adipocytes no longer respond to an insulin stimulus, shown both through immunoblotting against the insulin-signaling mediator phospho-Akt (phospho-Protein Kinase B) and through phosphoproteomics analysis of all cellular proteins. *Lgr4*-deficient mouse adipocytes also accumulate less lipid. These effects, described in the entirely new Figure 6, closely mirror the insulin-resistance and lipid phenotypes observed in *Drosophila* adipocyte-like cells (cells of the fat body) lacking Rickets, illustrated in Figure 5. The effect of *Lgr4* on the response to insulin stimulation was quite unexpected: almost all of this response depends on LGR4. These mouse white adipocytes became almost completely insulin-resistant following the loss of *Lgr4* expression (through either of our two siRNAs). Insulin resistance is the main cause of type 2 diabetes. Our findings indicate that LGR4 signaling is fundamental for adipocytes to be sensitive to insulin, possibly affecting the initial stages of the insulin signaling cascade, either at the level of the insulin receptor itself or immediately downstream. This could potentially be a very important new finding. Below, we have addressed the concerns raised in individual points.

Reviewer point 1. The title seems better to add “in *Drosophila*”.

Response: This is a good suggestion for our previous manuscript. However, during the revision process we have incorporated a great deal of work in mammalian cell culture (the new Figure 6, which clearly shows that LGR4 is required for normal insulin signaling in mouse adipocytes) that directly link our *Drosophila* findings to the orthologous mammalian systems. Therefore, we propose to change the title entirely to something that reflects this broader applicability – “Relaxin/LGR signaling mediates muscle-adipose tissue crosstalk and protects against diet-induced insulin resistance”.

Reviewer point 2. It has been designed in a good way in experiments.

Response: We sincerely appreciate the reviewer's positive feedback on the design of our experiments. We have made every effort to ensure that our experimental design is robust and methodologically sound, and it is gratifying to receive recognition for this aspect of our work. We have added a substantial number of new experiments, which we hope will be equally well received. Thank you for your encouraging comments!

Reviewer point 3. In the abstract, the conserved BMP and LGR signaling pathways need the full name at first time.

Response: We thank the reviewer for pointing this out. Bone Morphogenetic Protein (BMP) and Leucine-rich repeat-containing G-protein coupled Receptor (LGR) signaling are now fully spelled out in the abstract.

Reviewer point 4. In the introduction section, the fruit fly *Drosophila* has become a valuable model for understanding the contribution of nutrition to metabolic disorders needs the reference(s) to support.

Response: We apologize for not having supported this statement with citations. To rectify this oversight, we have now incorporated the following references that highlight the value of *Drosophila* in metabolic research:

(1) “*Drosophila* as a model to study metabolic disorders,” J. Hoffmann, R. Romey, C. Fink, and T. Roeder, *Advances in Biochemical Engineering/Biotechnology* 2013. This reference emphasizes the suitability of *Drosophila* for understanding the genetic and environmental factors influencing metabolic disorders.

(2) “*Drosophila* as a model to study obesity and metabolic disease,” L. P. Musselman and R. P. Kühnlein, *Journal of Experimental Biology* 2018. This article explores how *Drosophila* has been used to study the complexities of lipid metabolism, genetic predisposition to obesity, and the interplay between diet and metabolic health. The article discusses the conservation of metabolic pathways and the relevance of fly models to human conditions.

We would like to note that with our addition of extensive supporting results from mouse experiments, our revised work now perfectly illustrates the value of *Drosophila* studies in guiding mammalian investigations.

Reviewer point 5. Bursicon signaling is important in metabolic regulation in fruit fly *Drosophila*. Is it same in humans or animals?

Response: We appreciate your insightful question regarding the relevance of Bursicon signaling, observed in *Drosophila*, to metabolic regulation in humans and other animals. Our research focuses on elucidating the role of the *Drosophila* LGR4 homolog Rickets (Rk) in metabolic processes, offering foundational insights into the potential functions of LGR4 in human metabolism. The conserved nature of LGR4 signaling pathways across species suggests a potentially significant role in mammalian metabolic regulation, particularly concerning sugar tolerance, adipose-tissue sensitivity to insulin, and insulin secretion. In our revised manuscript (as described in detail in response to point 7 below), we have included data summarizing the substantial genetic evidence linking mammalian LGR4 to obesity and diabetes, and indeed we have built upon this foundation by clearly showing (in the new Figure 6) that LGR4 is indeed required for normal adipose-cell responses to insulin. This information highlights the implications of our work for understanding the metabolic role of human LGR4, providing interesting perspectives and insights into the human/mammalian phenotypes associated with LGR4 variants.

Reviewer point 6. An interorgan communication between the musculature, the nervous system, and adipose tissue influences sugar tolerance and insulin resistance needs to prove in animals also, or cite the reference(s) to support.

Response: Addressing the need for evidence of similar interorgan communication in animals, our manuscript draws parallels between *Drosophila* and mammalian systems, particularly in the context of LGR4 signaling. Notably, mammalian LGR4 is expressed in key metabolic organs, including the pancreas, liver, and adipose tissue. Studies in humans and mice have linked LGR4 mutations and expression levels to obesity and type-2 diabetes, suggesting a critical role in metabolic homeostasis and disease (reviewed in new Supplementary Table 4)¹⁻³, and our new mouse-adipocyte results clearly confirm the essential role of LGR4 in mammalian insulin sensitivity (new Fig. 6). These findings underscore the translational potential of our *Drosophila*-based research, providing a framework for understanding the complex interplay between muscle, the nervous system, and adipose tissue in regulating metabolic health across species. The identification of the salient mammalian ligand(s) –

several known possibilities are reported, including R-Spondin (RSPO) 1, 2, 3, and 4 and the Bursicon-like cystine-knot protein Norrin⁴ – and the characterization of relevant source tissues, dietary regulation, and effects on other *Lgr4*-expressing target cells will be an exciting area of future investigation.

Reviewer point 7. Linkage of the obtained findings with human physiology was ignored. Why?

Response: In addressing the reviewer's comments on the translational relevance of our study, it is pertinent to emphasize that our research was conducted as a collaboration between our university and Novo Nordisk, a leading authority in the treatment of diabetes and metabolic disorders. This partnership was strategically formed with the objective of identifying potential therapeutic targets for type-2 diabetes. Throughout the course of our study, we have held several meetings with Novo Nordisk to evaluate how our data could be leveraged to advance diabetes-treatment strategies. These discussions have underscored the medical significance of our findings within the scope of human health. To further align our study with clinical relevance, a comprehensive review was undertaken by a senior scientist at Novo Nordisk, focusing on connections between the human *Lgr4* gene and diabetes. The results of this review have been included in the revised manuscript as a summary table (Supplementary Table 4) that outlines genetic evidence linking LGR4 to metabolic disorders in humans and mice, thereby highlighting the translational relevance of our work. The insights provided by the R&D division at Novo Nordisk deem the secretome and receptome hits identified in our study as compelling from a human-physiology perspective. We have therefore broadened the discussion in our manuscript to better illustrate our study's implications for human diabetes research and therapy. This enhancement clarifies the potential of *Drosophila* Rickets/LGR4 research to shed light on the functions of human LGR4 variants in metabolic conditions. Additionally, the screen hits presented in Figure 1d,e are notably relevant for translational science. Our screen is unique as it is the only *in-vivo* study encompassing the entire secretome and receptome, covering 2,256 genes and identifying 119 of these as important for sugar tolerance. Collaborative reviews with Novo Nordisk staff, supported by the data in Figure 1d,e, indicate that many of these genes correlate with diabetes and obesity in humans. The importance of our research for human physiology therefore goes beyond merely outlining the Bursicon-related muscle-neuron-fat communication pathways, as it provides direct functional evidence for the involvement in sugar metabolism of genes previously identified solely by GWAS. As such, our findings offer a valuable resource for understanding how these genes contribute to metabolic disorders, contributing significantly to the field of human metabolic physiology. We have emphasized this in the revised version. This comprehensive approach aims to clarify the significance of our study and of LGR4 in human metabolism and its potential as a target for therapeutic intervention in metabolic disorders.

Although we believe that the extended discussion and summarizing table mentioned above address Reviewer's point about linking our findings to human physiology to the degree that could be reasonably required, we chose to further explore the intriguing possibility that LGR signaling regulates adipocyte insulin sensitivity in mammals, as our results indicate it does in *Drosophila*. We therefore chose to knock down LGR4 – an ortholog of the *Drosophila* Bursicon receptor, Rickets – in cultured mouse adipocytes, using two different siRNAs. We have included an entirely new Figure 6 showing the importance of LGR4 in mouse adipocyte insulin sensitivity and signaling. *Lgr4*-deficient mouse adipocytes store reduced amounts of TAG, and they exhibit impaired insulin sensitivity as measured by phospho-AKT Western blotting after treatment with insulin – both phenotypes mirroring our observations in *Drosophila* fat tissue lacking the orthologous receptor Rickets. Furthermore, we employed quantitative mass-spectrometric phosphoproteomics, using 18-plex protein labeling to analyze multiple conditions in triplicate, revealing LGR4's important role in adipocytes' insulin response (Fig. 6e). Phosphoproteomics data highlighted a significantly reduced insulin reaction in *Lgr4*-deficient adipocytes, showing fewer phosphorylation changes after a 30-minute insulin treatment compared to controls (new Fig. 6f,g; Supplementary Table 2). Specifically, control adipocytes had 1,974 significant phosphorylation changes, while *Lgr4*-knockdown adipocytes showed dramatically fewer changes. The diminished insulin signaling in *Lgr4*-deficient cells underscores a significant insulin resistance following *Lgr4* loss. Pathway analysis confirmed the importance of LGR4 in insulin signaling and metabolic regulation (new Fig. 6l; Supplementary Table 3). Insulin stimulation in *Lgr4* siRNA-treated adipocytes showed a reduced phosphorylation response, indicating the important role of LGR4

signaling in maintaining insulin sensitivity and signaling in mouse adipocytes. Our study now not only connects *Drosophila* findings to mammalian models but also highlights the potential for LGR4-targeted therapies to enhance insulin sensitivity and treat diabetes.

Reviewer #2 (Remarks to the Author):

Reviewer: Resolving neuroendocrine circuits is a challenge and the authors provide substantial work towards this end. However, while the initial part is potentially interesting—provided the concerns below are addressed—and it is backed up by such substantial amount of work and experiments, the *NLaz* and *Relish* parts are underdeveloped or are somewhat out of focus, respectively. The *NLaz* finding is directly related to the study narrative, so it should be further substantiated with additional experiments. The *Relish/NFkB* part, in contrast, appears to be floating around and brings little to the discussion of the *Burs*-pathway role in sugar tolerance. As it is somewhat unrelated to the main narrative, it could be removed from the ms and explored more deeply in another manuscript, or placed in supplementary data.

The manuscript is overall very well written and if the major and minor concerns below can be addressed, the work should be a valuable contribution to different fields, including development, physiology, neurobiology, and metabolism.

Thank you for the opportunity to review this manuscript.

Alisson M. Gontijo

Response: We are grateful to Dr. Gontijo for his positive, thorough, and constructive feedback. We have improved our manuscript by incorporating additional genetic controls, offering more precise explanations of genotypes, and supplying further evidence of the insulin-stimulating effects of *Bursicon* signaling. Our revised work includes 32 new figure panels for the *Drosophila* part, plus an entirely new figure with 12 panels that extend the work to mouse adipocytes, and 5 new Supplementary Tables. This includes observations of gain-of-function phenotypes and variations in circulating insulin levels. We have also explored the effects on key genotypes under low-sugar-diet conditions. The section concerning *Nlaz* has been strengthened with multiple experiments that further support a role of *Nlaz* in modulating insulin signaling and metabolic regulation downstream of the *Rickets* receptor in adipose tissue under high-sugar conditions. As recommended, we have omitted the section on *Relish/NFkB*. Moreover, using *ex-vivo* live imaging, we have more clearly demonstrated the link between muscle signaling and *Bursicon* neuronal activity in high-sugar environments, and we have provided additional proof that *Tkv*, the receptor for muscle-secreted *Gbb*, is important in *Bursicon*-positive neurons for promoting insulin production and secretion from the insulin-producing cells (IPCs). These adjustments were made in alignment with your helpful suggestions and the revision strategy that we discussed. In addition, we have extended our findings into mammals, clearly showing that one of the mammalian *Rickets* orthologs, *Lgr4*, is required for normal insulin responses in cultured mouse adipocytes (new Figure 6). We now show that *Lgr4*-deficient mouse adipocytes store reduced amounts of TAG, and they exhibit impaired insulin sensitivity as measured by phospho-AKT Western blotting after treatment with insulin – both phenotypes mirroring our observations in *Drosophila* fat body lacking the orthologous receptor *Rickets*. Furthermore, we employed quantitative mass-spectrometric phosphoproteomics, revealing an essential role of LGR4 in the insulin response of mouse adipocytes. We believe that this perfectly illustrates the utility of fundamental exploration in *Drosophila* as a guide for translational science. Further details are provided below.

Major concerns

Reviewer point 1: Results Fig 2 and Extended data, Figure 2 a, b, and c. What are the exact genotypes of the controls? Without them it is difficult to evaluate the experiment and results. A file with the full genotypes of the animals used in each figure panel should be provided. Regardless, Figure 2b, for instance (this applies to many other experiments), is lacking many experimental controls. If one considers that the control is *Burs*^{>/+} (*Burs*[>] crossed to w[1118]?) alone, ideally, the controls UAS-

Burs-RNAi alone, Burs>Cas9 alone, UAS-Cas9 UAS-gRNA alone would be assayed for this experiment and for each other similar experiment (eg, Extended Data Fig 2 b,c), or at least in these initial fundamental experiments to establish the responses of each genetic component to the HSD treatment.

Response: We acknowledge the reviewer's feedback concerning the clarity of the genotypes and their recommendation to incorporate additional UAS controls in our initial experiments. We regret this oversight and are grateful for the chance to clarify our experimental approach and the genotypes involved. We have now included a supplementary file that gives the complete genotypes of the animals examined in each figure panel, which can be found as Supplementary Table 5.

We have improved the rigor of our work by refining our experimental controls and methodologies. We appreciate the suggestion to incorporate additional UAS-only controls in our initial experiments, and we apologize for omitting them in the first round. To ensure the highest standards in our genetic manipulations, we have developed an in-house genetic background. This maintains our fly lines within a w^{1118} -derived genetic population that retains a small amount of genetic variation, avoiding the introduction of artifacts that complete isogenization might cause. Our dedicated genetics expert oversees the creation of all transgenic lines, ensuring consistency within this genetic background. This approach includes careful selection of genetic backgrounds for our transgenes, even when combining them on different chromosomes or recombining them, to maintain uniformity across our experiments. We are advocating for the fly community to use a consistent but not fully isogenized genetic background, to maintain some genetic diversity without the health issues associated with complete homozygosity. Dr. Gontijo's insight into the necessity of testing UAS-RNAi lines alone is recognized. However, given our consistent genetic background, we believe that UAS-RNAi heterozygotes may not always serve as the most appropriate controls due to their differing genetic population backgrounds. Instead, the GAL4 driver controls, crossed into the same genetic background as the UAS-RNAi lines, offer a nearly identical genetic makeup, differing only by the RNAi construct itself. Our initial screenings utilized UAS-driven RNAi lines with a shared genetic background, differing only in the RNAi construct inserted. This method effectively rules out genetic background concerns and the potential for "leaky" UAS expression without a GAL4 driver. Our consistent genetic background approach across RNAi and CRISPR lines, combined with the insertion of constructs at the same genomic location, mitigates concerns over variable positional effects and off-target RNAi impacts. Our findings with *Burs* and *Pburs* knockdowns, as well as CRISPR constructs targeting these loci and the receptor gene *ricketts*, consistently demonstrate the expected phenotypes, validating our methodology.

We have now incorporated UAS-only controls for both RNAi and CRISPR lines throughout the manuscript as initially suggested, and new results (Supplementary Fig. 2a,f; 3c; 4b; and 5a,b in the revised manuscript) conclusively show that the observed reduction in sugar tolerance is not attributable to genetic background differences or unintended UAS expression. Furthermore, our comprehensive investigations employing diverse GAL4 drivers (*Mhc-GAL4*, *R57C10-GAL4*, *Burs-GAL4*, *Ilp2-GAL4*, *Cg-GAL4*, and the newly added *ppl-GAL4* in Supplementary Figure 4a) and RNAi/CRISPR lines targeting an array of genes (*UAS-Gbb-RNAi*, *UAS-Burs-RNAi*, *UAS-Burs-KO*, *UAS-Pburs-RNAi*, *UAS-Pburs-KO*, *UAS-rk-RNAi*, *UAS-rk-KO*) are consistent with one another. This consistency, along with the newly added TrpA1-mediated gain-of-function rescue experiments (new Fig. 7i,j and Supplementary Fig. 3e), gives strong support to the proposed sugar-tolerance signaling pathway connecting muscle to brain to IPCs and fat body.

In our revised manuscript, we have also provided this explanation of our genetic strategy in the method section, based on the arguments outlined above, highlighting how our methodology addresses any concerns related to controls: “A rigorous genetic standardization protocol for all GAL4 and GAL80 lines, as well as for combinations of transgenes, was implemented to ensure they shared a genetic background closely aligned with the w^{1118} strain. This standardization was achieved by initially crossing the balancer lines with our in-house w^{1118} strain to incorporate a genetic background analogous to w^{1118} . To introduce a controlled degree of genetic variation across all lines, we utilized at least 15 w^{1118} individuals for backcrossing purposes. Furthermore, by selecting females for these crosses, we standardized the genetic background of the X chromosome. As a result, any combination of transgenes

maintained this uniform genetic background. This approach ensured that the GAL4 driver controls were genetically almost identical to the RNAi or CRISPR animals, with the exception of the RNAi or gRNA construct itself. This was accomplished by crossing the GAL4 drivers into the w^{1118} genetic background in which the UAS-RNAi or UAS-gRNA lines were maintained, thereby aligning the genetic structure of our driver controls closely with those of the RNAi or CRISPR experimental groups."

We believe that this addition and the inclusion of these UAS controls has thoroughly addressed the reviewer's concerns.

Reviewer point 2: Extended data, Figure 2a-c: a rationale justifying the need for these analyses is not presented. The x-axis variable "Delta 50%-pupariation time (%)" is confusing or not very clear. I suggest using simple ratios or % change (of the 50% pupariation time). Regardless, please clarify how the calculations were made in the figure legend and/or methods - the way it is written in the manuscript is neither complete nor precise). Please apply the rationale to the data on Fig 4b and 4f: What does it show? Same to other similar figures of the ms, eg, Fig 5a, 6a, 6d, 7j-k. How do these results affect the conclusions?

Response: We appreciate the reviewer's attention to this detail. We have updated Extended Data Fig. 2a-c (now Supplementary Fig. 2b-e in the revised manuscript, according to *Nature Communications* guidelines) to make them congruent with the other developmental-timing graphs, now displaying pupariation with the 50% pupariation time indicated, adhering to the standard in the field. Additionally, we have clarified the calculation method in the figure legends throughout the manuscript, as suggested by the reviewer. This change in presentation style has no effect on the data themselves or on our interpretations or conclusions.

Reviewer point 3: Results relative to Figure 3: What is the effect of Burs-KO and RNAi in Burs cells on *ilp2-5* transcript levels in animals raised under a normal (non-HSD) diet? This is relevant as the effect of Burs CRISPR-KO or RNAi on *ilp* transcript levels is attributed to HSD conditions (as in this sentence "Taken together, these findings suggest that, ***in response to consumption of dietary sugar***, Bursicon signaling acts on the IPCs to promote the production of *Ilp2* and *Ilp3* to increase insulin signaling and enhance tissue uptake of circulating glucose.", but no data is shown in normal diet, so this effect could be diet independent.

Response: We appreciate the reviewer's emphasis on the importance of including non-HSD transcript levels to support statements such as "in response to consumption of dietary sugar, Bursicon signaling acts on the IPCs to promote the production of *Ilp2* and *Ilp3*, thereby increasing insulin signaling and enhancing tissue uptake of circulating glucose." As mentioned in our initial inquiry, our investigations into developmental timing under various dietary conditions revealed that manipulating Bursicon signaling led to phenotypes only under high-sugar-diet conditions (HSD, 30% sugar), giving no observable effects on food containing a "normal" 6% sugar level (normal diet, ND). Whole-animal *Thor/4EBP* expression data (Fig. 2f), (inversely) reflective of systemic insulin signaling, makes a compelling case, showing no response to *Burs* loss on ND while exhibiting upregulation on HSD when *Burs* function is lost. This observation suggests that systemic insulin signaling is indifferent to *Burs* on ND but requires *Burs* for full activity on HSD. The demonstration in Fig. 5f,g that the loss of the Bursicon receptor *Rickets* more strongly diminishes fat-body insulin response on HSD than on ND further supports this conclusion. As part of our revision process, we analyzed both whole-body pAKT and fat-body FOXO localization as insulin-signaling readouts under ND and HSD, fulfilling our promise to include at least one of these measurements. We depleted our anti-FOXO supply and received an apparently degraded batch from Pierre Léopold at the Institut Curie, which we attempted to use in extensive experiments, but we were unable to obtain satisfactory staining. However, our anti-pAKT assays gave quite clear results and offer direct insight into early insulin-pathway activity, whereas FOXO acts near the end of the pathway. The revised manuscript now incorporates our findings as a new Fig. 2g, showing that *Burs* loss has no detectable effect on AKT phosphorylation under ND conditions, whereas this loss strongly reduces pAKT in HSD-fed animals. Furthermore, we have also included new data showing that *ricketts* knockdown in the IPCs affects systemic TAG levels only under

high-sugar conditions (new Fig. 4g), which is also related to the point #5 below. Thus, our work indicates that the Burs-related phenotypes and changes in systemic insulin signaling and metabolism that we assessed are observable solely under HSD conditions, implying that Bursicon's effect on insulin signaling may not be great under ND.

To further address this point we have also revised the manuscript to specify that Bursicon signaling affects Ilp2 and Ilp3 production and subsequent insulin signaling and glucose uptake specifically in conditions of high dietary sugar, without suggesting Bursicon promotes these effects “in response” to high sugar intake. This revision aims to provide clarity and avoid potential misunderstandings. We also justify our focus on HSD conditions in the revised text by saying that: *Collectively, these findings suggest that loss of Burs expression impairs sugar tolerance, metabolic adaptation, and insulin signaling in animals consuming a high-sugar diet. Therefore, we focused our subsequent investigations on the effects of Bursicon signaling under high-sugar conditions.* We are grateful for the reviewer's feedback, and we hope these additions and changes resolve the matter positively.

Reviewer point 4: Results, Page 11: “We observed a trans-Tango-dependent tdTomato signal in the IPCs when the presynaptic ligand was expressed in Burs+ neurons (Fig. 3c),” The technical description of the experimental conditions of the trans-Tango is insufficient and the results depicted are lacking controls (especially the conditions without the GAL4 in normal diet and HSD diet; and images of the tdtomato and myrGFP channels for the IPCs should be included in all conditions). In addition, it would be helpful to show lower magnification projection stacks of whole CNS preparations in supplementary data. Full conditions of the experiments should be reported somewhere (such as in a dedicated methods section) or for all figures in supplementary materials. What are we looking at? anti-Ilp2 detected with which secondary antibody? Endogenous tdtomato fluorescence of the trans-tango and/or antibody stainings?

Response: We have undertaken additional experiments, as promised, to address the concerns raised here regarding the trans-Tango experimental setup and the omitted controls. We conducted new experiments using the trans-Tango technique under both normal diet (ND) and high-sugar diet (HSD) conditions, with and without the presence of the GAL4 driver, and included images captured at lower magnification. These additional data are included in a new Figure 3d and reinforce the conclusions drawn from our initial findings. To clarify the observations in Figure 3d, we have now incorporated a detailed description in the figure legend, specifying that the image depicts anti-Insulin-like peptide 2 (DILP2) staining in green and anti-tdTomato in red. This labeling allows for the clear visualization of the trans-Tango-dependent postsynaptic tdTomato signal in the insulin-producing cells (IPC) upon expression of the presynaptic ligand in the Bursicon-positive (Burs⁺) neurons. Furthermore, we have included the complete genotypes associated with these experiments in Supplementary Table 5, ensuring full transparency regarding the experimental setups. The revised experiments include tests with and without the GAL4 driver under ND and HSD, providing a comparison that strengthens the validity of our results. Our conclusion is that these results “indicate that the IPCs are directly postsynaptic to one or more the Burs⁺ neurons,” which also avoids overinterpretation. We believe these revisions address the reviewer's point.

Reviewer point 5: Results, Page 11. Relative to Figure 4: The delays in pupariation in Figures 1 and 2, for instance, are linked to compromised sugar tolerance, which is then linked to reduced Ilp levels, lowered TAG levels, and increased circulating sugar. rk-KO/RNAi and PKA-R1[BDK] manipulations in ilp2-GAL4 cells lead to sensitivity to HSD (HSD-dependent delay) and reduced Ilp levels, but have no effect on TAG levels (this is later justified by attributing fat body effects to a second, Burs-dependent role of Rk in the fat body). However, if there is no effect on TAG levels in ilp2>rk-KO/RNAi animals, but there is compromised high-sugar tolerance, how does this happen? What are the levels of circulating sugars in these animals? Why wasn't the ND condition also included in Fig 4g? The TAG levels in Fig 4g (HSD (10h) diet) look more similar to those of ND levels presented in other figures (eg Fig 2d-e and 5b). Is this variation normal? Does the ilp2> background respond differently to HSD? The authors need to address these questions experimentally and discuss how these results fit or not their model.

Response: We thank Dr. Gontijo for these comments. We have addressed the reviewer's concern by measuring triacylglyceride (TAG) levels in animals with *ricketts* knockdown and knockout in the IPCs under both normal diet (ND) and high-sugar diet (HSD) conditions, presented in a new Figure 4g. These results reveal that animals with IPC-specific *ricketts* loss exhibit reduced TAG levels under HSD but not ND, in agreement with the observed phenotypes of *Burs* or *Pburs* loss in the Burs-expressing neurons or the entire nervous system (Fig. 2c-e). We have also included new measurements of circulating glucose (Fig. 4h) that show that animals with IPC-specific knockdown or knockout of *ricketts* carry aberrantly high hemolymph glucose when they are fed for a short time on high-sugar diet.

Moreover, we have added multiple lines of evidence that strengthen our conclusion that Bursicon signaling in the IPCs enhances the production and secretion of insulin-like peptides (ILPs). Our data indicate that the developmental delays exhibited by animals lacking *ricketts* in the IPCs on a high-sugar diet are associated with decreased IPC function and lower circulating insulin levels. To directly test this hypothesis, we used the thermosensitive cation channel Transient Receptor Potential A1 (TrpA1) to artificially activate the IPCs. Exogenous stimulation of the IPCs with TrpA1 fully reversed the developmental delays caused by *rk* suppression in these cells (new Supplementary Fig. 3e). This finding confirms that the observed developmental delays are indeed a consequence of diminished IPC activity due to disrupted Bursicon-Rk signaling.

Furthermore, to elucidate the functional relationship between Bursicon-positive ($Burs^+$) neurons and the IPCs, we also used TrpA1 to artificially stimulate the $Burs^+$ neurons. This activation led to a large and statistically significant reduction of Ilp2 peptide staining intensity in the IPCs, as depicted in new Fig. 7i. This decrease in staining was not accompanied by changes in *Ilp2* expression (new Fig. 7j), suggesting that the reduction arises from enhanced peptide release rather than decreased synthesis. Conversely, the knockdown of the *tkv* receptor in the $Burs^+$ neurons resulted in apparent Ilp2 peptide accumulation within the IPCs and a concurrent decrease in *Ilp2* and *Ilp3* mRNA levels in the CNS, indicating inhibition of both insulin expression and release (Fig. 7i,j). Activation of the $Burs^+$ neurons through TrpA1 completely restored CNS *Ilp3* mRNA levels and mitigated Ilp2 peptide accumulation, even in the presence of *Tkv* knockdown. These comprehensive results support the notion that the $Burs^+$ neurons play a role in regulating the IPCs by promoting insulin expression and release. The inclusion of these new findings in the revised manuscript further underscores the significance of BMP signaling within the $Burs^+$ neurons as a critical regulatory mechanism in this context.

Reviewer point 6: Results Page 12, first paragraph: “Consistent with this, loss of Rk activity in the IPCs led to reduced levels of circulating levels of Ilp2 peptide in the hemolymph (Fig. 4e).” These results appear to be in the Normal Diet (ND) condition. This is a critical finding, suggesting that the effects of reduced Rk signaling on Ilp levels are independent of the HSD treatment. This further highlights the requirement of ND controls for the qPCR experiments for ilps on Fig 2 and 3. Please adjust the text to reflect this finding and limitation.

Response: In response to Reviewer Point 6, we recognize the significance of evaluating the effects of Rk signaling on Ilp levels under high-sugar conditions. We have now included the suggested measurements of circulating Ilp2 peptide levels in animals expressing *rk* knockdown or knockout in the IPCs, specifically under high-sugar diet (HSD) conditions. The results of these additional experiments are presented in a new Figure 4e and demonstrate that the loss of *rk* expression in the IPCs leads to reduced circulating Ilp2 levels under HSD conditions, consistent with the observed insulin-expression and -retention phenotypes. We have also included a new Supplementary Fig. 3d showing that whole-body *4EBP* transcript levels are reduced in animals with *ricketts* knockdown in the insulin producing cells under HSD-conditions, indicating lower systemic insulin signaling. We have revised the manuscript accordingly to reflect these findings.

Reviewer point 7: Results Page 13, Figure 5: Please indicate whether data on panels 5e are reused on Panel 5j. Specifically, at least condition CG>rk-KO in HSD looks exactly duplicated in these panels. The controls (CG>?) are nevertheless different. Why are the controls different and the CG>rk-KO condition reused? Data duplication must be explicitly justified and acknowledged. Please revise the

whole manuscript for such instances and clearly point them out in the manuscript methods and/or figure legends at least. For instance, Figure 2 panels 2d and 2e have duplicated controls in ND and HSD and this is not acknowledged.

Response: We thank the reviewer for the attention to detail. In our manuscript, two datasets were divided across different figure panels to maintain a degree of thematic separation. These relevant experiments were conducted simultaneously, facilitating a comprehensive analysis. In Fig. 2d and 2e, we measured TAG levels in *Burs*- and *Pburs*-knockdown and -knockout animals, along with a common control genotype; we chose to present the datasets for the two genes in separate panels, thus repeating the shared the control data. Similarly, datasets in Fig. 5e and 5j, showing hemolymph glucose concentrations in animals with fat-body-specific *rk* knockdown and in combination with *NLaz* knockdown, were obtained together but are presented in separate figures for clarity. Control data of absolute glucose values were measured in independent samples, showcasing the reproducibility between measurements. To ensure transparency and address the reviewer's concerns, we have revised the manuscript to ensure that the control is consistent in Fig. 5e and 5j, and we have acknowledged and explained this change in the figure legends.

Reviewer point 8: Please acknowledge/discuss the possible limitations for the model and conclusions of using CG-GAL4 as a driver for fat body to study systemic effects, considering that known fact that CG-GAL4 is also expressed in hemocytes. Can this be ignored? If yes, why?

Response: We acknowledge the concerns regarding the use of *Cg-GAL4* as a driver for studying systemic effects, given its expression in both the fat body and hemocytes. The *ex-vivo* insulin-stimulation data shown in Figure 5h, in which isolated fat-body explants were interrogated with an insulin treatment, show that the differences in acute insulin response are indeed fat-body-autonomous, since the preparations were devoid of hemocytes, although this does not show that the *origins* of this defect are cell-autonomous.

We have now conducted experiments using the *pumless (ppl)-GAL4* driver, which is known to specifically target the fat body to lack expression in hemocytes. The results, included in the revised manuscript as new Supplementary Figure 4a, demonstrate a similar delay induced by *rickets* knockdown in the fat body under high-sugar diet (HSD) conditions but not on a normal diet (ND). These findings with a second fat-body driver corroborate our initial results obtained with *Cg-GAL4* and reinforce the conclusion that the observed delay is indeed a consequence of fat-body-specific modulation of systemic physiology in response to HSD.

Reviewer point 9: Results Page 15, first paragraph: The evidence leading to the conclusion on the downstream placement of NLaz relative to Burs signalling is relatively weak. Only one readout (circulating sugar) of the *rk*-KO rescue is shown (what about the HSD-dependent delay, insulin-like peptide transcript levels (there should be no change?), and TAG levels?). The ND diet conditions are not shown, yet the effects under these conditions are required to interpret the results at HSD. Does NLaz RNAi in the fat body rescue the neuronal *Burs*-KO/RNAi effects and the *Gbb*-KO/RNAi effects? (see comment below on weaknesses regarding model testing).

Response: We appreciate the feedback on the initially presented evidence concerning the downstream placement of NLaz relative to Bursicon signaling. To address this concern and strengthen our conclusions, we have conducted additional experiments focusing on a range of phenotypic readouts beyond circulating sugar levels in *rk*-knockout conditions, specifically under high-sugar-diet (HSD) circumstances. Our initial investigations revealed that attenuating NLaz in fat-body tissue lacking *rk* is effective in restoring glycemic control. In our revised manuscript, we have further explored whether NLaz knockdown can also counteract other high-sugar-induced phenotypes in animals with fat-body-specific *rk* deficiency, including insulin signaling and lipid storage. In a fat-body-specific *rickets*-deletion background, concurrent NLaz silencing induces a significant increase in transcription of *Insulin receptor* measured in the whole larva (new Fig. 5k and new Supplementary Fig. 4d). This suggests that NLaz knockdown may enhance cellular sensitivity to insulin in the absence of Bursicon signaling,

potentially facilitating increased insulin-induced glucose uptake from the circulation and thus reversing hyperglycemia.

Since increased insulin-driven glucose uptake in the fat might be expected to promote storage of this material as lipid, we investigated the impact of NLaz loss on whole-animal TAG levels. Consistent with the hypothesis that NLaz attenuation enhances insulin signaling, we found that *NLaz* knockdown increases triacylglyceride (TAG) amounts in animals with fat-specific *rk*-locus deletion (new Fig. 5l and new Supplementary Fig. 4e), which otherwise exhibit a reduced capacity for fat storage under high-sugar conditions (Fig. 5b).

Please note that we have conducted new TAG measurements, now included for IPC knockdown of *ricketts* and neuronal *Bursicon* knockdown under both normal and high-sugar diets, to address the reviewer's comments #5 and #23. These are newly presented in Fig. 2c and 4g, using the same reagents that were employed elsewhere, ensuring consistency in concentration across all measurements. However, the Sigma kit that we used has ceased distribution and has a waiting period of over eight months, and therefore we were compelled to adopt a new TAG kit for additional NLaz-related experiments. After consulting with colleagues in the field, including Elizabeth Rideout (University of British Columbia, Canada) and Jason Tennessen (Indiana University, USA), who have both transitioned to Stanbio TAG kits – a kit not available in Europe – we managed to obtain a kit from Elizabeth Rideout for our revision experiments. Upon comparison of this Stanbio kit with the Triglyceride Trigs kit from Randox (#TR210), which *is* available in Europe, we found the latter to be more sensitive in our hands and opted to use it for the revision, suspecting potential degradation of the Stanbio kit during its transport from Canada to Denmark. Although the Randox kit reliably detects TAG and has been thoroughly tested in our lab, it is not as sensitive as the presently unavailable Sigma kit. We have therefore indicated in the methods section and the legends for Fig. 5i and Supplementary Fig. 4e that, due to the original kit's discontinuation, we have used the Triglyceride Trigs kit (Randox, #TR210). This alternative kit's seeming reduction in efficiency for *Drosophila* samples, resulting in lower TAG measurements, is noted. Since all the genotypes within each panel were assessed in the same way, intra-experiment comparisons remain valid, although these panels cannot be numerically compared with others.

Considering the distinct pathways regulated by insulin signaling for glucose uptake and lipogenesis (regulated via relocalization or post-translational modification of existing proteins) versus FOXO-mediated transcriptional regulation of *4EBP*, which primarily controls growth, the interaction point of NLaz within the insulin pathway, particularly in reversing insulin resistance, might selectively improve glucose and lipid metabolism without impacting growth-related pathways. Our examination of the impact of NLaz inhibition on *4EBP* expression revealed no significant short-term effect or even an increase in *4EBP* expression in animals with fat-specific *rk* knockout (new Fig. 5m and new Supplementary Fig. 4f), indicating a potential impairment in the pathway affecting FOXO.

Thus, while NLaz loss appears to ameliorate some metabolic phenotypes downstream of *Rk*, it does not seem to efficiently rescue the developmental-delay phenotype on a high-sugar diet, which was not fully reversed by NLaz knockdown (new Fig. 5n). This suggests that while NLaz inhibition may improve sugar tolerance, it may not fully correct growth defects. These comprehensive findings further support the notion that NLaz regulates aspects of metabolism downstream of *Bursicon*/*Ricketts* signaling in the fat body. We have discussed these findings in our revised manuscript.

In response to the suggestion of employing tGPH to examine the influence of NLaz on insulin sensitivity in the fat body through ex-vivo insulin stimulation experiments, we initially considered this approach as a potentially valuable method to directly assess changes in insulin signaling. However, this method required the generation of a transgenic combination involving fat-body *GAL4*, *rk* knockdown/knockout, *NLaz* knockdown, *GAL80^{TS}*, and the tGPH reporter. Unfortunately, the practical execution of this strategy proved to be infeasible. The technical challenge arose from the extremely low estimated frequency of obtaining the necessary recombinant strains containing the required transgenes, which was less than 1 in 500. Despite our efforts, we did not succeed in obtaining any recombinant offspring in our attempts. This limitation significantly hindered our ability to directly investigate the effects of NLaz knockdown on insulin sensitivity in the fat body using the proposed tGPH methodology. However, we believe that the extensive additions mentioned above strengthen the section on NLaz in relation to *Bursicon* signaling and address the reviewer's point.

Reviewer point 10: Results, Page 17-18. Relative to Fig 7b. This is a particularly weak experiment. The 29% increase albeit statistically significant is a minute effect and the range of the CaLexA signal in the assayed cells is completely overlapping with the control condition. Hence, conclusions on this single readout alone should be taken with great caution, but are not. This part could be stronger if the authors provided an independent measure of neuronal activity, e.g., GCaMP-based live imaging. In addition, parallel quantification of tdTomato fluorescence in the same cells assayed for CaLexA expression could be helpful to control for artefacts such as minute dietary effects on Burs-GAL4 expression levels (if there is no modulation, one would expect to see no difference between ND x HSD). This is critical because a small HSD-dependent Burs-GAL4-expression increase could lead to the observed CaLexA effects. The fact that Burs protein levels in Burs neurons are Gbb-signaling dependent (Veveritysa and Allan, Development 2011) and are slightly increased (Fig 7c-d, albeit not statistically significantly) in these cells indicate that this is indeed a possibility. Furthermore, increased pMad immunoreactivity (Veveritysa and Allan, Development 2011) could be a more direct alternative reading of increased BMP signalling in Burs neurons via Gbb.

Response: We appreciate the critical evaluation of our experiment presented in Figure 7b and the suggestion for more robust methods of assessing neuronal activity. We have conducted the technically challenging experiment proposed in our earlier response – using real-time GCaMP imaging to monitor Bursicon neuronal activity in larval preparations that maintained intact neuromuscular connections as the reviewer suggested. This approach allowed us to directly observe the neurons' response to high glucose levels. Our results, presented in Figure 7f, show strong calcium activity in Bursicon-positive neurons (compared to a coexpressed tdTomato ratiometric control) in muscle+brain preparations exposed to high-glucose hemolymph-like solution, corroborating the notion that these neurons are indeed responsive to metabolic cues. Strikingly, when the Bursicon-expressing neurons were severed from their muscle targets, or intact preparations were incubated in low-sugar medium, the activation of these neurons was eliminated or strongly reduced (respectively). These observations underscore the importance of continual acute retrograde signaling from the muscle to the Bursicon neurons for their activation in response to high glucose; this acute activation effect would be superimposed on the reported Gbb-induced effects on Burs and Pburs expression described by Veveritysa and Allan. We believe this additional data robustly supports our hypothesis and addresses the points raised by the reviewer. (We note again that the CaLexA results were indeed normalized to a UAS-tdTomato ratiometric control, an important experimental detail that we failed to mention clearly and have now endeavored to highlight. We also note again that the pMad staining is likely not feasible as explained in our initial response.

Reviewer point 11: The authors have the challenge of teasing out developmental and nutritional roles of BMP signalling. This could be achieved with GAL80 and temperature shift experiments at least within the context of the 10-h HSD assays. Alternatively, the authors should discuss the limitations of the study and alternative interpretations.

Response: We have discussed in the revised text the limitations of our approach and alternative interpretations of our results as agreed upon in our inquiry.

Reviewer point 12: In addition, the wording of the Gbb finding is discussed within the context of a systemic signal (see also summary figure 7L, where a black arrow goes from the muscle to the Burs-positive neurons), whereas the evidence is that Gbb functions as a non-systemic, localised retrograde signal from muscles to the efferent Burs-positive neurons, which project towards the body wall (Veveritysa and Allan, Development 2011).

Response: We have changed the summary figure to clearly illustrate the local retrograde nature of the Gbb signal.

Reviewer point 13: The proposed model implies that Gbb-IR or KO in the fat body will lead to increased circulating sugar levels. Ideally, this would be shown in Fig 7.

Response: We have changed the summary figure to show effects on circulating sugar.

Reviewer point 14: Finally, the NLaz and Gbb findings should be substantiated with rescue experiments that test the hypothetical models proposed. The experiments presented for NLaz and Gbb are very limited and the lack of such experiments are a weakness of this paper. For instance, an attempt to rescue the effects of Gbb-IR/KO in fat body could be made in many ways, such as by activation of Burs+Pburs neurons (using TRPA1 or optogenetics) or activation of Rickets-signaling in IPCs (maybe using constitutively active PKA? PKA-CQR?).

Response: We appreciate the reviewer's suggestion regarding the need for rescue experiments to substantiate our findings on NLaz and Gbb. We have thoroughly addressed this point by performing the suggested rescue and TrpA1 activation experiments that we proposed in our initial inquiry and are detailed in our response to point 5. In brief, as highlighted in our revised manuscript, we conducted experiments using the thermosensitive cation channel TrpA1 to artificially activate control and *rk*-deficient IPCs or control and *tkv*-deficient Burs⁺ neurons. These experiments successfully reversed the developmental delays caused by *rk* suppression in IPCs and significantly altered the staining intensity of Ilp2 peptide levels in the IPCs, directly testing and supporting our hypothesis regarding the regulatory role of Bursicon signaling in IPC functionality and insulin sensitivity. These findings, presented in the new Supplementary Fig. 3e and Fig. 7i and 7j, provide robust support for our proposed models. We believe these additions significantly strengthen the manuscript and invite the reviewer to examine our more detailed response to point 5 for further clarification.

Minor concerns

Reviewer point 15: Fig 1. Statistics: “ANOVA with Šidak’s correction” Please verify and clarify. Sidak’s correction is applied to a post-hoc multiple comparison test, not to ANOVA per se. Which tests were performed here and corrected with Sidak’s correction after the ANOVA? T-tests?

Response: Thanks for pointing this out, which we have now fixed.

Reviewer point 16: “Statistics” in other figures: “Dunnet’s correction” is mentioned many times. Please verify and clarify. Do the authors mean the Dunnet’s multiple comparison test? If yes, it is not a correction applied to a test, but rather a post-hoc test itself.

Response: We have corrected the terminology. Thanks for pointing this out.

Reviewer point 17: Fig 2a: please confirm that the figure legend indeed corresponds to the plotted lines. (specifically the control and heterozygote in HSD). If it is correct, the effect of heterozygote animals is quite large and to the opposite direction as to the homozygote animals. This should be acknowledged in the text.

Response: We confirm this and now draw attention to it in the figure legend.

Reviewer point 18: Fig 2 (a-b), Extended data Figs 1, 2 (d), Fig 4 b,f - legends say pupation time, but figures say pupariation time. Please clarify.

Response: We thank the reviewer for catching this, which we have now corrected to “pupariation” throughout.

Reviewer point 19: Results Page 7, 1st paragraph dLgr2/Rickets (orthologous with mammalian Lgr4/-5). Please include -6 (mammalian Lgr6). dLgr2 is equally distantly related to mammalian Lgr4, Lgr5, and Lgr6.

Response: We have clarified that *Drosophila* Rickets is orthologous with mammalian Lgr4, Lgr5, and Lgr6.

Reviewer: point 20 Results, Page 8, 3rd paragraph, last sentence “This further supports to the importance...” maybe revise the wording.

Response: We have corrected the sentence.

Reviewer point 21: Extended data, Figure 2, Statistics :”Error bars represent mean and SEM.” please verify and confirm that the bars in these panels indeed indicate the standard error of the mean. If not, please correct. The error bars seem inconsistent with SEMs considering the distribution of the data points depicted. They look more like SD—which is good, as it is preferable than SEM—or some other measure. Please also verify and correct if necessary Fig 1C.

Response: We confirm that error bars correspond to the SEM in Figure 2, while in Figure 1c, they do indeed show the SD, which is now described correctly in the legend. Thanks for catching this.

Reviewer point 22: Results, Fig 2: What was the Cas9 construct used for the somatic KO eg on Fig 2? I did not find it in the methods section. Full genotypes should be reported somewhere.

Response: We will have included details about the Cas9 construct. We have also included a new supplemental table 4 listing every genotype for each figure panel.

Reviewer point 23: Results, Fig 2 panels c-e: please show TAG levels as mg/mgprotein as in panels d and e.

Response: We no longer retained the raw numbers, so we have performed the experiment again, and the new data that show TAG levels as mg/mg protein is included in a new Fig. 2c in the revised manuscript.

Reviewer point 24: Results, Page 9: “an effect that was abolished by Burs loss (Fig. 2c).” should read “..by RNAI or CRISPR-targeting of Burs in Burs[>] cells” to avoid confusion with the Burs mutation analyses.

Response: We have changed the wording to comport with the reviewer’s suggestion.

Reviewer point 25: The fact that neuronal Burs knockdown (partial attenuation of TAG increase) only partially mimics Burs knockdown in Burs[>] cells (complete abrogation of TAG increase) should be discussed.

Response: We have included new TAG measurements in Fig. 2c and discussed them, which addresses this point.

Reviewer 26: Results, page 9: “Consistent with reduced insulin signaling, loss of Burs led to elevated glycemic levels after chronic or short-term (10-hour) exposure to a high-sugar diet (Fig. 2h,i).” Please mention that the effect of the Burs CRISPR-targeting in the short-term (10h) treatment condition is not statistically significant, so this statement is only statistically precise for the RNAi condition.

Response: We have corrected this statement in the revised manuscript.

Reviewer point 27: Results Page 12, first paragraph: “This is further supported by single-nucleus transcriptomics³⁷.” Please show the specific data/analyses of the data that supports the claim that *rk* is expressed/enriched in IPCs in a supplementary figure.

Response: We have revised the manuscript to include single-nucleus transcriptomics data in a new Supplementary Fig. 3b. These data, from FlyCellAtlas⁵, demonstrate overlap between *Ilp2* and *rk* in an IPC-enriched sample, indicating Rickets expression in these cells.

Reviewer point 28: Results Page 12: "...IPCs led to HSD-dependent delay..." maybe missing an "an" before HSD

Response: Thanks for catching this, and we have fixed it.

Reviewer point 29: Results Page 12, 2nd paragraph: "To identify the target tissues mediating any effect of humoral Bursicon signaling on fat storage, we assessed the tissue expression of rk and found strong expression in the fat body..." This statement lacks support of data or reference. Please show the data.

Response: We have now provided a reference to the FlyAtlas transcriptomics data showing expression of *ricketts* in the fat body.

Reviewer point 30: Results Page 13, Figure 5: again several panels missing controls. eg, panel 5i missing RNAi alone. Panel 5j missing minimal controls such as UAS transgenes alone and CG>NLaz-IR alone. These controls are critical to interpret the results.

Response: Please see our response to Point #1 above regarding UAS controls and our response to Point #9 with respect to additional data to strengthen the NLaz angle.

Reviewer point 31: Results Page 14, first paragraph: "...found that reducing NLaz expression in the fat alone completely.." maybe missing "body" before alone?

Response: Thanks – we have fixed it.

Reviewer point 32: Results Page 15, first paragraph: "We then tested the ability of NLaz deficiency to..." Please use NLaz RNAi instead of deficiency.

Response: Thanks – we have changed this.

Reviewer point 33: Results, Page 15, 3rd paragraph - "We observed that, in animals fed a high-sugar diet, knockdown of relish in the fat body led to reduced nuclear localization of FOXO (Fig. 6c) and an associated reduction in 4EBP expression (Fig. 6b), indicating increased insulin signaling in the fat tissue ". Again, what occurs in ND? Is the effect of relish on FOXO localisation dependent on HSD?

Response: We have removed the Relish-related data, which addresses this point.

Reviewer point 34: Results, Page 17, 1st paragraph: "Strong calcium-induced GFP signal was observed solely in Burs+ neurons that produce the Burs:Pburs heterodimer, indicating that these are the active Burs+ neurons (Fig. 7a). " It is not clear if this sentence and Fig 7a refer to a CNS preparation under ND or HSD. Please clarify. Also, the statement "that these are the active Burs+ neurons" is too ample and vague. It can read true or false under different conditions, so please specify which specific condition it applies to."

Response: We have clarified this in our statements, and information is also given in the figure legend.

Reviewer point 35: Results, Page 17: 2nd paragraph: Typo "...pathway were identified as a hits in our screen..."

Response: Thanks – we have fixed it.

Reviewer point 36: Results Page 18, Fig 7L: the label "Ricketts (Lgr4)" is a bit confusing. Ricketts is Lgr2 in *Drosophila* and, in *Drosophila*, Lgr4 is another receptor (orthologous to mammalian Lgr7-8). Hence, please indicate the fly equivalent (Lgr2 (or dLgr2)) or include "mammalian Lgr4/5/6".

Response: Thanks. We have now included this in the figure and figure legend.

Reviewer point 37: Discussion, Page 19, 2nd paragraph: “We discovered that Rickets,...” Please include its alternative name “We discovered that Rickets/Leucine-rich repeat- containing G-protein coupled receptor 2 (Lgr2 (or dLgr2, if necessary/preferable)),...” to avoid confusion (see above). Please refer to the orthology to the other mammalian Lgrs. Please mention known ligands to the members of this mammalian Lgr4/5/6 family (R-spondins (Roof plate specific spondins (RSPOs))/Norrin (Norrie Disease Protein, specific for Lgr4)/RANKL (Receptor Activator for Nuclear Factor κ B Ligand)), at least one of which is a cysteine knot protein, Norrin).

Response: Thank you for your valuable feedback. In light of your suggestion, we have revised the text in the discussion to clarify that the ligands for these mammalian receptors are known to include R-spondins, RANKL, Nidogen-2, and Norrin, a cysteine-knot protein orthologous to the *Drosophila* Burs/pBurs, and included references^{6,7}.

Reviewer point 38: Methods, Metabolite assays, Line 13 revise question mark: “collecting the exudate with a pipette?”

Response: Nice catch – we have fixed it.

Reviewer point 39: Methods, transgene construction: Is the FLAG::Burs::HA viable or have any Burs-like phenotypes?

Response: The FLAG::Burs::HA knock-in is homozygous-viable and has no obvious phenotype. We have now mentioned this in the revised document.

Reviewer #3 (Remarks to the Author):

Reviewer: Even though insulin insufficient and insulin resistance have been well established to promote high-caloric diet-induced hyperglycemia, the endocrinal signals mediating different organs to amount systemic responses still remain largely unclear. Integrating RNAi screening against secreted proteins and receptors, Kubrak et al uncovered neuronal Burs governs both insulin synthesis and peripheral insulin sensitivity through its receptor RK in the context of HSD. Further, they found that neuronal Burs production is regulated by muscle-produced Gbb. The findings are pretty interesting and should provide significant impacts in the field of diet-induced hyperglycemia and disease development such as obesity and type 2 diabetes. However, some of the experiments were not designed very carefully. The authors also need to address a few important comments prior to publication in Nature Communications.

Response: We are grateful to the reviewer for the encouraging comments regarding our work. We deeply appreciate the time spent and the constructive feedback offered. We are pleased that the reviewer finds our work interesting and believes it will have a significant impact on the field of diet-induced hyperglycemia and related disease development, such as obesity and type-2 diabetes. We have substantially revised our manuscript, incorporating a wealth of new data from additional *Drosophila* experiments (32 new figure panels in the revised manuscript) to strengthen various aspects of our study and address the points raised by the reviewers. Additionally, we have introduced new findings on mammalian LGR4, an ortholog of the *Drosophila* Bursicon receptor Rickets. These data are included in a new Figure 6 with 12 panels that extend the work to mouse adipocytes. Our revised work also includes 5 new Supplementary Tables.

Our *Drosophila* research demonstrates that Rk plays an important role in insulin sensitivity within the fat body, showing that a loss of Rk decreases triglyceride levels and impairs the response to insulin stimulation. In the revised manuscript, we have extended our findings to mammals, demonstrating through cell-culture experiments that cultured mouse adipocytes require LGR4 for normal insulin sensitivity (new Fig. 6). Knockdown of *Lgr4* in these cells using either of two siRNAs leads to reduced

TAG levels and a significantly diminished response to insulin stimulation. Furthermore, we employed quantitative mass-spectrometric phosphoproteomics, indicating that *Lgr4* is essential for maintaining adipocyte insulin sensitivity. These new data not only strengthen our findings but also broadens the relevance of our research to the fields of diet-induced hyperglycemia, obesity, and type-2 diabetes. In the sections that follow, we detail how we have addressed each point raised by the reviewer, specifically focusing on improvements made to our manuscript in response to concerns about the experimental design.

Major comments:

Reviewer point 1: Previous studies have shown gut-derived Burs dramatically affects systemic energy homeostasis (PMID: 30344016). I recommend authors to check whether gut-derived Burs is involved in the condition in this study using *pros-Gal4* and other gut specific *Gal4* lines, even though they proposed the predominant roles of neuronal Burs. They also need to confirm the Burs source in brain using specific *nSyb-Gal4* that does not target gut cells (PMID: 32917721). No matter Burs mutation or *Burs*>*Burs*-RNAi diminishes Burs production in the whole body. Figuring out the real sources of functional Burs would be an important question to be addressed.

Response: We thank the reviewer for highlighting this matter, and we apologize for not previously addressing the potential contribution of gut-derived Bursicon in our initial submission. Indeed, it is a pertinent inquiry, since enteroendocrine cells (EECs) of the adult midgut have been identified as a source of Burs that impacts metabolism⁸. The biologically active Bursicon hormone that activates the Ricket receptor is a heterodimer of Burs (Burs-alpha) and Pburs (Partner of Bursicon or Burs-beta), synthesized by a specific subset of neurons in the larval central nervous system (CNS)^{9,10}. However, the EECs under discussion exclusively produce Burs-alpha and not Pburs, and therefore they are unable to produce the active Burs:Pburs heterodimer essential for canonical Bursicon signaling.

Our research concentrated on these neurons that produce both Burs and Pburs, especially given that we had found they were activated under high-sugar conditions. To support this further in our revised manuscript, we have included new real-time imaging data using GCaMP to observe the activity of these neurons that coexpress Burs and Pburs, allowing us to directly monitor their response to elevated glucose levels (new Fig. 7f).

Although Burs and Pburs can each homodimerize, the biological significance of these homodimers remains largely undefined, with some evidence pointing to a role in innate immunity¹¹ and functioning from the gut in adult stages⁸. However, it is unclear whether these homodimers act through the Rickets receptor, as the immune modulatory functions of Burs homodimers appear to be independent of Rickets. Our findings, indicating the involvement of both Burs and Pburs, along with the Rickets receptor, suggest the nervous system as the primary source. Importantly, our data demonstrate that knockdown of either *Burs* or *Pburs* in the nervous system results in the sugar tolerance-phenotype, using a driver (*R57C10-GAL4*) confirmed to target neurons exclusively and not EECs^{12,13}.

In response to the reviewer's suggestion, we have conducted additional experiments to explore the role of gut-derived Bursicon, employing the *pros-GAL4* (also known as *voilà-GAL4*) driver for EEC-specific knockdown. Our results, now included as new Supplementary Figure 2g, show that *Burs* knockdown in the EECs does not alter the sugar-tolerance phenotype. This new data and its implications are discussed in our revised manuscript to comprehensively address the reviewer's point.

Reviewer point 2: Only loss-of-function assays were performed in this study. Will gain-of-function of Burs signaling rescues diet-induced insulin resistance and hyperglycemia? How about overexpression of *Rk* or *TrpA1*-induced activation of Burs cells (neurons and gut cells)?

Response: We thank the reviewer for raising this question. We have now included gain-of-function experiments using the thermosensitive cation channel Transient Receptor Potential A1 (*TrpA1*), as the reviewer suggested. The exogenous activation of the IPCs with *TrpA1* fully reversed the developmental delays caused by *rk* loss in these cells (new Supplementary Fig. 3e). This finding confirms that the observed developmental delays are indeed a consequence of diminished IPC activity due to disrupted Bursicon-Rk signaling.

Furthermore, to elucidate the relationship between Bursicon-positive (Burs⁺) neuron activation and IPC functionality, we also used TrpA1 to artificially stimulate the Burs⁺ neurons. This activation led to a significant reduction of insulin-like peptide 2 (Ilp2) peptide levels in the insulin-producing cells (IPCs), as shown in new Fig. 7i. This decrease in peptide levels was not accompanied by changes in *Ilp2* mRNA expression (Fig. 7j), suggesting that the observed reduction arises from enhanced peptide release rather than decreased synthesis. Conversely, the knockdown of the Tkv receptor in the Burs⁺ neurons resulted in Ilp2 accumulation within the IPCs and a concurrent decrease in *Ilp2* and *Ilp3* mRNA levels in the CNS, indicating disruptions in both peptide synthesis and release (Fig. 7i,j). Activation of the Burs⁺ neurons through TrpA1 completely restored CNS *Ilp3* mRNA levels and mitigated Ilp2 peptide accumulation, even in the presence of *tkv* knockdown. These comprehensive results support the notion that Burs⁺ neurons regulate the IPCs by promoting ILP expression and release. The inclusion of these new findings underscores the connection between Burs⁺ neuronal activity and IPC release of ILPs as well as the significance of BMP signaling within the Burs⁺ neurons as a critical regulatory mechanism in this context.

Reviewer point 3: The authors need to repeat most of the development assays on HSD to make results of control flies consistent. For example, pupation occurs between day 6-8 (Fig. 1a), day 8-10 (Fig. 1b, 4f, 6b), day 8-9 (Fig. 4b, 5a, 7j), day 7-10 (Fig. 6a), day 9-12 (Fig. 7k). The differences could be caused the problem of control flies but not manipulations.

Response: The primary reason for the observed differences in developmental timing among different figures is that some experiments were conducted at 25 °C, whereas others were performed at 29 °C. This approach was adopted because neuronal gene knockdown and tissue-specific somatic CRISPR exhibit significantly higher efficiency at 29 °C compared to 25 °C. Development proceeds approximately 24 hours faster on a normal diet at 29 °C than at 25 °C, which accounts for some of the discrepancies the reviewer has highlighted. We regret not having clearly specified this in our initial submission. In our revised manuscript, we have now clearly stated the temperatures in the figure legends to address this point.

Furthermore, the gain-of-function experiments included in the revised manuscript mentioned above in our response to the reviewer's point #2 further underscore the specificity of the developmental timing phenotypes observed. Additionally, we have added a detailed description in the revised methods section on how the genetic background was carefully controlled. This was achieved by crossing lines into a consistent genetic background that retains a small amount of genetic variation. This strategy prevents the introduction of artifacts that complete isogenization might cause, while ensuring consistency within this genetic background. It is important to note that due to this consistency and our meticulous approach, our measurements of developmental timing to pupariation under normal dietary conditions are consistent within each temperature. However, developmental timing on a high-sugar diet can exhibit some batch-to-batch variation, perhaps due to osmotic balance and evaporation that can affect each batch's properties. Therefore, we performed all experiments comparing genotypes on both a normal diet and a high-sugar diet concurrently; no experiments were split across multiple food batches. This method ensures the highest quality of results and ensures that intra-experiment comparisons are valid. It is not feasible to retest every genotype simultaneously, as this would involve managing many hundreds of vials containing many thousands of animals.

In revising the manuscript, we have also incorporated UAS-construct-alone controls for both RNAi and CRISPR lines throughout, as initially suggested. This is now illustrated in six additional figures in the revised manuscript (Supplementary Fig. 2a,f; 3c; 4b; 5a,b), which conclusively demonstrate that the observed reduction in sugar tolerance is not attributable to differences in the genetic background or to unintended UAS expression. We believe these additional controls, together with our comprehensive approach—employing diverse GAL4 drivers (*Mhc-GAL4*, *R57C10-GAL4*, *Burs-GAL4*, *Ilp2-GAL4*, *Cg-GAL4*, and the newly added *ppl-GAL4* in Supplementary Figure 4a) and RNAi/CRISPR lines targeting distinct genes (*UAS-Gbb-RNAi*, *UAS-Burs-RNAi*, *UAS-Burs-KO*, *UAS-Pburs-RNAi*, *UAS-Pburs-KO*, *UAS-rk-RNAi*, *UAS-rk-KO*)—along with the newly added gain-of-function phenotypes that show TrpA1 activation can rescue the tested phenotypes (Fig. 7i,j and Supplementary Fig. 3e), strongly support the delineated signaling pathway from muscle to brain to IPCs and fat body. We have taken extensive measures to ensure the reliability and reproducibility of our findings, considering the

inherent variability of developmental timing on a high-sugar diet. Our methodology, including parallel experiments on normal and high-sugar diets and careful control of genetic backgrounds, addresses these concerns by ensuring high-quality, consistent results across the study, thereby addressing the reviewer's point.

Minor issues:

Reviewer point 4: For most ILP production assays, the results of decreased ILP2 level in both intracellular accumulation in IPCs and hemolymph do not support the regulation of SECRETION. Please modified the statement as “production/synthesis”.

Response: We acknowledge the reviewer's point regarding our interpretation of the insulin-like peptide (ILP) production assays. We agree that the observed decrease in ILP2 levels within the insulin-producing cells (IPCs) and hemolymph suggests changes in production/synthesis rather than secretion, since we also observe decreased *Ilp* gene expression (*e.g.*, Fig. 3a-c). Accordingly, we have updated our manuscript to reflect these findings accurately. Additionally, we introduce new data showing that the acute activation of Bursicon neurons via TrpA1 reduces *Ilp2* staining in the IPCs (new Fig, 7i,j). This suggests that Bursicon signaling may influence both synthesis and secretion of ILPs, offering a more nuanced understanding of ILP regulation. We believe this adjustment and the inclusion of new data provide a clearer and more comprehensive view.

Reviewer point 5: The knockdown assays of Burs and signaling worsened the HSD-induced hyperglycemia. The authors should modify some statements like “cause HSD-induced hyperglycemia”.

Response: We appreciate the reviewer's insightful comment and agree that our language regarding the effects of Burs and its signaling pathway on HSD-induced hyperglycemia needed clarification. In response, we have thoroughly revised the manuscript to more accurately reflect the role of Bursicon signaling in *modulating* hyperglycemia under a high sugar diet. Instead of stating that knockdown "causes HSD-induced hyperglycemia," we have adjusted our language to indicate that the disruption of Bursicon signaling exacerbates or contributes to the severity of HSD-induced hyperglycemia.

Reviewer point 6: Fig 6 should be moved into supplementary data, as the results were irrelevant to the main conclusion and complicated the molecular mechanisms of Burs signaling.

Response: In response to Reviewer 2's feedback, we have removed the Relish-related data from the main manuscript and shifted our focus towards reinforcing the role of Nlaz. This adjustment is reflected in new data presented in Figures 4k,l,m and Supplementary Figures 4d,e,f. These changes aim to simplify the narrative and strengthen the evidence supporting our conclusions, directly addressing the reviewer's concerns.

Reviewer point 7: Remove the “FoxO” in the regulation of muscle Gbb production in working model in Fig. 7l. the authors did not provide evidence. Moreover, it might lead to a conflict mechanism proposal like “muscle insulin resistance -> Gbb -> Burs -> fat body insulin resistance”. Insulin resistance in the muscle is earlier than the fat body?

Response: We have removed the mention of "FoxO" in the regulation of muscle Gbb production from our working model in Fig. 7l as suggested. We acknowledge the lack of direct evidence in our study for FoxO's involvement and the potential for confusion regarding the sequence of insulin resistance development. This adjustment aims to clarify our model and avoid any misleading interpretations.

References

- 1 Filipowska, J., Kondegowda, N. G., Leon-Rivera, N., Dhawan, S. & Vasavada, R. C. LGR4, a G Protein-Coupled Receptor With a Systemic Role: From Development to Metabolic Regulation. *Front Endocrinol (Lausanne)* **13**, 867001 (2022). <https://doi.org/10.3389/fendo.2022.867001>
- 2 Styrkarsdottir, U. *et al.* Nonsense mutation in the LGR4 gene is associated with several human diseases and other traits. *Nature* **497**, 517-520 (2013). <https://doi.org/10.1038/nature12124>
- 3 Wang, J. *et al.* Ablation of LGR4 promotes energy expenditure by driving white-to-brown fat switch. *Nat Cell Biol* **15**, 1455-1463 (2013). <https://doi.org/10.1038/ncb2867>
- 4 Zhang, N., Yuan, M. & Wang, J. LGR4: A New Receptor Member in Endocrine and Metabolic Diseases. *Endocr Rev* **44**, 647-667 (2023). <https://doi.org/10.1210/endrev/bnad003>
- 5 Li, H. *et al.* Fly Cell Atlas: A single-nucleus transcriptomic atlas of the adult fruit fly. *Science* **375**, eabk2432 (2022). <https://doi.org/10.1126/science.abk2432>
- 6 Deng, C. *et al.* Multi-functional norrin is a ligand for the LGR4 receptor. *J Cell Sci* **126**, 2060-2068 (2013). <https://doi.org/10.1242/jcs.123471>
- 7 Glinka, A. *et al.* LGR4 and LGR5 are R-spondin receptors mediating Wnt/beta-catenin and Wnt/PCP signalling. *EMBO Rep* **12**, 1055-1061 (2011). <https://doi.org/10.1038/embor.2011.175>
- 8 Scopelliti, A. *et al.* A Neuronal Relay Mediates a Nutrient Responsive Gut/Fat Body Axis Regulating Energy Homeostasis in Adult Drosophila. *Cell Metab* **29**, 269-284 e210 (2019). <https://doi.org/10.1016/j.cmet.2018.09.021>
- 9 Honegger, H. W., Dewey, E. M. & Ewer, J. Bursicon, the tanning hormone of insects: recent advances following the discovery of its molecular identity. *J Comp Physiol A Neuroethol Sens Neural Behav Physiol* **194**, 989-1005 (2008). <https://doi.org/10.1007/s00359-008-0386-3>
- 10 Luo, C. W. *et al.* Bursicon, the insect cuticle-hardening hormone, is a heterodimeric cystine knot protein that activates G protein-coupled receptor LGR2. *Proc Natl Acad Sci U S A* **102**, 2820-2825 (2005). <https://doi.org/10.1073/pnas.0409916102>
- 11 An, S. *et al.* Insect neuropeptide bursicon homodimers induce innate immune and stress genes during molting by activating the NF-kappaB transcription factor Relish. *PloS one* **7**, e34510 (2012). <https://doi.org/10.1371/journal.pone.0034510>
- 12 Kubrak, O. *et al.* The gut hormone Allatostatin C/Somatostatin regulates food intake and metabolic homeostasis under nutrient stress. *Nat Commun* **13**, 692 (2022). <https://doi.org/10.1038/s41467-022-28268-x>
- 13 Malita, A. *et al.* A gut-derived hormone suppresses sugar appetite and regulates food choice in Drosophila. *Nature Metabolism* **4**, 1532-1550 (2022). <https://doi.org/10.1038/s42255-022-00672-z>

REVIEWER COMMENTS

Reviewer #1 (Remarks to the Author):

It has been revised mainly according to comments. However, in the reply, "your study now not only connects Drosophila findings to mammalian models but also highlights the potential for LGR4-targeted therapies to enhance insulin sensitivity and treat diabetes" seems extended too much. Please add limitation(s) to strengthen this report.

Reviewer #2 (Remarks to the Author):

The reviewed manuscript is greatly improved and it has an overall very high standard. The authors have addressed all of my comments. The manuscript will be a valuable contribution to different fields.

Minor comments:

Please remove Relaxin from the new title and maybe provide a more specific LGR pathway. The study does not address relaxin signaling in any aspect, so the new title is inaccurate and misleading. Indeed the only time Relaxin is used is at the title. "LGRs" are a large family of GPCRs with a large diversity of ligands and functions, which are classified as multiple types (Type A, B, C1 or C2; Van Hiel et al., 2012). The type studied in the manuscript (Fly Lgr2 (rickets)/human LGR4) are Type B LGRs, while the relaxin receptor family of LGRs are Type C1, as they have a different domain architecture, different ligands, and different roles. Hence neither relaxin receptors, nor relaxin peptides, nor relaxin signaling was addressed in the manuscript. This is further important here to avoid confusion between Human LGR4 (a Type B LGR, orthologous to Drosophila Lgr2) and Fly Lgr4 (a Type C1 relaxin receptor, orthologous to human LGR7/8).

In the abstract (line 32) it is stated "interaction between ..BMP.. and .. Leucine-rich repeat-containing G-protein coupled Receptor (LGR) signaling pathways". As discussed above, this is too vague and uninformative. Please specify the pathway.

Introduction Line 83 - please double-check if "analog" is the appropriate term here.

Figure 1A - maybe the fly cartoon is missing one red eye?

Results, line 189 (Fig2c): "... an effect that was abolished by RNAi or CRISPR-targeting of Burs in Burs+ cells (Fig. 2c)." The new Fig 2c only shows RNAi data, so please adjust the text.

I recommend placing Supplementary Fig. 3e in the main figures.

Results Line 360, "These results elucidate the role of the Drosophila LGR4 ortholog, Rickets/dLgr2,..". Maybe adjust to "These results elucidate the role of the Drosophila ortholog of the mammalian Lgr4, Rickets/dLgr2,..".^[1]_{SEP}

Results line 399, maybe a typo in "in controls responded must less strongly...?"

Results Line 439-440: "We found that knockdown of gbb in the muscles or of tkv in the Burs+ neurons reduced the expression of Pburs in the CNS (Supplementary Fig. 4g)". I could not find the data on gbb RNAi in muscle cells regarding Pburs in the CNS in Supplementary Fig. 4g.

Reviewer #3 (Remarks to the Author):

The authors have addressed some of my comments, such as the functional Burs from neurons and inconsistent pupation times, in revised manuscript. However, the authors may have misinterpreted my concerns regarding "rescued phenotypes of delayed pupation" with "gain-of-functional assays." Specifically, it remains unclear whether the burs/rk gain-of-function is SUFFICIENT to increase insulin signaling and promote pupation under HSD conditions. Meanwhile, the newly included data on mammalian adipocytes, while interesting, shifts the focus towards adipose – not neuronal – metabolic regulation by rk/LGR4. This renders the results on ILP2 regulation somewhat irrelevant to the core findings. The authors should consider reorganizing the manuscript to deprioritize the neuronal regulation and address CRITICAL issues of adipose regulation. While the authors have incorporated a lot of efforts and additional experiments, a major revision is still needed to focus the manuscript and address the key remaining concerns.

Major comments:

1. Burs/rk gain-of-function under HSD: The manuscript demonstrates that burs/rk loss-of-function exacerbates HSD phenotypes, but it doesn't establish whether burs/rk activation is SUFFICIENT to promote pupation and insulin signaling under HSD conditions. To address this, consider experiments like TrpA1 activation or Burs overexpression in Burs+ neurons. Examining insulin signaling markers (tGPH, 4EBP), pupation rates, and hemolymph glucose levels under HSD could validate a real "protective" role for burs/rk as the authors claimed in the manuscript. These fly results would also strengthen the connection to mammalian LGR4 as a potential diabetes target, since they included mammalian results to highlight the disease relevance.

2. rk expression in larval fat body. Previous studies suggest that rk is not likely to be expressed in the adult fat body. The flycellatals also revealed that rk is actually expressed in the adult hemocytes, but not fat body (bmm expression). I really doubted that whether rk is expressed in the larval fat body. While FlyAtlas microarray data shows rk expression in the larval fat body, FlyAtlas 2 RNAseq data contradicts this. That could be the false positive results from non-specific signals in microarray. To confirm rk expression, I recommend authors to examine rk>GFP expression in the larval fat body using existing lines (Fig. 4A) and perform qPCR to validate rk RNAi efficiency in the larval fat body.

3. Burs in the hemolymph. Burs-associated neuronal regulation could involve local interactions, while brain-fat body communication requires hormone signaling in the hemolymph. To investigate hormonal regulation, examine tagged Burs amounts (Fig. 3C, 7C) in the hemolymph under HSD, Burs RNAi/KO, and tkv RNAi conditions.

Minor comments:

4. Focusing the manuscript. Since the revised manuscript emphasizes adipose effects (title and mammalian results), consider moving neuronal results (Figs 3, 4, some of Fig. 7) to supplemental data. The remaining results could be condensed into a 4-figure story for a clearer narrative.

5. Improved pupation in RNAi screen. Explain why the RNAi screen yielded no hits that improved delayed pupation under HSD. Known negative insulin signaling regulators like ImpL2, Upd3, and Dome should have been identified. Did the screen have limitations?

6. Mechanism of rk and insulin signaling. The link between rk signaling and insulin signaling (via Nlaz) is unclear. Nlaz RNAi only mildly affects pupation and even increases 4EBP. Additionally, if rk is expressed in the fat body, it might activate cAMP/PKA signaling as in neurons, possibly suppressing insulin signaling.

7. Gbb targeting fat body. Previous results have shown that the larval fat body expresses Tkv and Mad to respond to Gbb. It is likely that muscle-derived Gbb directly targets fat body to impair insulin signaling, independent of burs neurons. To exclude this, perform Tkv RNAi in the fat body. Additionally, Fig. 7O is not accurate. It appears to depict Gbb retrograde signaling, which might imply Burs neuron innervation of muscle. However, there's no current evidence to support this concept.

Response to reviewers' comments

Reviewer #1 (Remarks to the Author):

Reviewer: It has been revised mainly according to comments. However, in the reply, "your study now not only connects *Drosophila* findings to mammalian models but also highlights the potential for LGR4-targeted therapies to enhance insulin sensitivity and treat diabetes" seems extended too much. Please add limitation(s) to strengthen this report.

Response: We appreciate the reviewer's continued input of time and thought towards improving our work. To address the point raised here, we have revised the abstract and results section to be more circumspect. We have removed the statement about the potential for LGR4-targeted therapies. Instead, we now simply state that our results suggest LGR4 plays a crucial role in maintaining normal insulin sensitivity and signaling flux through the insulin-signaling pathway in mouse adipocytes. In the discussion section, we have introduced limitations as suggested by the reviewer regarding the question of whether activating LGR4 signaling could reverse insulin resistance, which may have implications for developing therapies to treat type-2 diabetes. We acknowledge several limitations that must be considered, such as the translational gap from *Drosophila* models and mammalian cell culture to human applications. We also note that further studies are necessary to fully elucidate the mechanisms by which LGR4 promotes insulin sensitivity. By adding these limitations, we have strengthened the report, making it more realistic and credible, and preemptively addressing potential criticisms or questions from future readers. We are grateful to the reviewer for this comment and believe we have comprehensively addressed it.

Reviewer #2 (Remarks to the Author):

Reviewer: The reviewed manuscript is greatly improved and it has an overall very high standard. The authors have addressed all of my comments. The manuscript will be a valuable contribution to different fields.

Response: We thank the reviewer for the positive comments recognizing the manuscript's enhanced quality and its potential contribution to various fields. We sincerely appreciate the time and effort the reviewer has dedicated to reviewing our manuscript and providing insightful feedback, which has been invaluable in guiding the improvements we have made. Thank you for your valuable contribution to refining our work.

Minor comments:

Reviewer point 1: Please remove Relaxin from the new title and maybe provide a more specific LGR pathway. The study does not address relaxin signaling in any aspect, so the new title is inaccurate and misleading. Indeed the only time Relaxin is used is at the title. "LGRs" are a large family of GPCRs with a large diversity of ligands and functions, which are classified as multiple types (Type A, B, C1 or C2; Van Hiel et al., 2012). The type studied in the manuscript (Fly Lgr2 (rickets)/human LGR4) are Type B LGRs, while the relaxin receptor family of LGRs are Type C1, as they have a different domain architecture, different ligands, and different roles. Hence neither relaxin receptors, nor relaxin peptides, nor relaxin signaling was addressed in the manuscript. This is further important here to avoid confusion between Human LGR4 (a Type B LGR, orthologous to *Drosophila* Lgr2) and Fly Lgr4 (a Type C1 relaxin receptor, orthologous to human LGR7/8).

Response: We thank the reviewer for highlighting the issue of including "Relaxin" in our title. We have removed this term and specified the LGR pathway more accurately, as suggested. Thank you.

Reviewer point 2: In the abstract (line 32) it is stated "interaction between ..BMP.. and .. Leucine-rich repeat-containing G-protein coupled Receptor (LGR) signaling pathways". As discussed above, this is too vague and uninformative. Please specify the pathway.

Response: We thank the reviewer for guidance; we have specified the pathway as suggested.

Reviewer point 3: Introduction Line 83 - please double-check if “analog” is the appropriate term here.

Response: We have reviewed the terminology and rephrased the text, now referring to it as "related" to address the concern. Thank you for pointing this out.

Reviewer point 4: Figure 1A - maybe the fly cartoon is missing one red eye?

Response: In our version of Figure 1A, the fly cartoon appears with both eyes correctly depicted. However, we appreciate the attention to detail and will ensure to double-check this aspect at later stages, particularly if we receive a proof version for final review. Thank you for bringing this to our attention.

Reviewer point 4: Results, line 189 (Fig2c): “... an effect that was abolished by RNAi or CRISPR-targeting of Burs in Burs+ cells (Fig. 2c).” The new Fig 2c only shows RNAi data, so please adjust the text.

Response: Thank you for pointing this out. We have now corrected the text to accurately reflect that Fig 2c shows only RNAi data.

Reviewer point 5: I recommend placing Supplementary Fig. 3e in the main figures.

Response: We have taken the recommendation into account and moved Supplementary Fig. 3e to the main figures as Fig. 4f. Thank you for the suggestion.

Reviewer point 6: Results Line 360, “These results elucidate the role of the Drosophila LGR4 ortholog, Rickets/dLgr2,..”. Maybe adjust to “These results elucidate the role of the Drosophila ortholog of the mammalian Lgr4, Rickets/dLgr2,..” to avoid confusion.

Response: We have made changes according to the reviewers' suggestion.

Reviewer point 7: Results line 399, maybe a typo in “in controls responded must less strongly...”?

Response: We fixed the typo. Thank you.

Reviewer point 8: Results Line 439-440: “We found that knockdown of gbb in the muscles or of tkv in the Burs+ neurons reduced the expression of Pburs in the CNS (Supplementary Fig. 4g)”. I could not find the data on gbb RNAi in muscle cells regarding Pburs in the CNS in Supplementary Fig. 4g.

Response: The reviewer is right, and we have corrected this oversight by now stating 'that knockdown of the Gbb receptor tkv in the Burs+ neurons reduced the expression of Pburs in the CNS.' Thank you.

Reviewer #3 (Remarks to the Author):

Reviewer: The authors have addressed some of my comments, such as the functional Burs from neurons and inconsistent pupation times, in revised manuscript. However, the authors may have misinterpreted my concerns regarding "rescued phenotypes of delayed pupation" with "gain-of-functional assays." Specifically, it remains unclear whether the burs/rk gain-of-function is SUFFICIENT to increase insulin signaling and promote pupation under HSD conditions.

Response: We appreciate the reviewer’s input of time and thought into improving our work. Since the same comment appears in major point #1, we have provided our detailed response below, aiming to avoid extensive repetition and save the reviewer's valuable time. We therefore kindly direct the reviewer to see our response to major point #1.

Reviewer: Meanwhile, the newly included data on mammalian adipocytes, while interesting, shifts the focus towards adipose – not neuronal – metabolic regulation by *rk/LGR4*. This renders the results on *Ilp2* regulation somewhat irrelevant to the core findings. The authors should consider reorganizing the manuscript to deprioritize the neuronal regulation and address CRITICAL issues of adipose regulation.

Response: As this comment pertains to the reviewer's minor point #4, we kindly direct the reviewer to our detailed response regarding that point below.

Reviewer: While the authors have incorporated a lot of efforts and additional experiments, a major revision is still needed to focus the manuscript and address the key remaining concerns.

Major comments:

Reviewer point 1: *Burs/rk* gain-of-function under HSD: The manuscript demonstrates that *burs/rk* loss-of-function exacerbates HSD phenotypes, but it doesn't establish whether *burs/rk* activation is SUFFICIENT to promote pupation and insulin signaling under HSD conditions. To address this, consider experiments like *TrpA1* activation or *Burs* overexpression in *Burs*⁺ neurons. Examining insulin signaling markers (tGPH, 4EBP), pupation rates, and hemolymph glucose levels under HSD could validate a real “protective” role for *burs/rk* as the authors claimed in the manuscript. These fly results would also strengthen the connection to mammalian *LGR4* as a potential diabetes target, since they included mammalian results to highlight the disease relevance.

Response: We appreciate the reviewer's feedback and apologize for any misunderstanding and lack of clarity in our initial response to the reviewer's original point. We recognize the point regarding the *Burs/rk* gain-of-function (GOF) assays, especially in relation to insulin signaling and pupation under high-sugar-diet (HSD) conditions. In our revised manuscript, we had sought to address these concerns by demonstrating that activation of *Bursicon* signaling via *TrpA1* (Fig. 7i) leads to a reduction in *Ilp2* staining in the insulin-producing cells (IPCs), without a corresponding decrease in *Ilp2* gene expression (Fig. 7j). These data are intended to show that enhanced *Bursicon* signaling is sufficient to modulate insulin signaling, as evidenced by the altered *Ilp2* staining intensity. These findings suggest that the activation of *Bursicon* signaling is not only sufficient but also functionally relevant in enhancing insulin signaling under HSD conditions. This addresses, at least in part, the reviewer's point regarding demonstrating sufficiency.

We opted for this approach to specifically investigate the direct effects of enhanced *Bursicon* signaling on the IPCs, which expresses the *Bursicon* receptor *Rickets*. This was due to the challenges associated with assessing systemic GOF effects, such as hemolymph glucose levels, systemic insulin-signaling markers, and pupariation timing as the reviewer suggest. These challenges arise from a lack of specificity and the difficulty in precisely manipulating *Bursicon* release through *TrpA1*-induced activation of *Burs*⁺ neurons, leading to ambiguous interpretations. A significant complications is that these *Burs*⁺ neurons express a number of additional peptide hormones with systemic effects, including (at least) crustacean cardioactive peptide (CCAP)¹, which is co-released with *Bursicon* and known to have systemic metabolic effects², and myoinhibiting peptide (MIP or Allatostatin B, AstB)³. The co-expression of *Bursicon* and other peptides complicates our ability to distinguish their respective roles in the systemic response of *Drosophila* larvae to a HSD. Consequently, the limitations of the *Bursicon* GOF tool restrict its use to examining more direct interactions such as with the IPCs, where we can rely on the specificity provided by the *Rickets* receptor (as demonstrated by the *Bursicon* effect on IPCs, Fig. 7i). At the systemic level, the confounding effects of CCAP and MIP, which would also be released by the *TrpA1*-induced activation of *Burs*⁺ neurons that the reviewer suggests, make it very difficult to isolate the effects of *Bursicon* alone on pupariation rate, circulating glucose, and tissue insulin signaling. Furthermore, enhancing *Bursicon* signaling in a HSD condition, where the organism is already experiencing insulin resistance and elevated insulin levels (hyperinsulinemia), could potentially exacerbate insulin resistance rather than ameliorate it, if it leads to a further increase in circulating insulin levels. This highlights the complexity of interpreting the systemic effects of *Bursicon* signaling enhancement under these conditions.

However, to further assess Bursicon GOF and address the reviewer's points, we adopted a more specific approach than activating the Bursicon neurons. We used a heterodimeric form of membrane-tethered Bursicon (tetBurs), in which a tandem heterodimer of Bursicon (Burs-alpha) and Partner of Bursicon (Burs-beta) remains anchored to the plasma membrane of the expressing cells and therefore specifically activates Rk in the targeted tissue⁴. This method allows us to explore the effects of Bursicon GOF in specific tissues where tetBurs is expressed. To investigate the effects of enhanced Bursicon in the fat body, we used the *Cg-GAL4* fat body driver to express tetBurs in this tissue. In the revised manuscript, we now show that activating Bursicon signaling results in increased fat storage under normal sugar conditions, evidenced by lipid-droplet accumulation. This effect mirrors the consequences of high-sugar-diet (HSD) feeding, under which conditions Bursicon signaling is required for increasing fat storage. We opted for lipid-droplet quantification over triacylglycerol (TAG) levels to precisely observe fat body-specific fat storage effects. These new findings, presented in Figure 5c of the revised manuscript, support the conclusion that Bursicon GOF in fat tissue produces effects contrary to those observed with loss-of-function.

As a geneticist by training, I have learned from distinguished fly geneticists, including Michael Ashburner and Michael O'Connor, that while GOF experiments can sometimes offer useful insights, sometimes they can also be misleading. LOF data, on the other hand, tend to provide more definitive information about gene function. Their view was that, from a genetic perspective, the comparison between LOF and GOF experiments favors LOF for several reasons, making them particularly valuable for understanding gene function and biological mechanisms. LOF mutations, such as knockouts or knockdowns, result in the reduction or complete absence of a gene product. This leads to more straightforward interpretations of a gene's role, as observed phenotypes can be directly attributed to the lack of the gene's function. In contrast, GOF can introduce a new or physiologically irrelevant function, which can be more difficult to interpret. In our case, TrpA1-mediated GOF suggested by the reviewer will lead to the release of multiple hormones, obscuring rather than clarifying our analysis. Thus, interpretation of GOF results must be done cautiously as they are not always physiologically or biologically relevant and can produce phenotypes that do not accurately reflect natural biological processes, potentially leading to misleading conclusions about gene function. For instance, both the loss and overexpression of the hormone *Ion Transport Peptide (ITP)* lead to a significant decrease in starvation resistance, which is difficult to reconcile⁵.

We respect the reviewer's perspective but hope that he or she understands the limitations in this particular case. Experimental limitations prevent us from unequivocally interpreting the systemic effects of such GOF with TrpA1 activation of the Bursicon neurons. However, we have employed TrpA1 in certain experiments in which we deemed it appropriate. Beyond the activation of the Burs⁺ neurons to demonstrate their regulatory effect on the IPCs (as mentioned above and illustrated in Fig. 7i), we have also used TrpA1 to activate the IPCs to show that the loss of Rk in these cells can be compensated for by TrpA1-induced release of insulin-like peptides (now included in the main figure 4f of the revised manuscript, as suggested by reviewer 2). This indicates that Rk is necessary for normal ILP release. Furthermore, LOF data at all levels support the conclusion that Bursicon is important for sugar-tolerance phenotypes. Additionally, we have revised the manuscript to ensure that we do not claim Bursicon's sufficiency beyond what our results can substantiate.

I also kindly request the reviewer to consider the extensive amount of data already included in the manuscript, alongside the years of dedicated effort by a team of talented individuals. The initial screening of more than 2,000 genes alone engaged three people for over a year, followed by six years of rigorous follow-up work and significant revisions. Normally, I refrain from using such arguments, but in this context, I cannot in good conscience ask my team to undertake further experiments unless absolutely necessary to bolster our conclusions. Since we are not claiming that Bursicon is sufficient for rescue of the HSD-induced phenotypes, additional experiments do not seem essential for supporting our conclusions. However, I am certainly open to further editing the text, should the reviewer deem it necessary.

Reviewer point 2: *rk* expression in larval fat body. Previous studies suggest that *rk* is not likely to be expressed in the adult fat body. The flycellatals also revealed that *rk* is actually expressed in the adult hemocytes, but not fat body (*bmm* expression). I really doubted that whether *rk* is expressed in the larval fat body. While FlyAtlas microarray data showed *rk* expression in the larval fat body, FlyAtlas 2 RNAseq data contradicts this. That could be the false positive results from non-specific signals in microarray. To confirm *rk* expression, I recommend authors to examine *rk>GFP* expression in the larval fat body using existing lines (Fig. 4A) and perform qPCR to validate *rk* RNAi efficiency in the larval fat body.

Response: With all due respect to the reviewer, and expressing true appreciation for all their suggestions to prove the specificity of the signaling network explored in our study, we disagree with the latter statement regarding *rk* expression in the larval fat body. Upon revisiting the relevant FlyAtlas 2 data, it is clear that *rk* expression is indeed significant in only two tissues in the larval stage, the fat body and the tracheal system (please see figure below from FlyAtlas 2), with an FPKM (Fragments Per Kilobase of transcript per Million mapped reads) value of 1.6 in the larval fat body. According to the general guidelines on FPKM values, in RNA-seq analysis practice, an FPKM value above 1 is considered as evidence of gene expression. (As a point of comparison, the insulin receptor is expressed at a level of only 2.6 FPKM in the larval fat body according to FlyAtlas 2). This is especially relevant when compared to tissues with much lower average expression levels, such as in this case, where even the larval CNS exhibits an FPKM of only 0.3 for *rk*. This comparison underscores the relative expression of *rk* in the larval fat body based on the RNA-seq data available in FlyAtlas 2.

Regarding the concerns raised about potential discrepancies between the original FlyAtlas microarray data and the RNA-seq data from FlyAtlas 2, and the recommendation to further validate *rk* expression using *rk>GFP* expression lines and qPCR to test the efficacy of *rk* RNAi in the larval fat body, we do recognize the value of such validation steps. However, given the explicit support from FlyAtlas 2 for *rk* expression in the larval fat body, we believe the RNA-seq data provides a solid foundation for asserting *rk's* expression in this tissue. We have validated the salient phenotypes with independent tissue-specific RNAi and CRISPR lines targeting *rk* in the fat body and have expanded our research to include the mammalian homolog *LGR4* in adipocytes. Additionally, the newly added data obtained using membrane-tethered Bursicon (tetBurs; we kindly refer the reviewer to our response to point 1 above) further supports fat-body expression of *rk*, because the effects observed from expression of Bursicon tethered to the membrane in the fat body can only occur if Rk is present in this tissue.

We wish to highlight that the additional experiments suggested were not part of the initial review request and do not pertain to questions raised about our new data in the revision. Given the conclusive support provided by the RNA-seq data from FlyAtlas 2 and the new tetBurs data, we consider the matter of fat-body Rickets gene expression has been effectively addressed.

Reviewer point 3: Burs in the hemolymph. Burs-associated neuronal regulation could involve local interactions, while brain-fat body communication requires hormone signaling in the hemolymph. To investigate hormonal regulation, examine tagged Burs amounts (Fig. 3C, 7C) in the hemolymph under HSD, Burs RNAi/KO, and *tkv* RNAi conditions.

Response: We appreciate the reviewer's suggestion to investigate the levels of circulating Bursicon in the hemolymph. However, we would like to address the challenges associated with measuring Bursicon,

or indeed most hormones, in the hemolymph. The technical difficulties arise from the small volumes of hemolymph that can be extracted from *Drosophila* and the low concentration of circulating hormones, combined with the relatively low sensitivity of Western blot or ELISA techniques for detecting the minute quantities of hormones typically present. Despite our efforts, these challenges have precluded successful measurement of Bursicon in the hemolymph, although it is acknowledged that Bursicon circulates within the hemolymph.

Furthermore, we would like to note that this experiment was not requested in the initial review and again, does not relate to questions about our new data added to the revised manuscript. We focused our revisions on areas explicitly highlighted for improvement in the first round of feedback. Bearing this in mind, along with the specific technical difficulties we have outlined, we respectfully chose not to proceed with the newly suggested experiments.

Minor comments:

Reviewer point 4: Focusing the manuscript. Since the revised manuscript emphasizes adipose effects (title and mammalian results), consider moving neuronal results (Figs 3, 4, some of Fig. 7) to supplemental data. The remaining results could be condensed into a 4-figure story for a clearer narrative.

Response: We appreciate the feedback and the suggestion to clarify the focus and structure of our manuscript. The reviewer's point regarding the newly included data on mammalian adipocytes and the perceived shift in focus towards adipose metabolic regulation by Rickets/LGR4 is well taken. However, we believe that each data set enriches the other and that together they illustrate a comprehensive pathway from large-scale *in vivo* screening in *Drosophila* to elucidating a complex inter-organ communication mechanism that includes muscle, neuronal, and fat tissues. Our results show that Bursicon is neuronally produced in response to sugar levels perceived by the musculature and acts to modulate insulin signaling via separate but converging mechanisms – both by regulating insulin-like-peptide production and by governing the sensitivity of downstream tissues to these peptides. Thus our work extends beyond a neuronal focus and encompasses a wider physiological narrative. The mammalian adipocyte data underscore the broader relevance of our findings in *Drosophila* and suggest a potential translational impact. We believe that the data on the mammalian LGR ortholog of *Drosophila* Rickets merely reinforces the concept that the function of the receptor in adipose tissue (rk/LGR4) is to promote insulin sensitivity, rather than shifting the focus.

The work on mammalian adipocytes was performed to address the comments by another reviewer, not to shift the focus. We hope the reviewer understands that it would be very complicated to satisfy the sometimes diverging views and suggestions of different reviewers. Our revised manuscript received positive feedback from the other two reviewers who did not suggest any changes in its structure. This structure is designed to guide the reader seamlessly through the discovery process, from identifying the pathway in *Drosophila* larvae to its implications in mammalian biology. We believe this narrative structure is one of the strengths of our manuscript and effectively highlights the potential of large-scale screening in model organisms to uncover mechanisms with significant translational relevance. While we acknowledge the reviewer's concerns regarding the focus on Ilp2 regulation, we argue that this detail enhances the comprehensive understanding of the system's regulation rather than detracting from the core findings. It demonstrates that Rk/LGR4 signaling promotes both ILP2 (insulin) regulation and tissue responses to insulin, thus playing a broader role in insulin signaling.

Given the positive feedback on the manuscript from the other reviewers and our conviction in the narrative's coherence and its capacity to convey our findings' significance, we prefer to maintain the current organization. We believe that any major structural changes might dilute the impact of our findings and disrupt the narrative flow, potentially confusing readers.

In conclusion, we respectfully request that the inclusion and presentation of the mammalian adipocyte data be considered an integral component of illustrating the broad applicability and significance of our

research findings with highlighting of future perspectives in the research topic, rather than a shift in focus. We are, however, open to further clarifying these connections within the text to ensure the clarity of the manuscript.

Reviewer point 5: Improved pupation in RNAi screen. Explain why the RNAi screen yielded no hits that improved delayed pupation under HSD. Known negative insulin signaling regulators like ImpL2, Upd3, and Dome should have been identified. Did the screen have limitations?

Response: The absence of known negative regulators of insulin signaling such as ImpL2, Upd3, and Dome among our screen's hits is indeed interesting – if reduced insulin signaling is the underlying cause of developmental delay on 5x sugar, then increasing insulin signaling might be expected to rescue this defect. Reasons for this absence might be attributed to limitations in the screening approach, including sensitivity or the severity of the HSD challenges, which could make mitigating the induced delay difficult. Additionally, reversing this phenotype might necessitate targeting multiple genes or pathways simultaneously, where a single-gene approach proves inefficient. Although knockdown of *Dome* has previously been shown to revert aspects of insulin resistance in response to a high-sugar diet – likely what the reviewer is referring to – it has not been demonstrated that the loss of *Dome* rescues the sugar-induced delay in pupariation⁶, which was our screening output. Furthermore, we wish to note that again this was a question not raised by the reviewer in the initial review phase. We have addressed this in the results section of our revised manuscript by incorporating a brief discussion on these screening limitations to clarify these points.

Reviewer point 6: Mechanism of rk and insulin signaling. The link between rk signaling and insulin signaling (via Nlaz) is unclear. Nlaz RNAi only mildly affects pupation and even increases 4EBP. Additionally, if rk is expressed in the fat body, it might activate cAMP/PKA signaling as in neurons, possibly suppressing insulin signaling.

Response: The reviewer asks about the relationship between Bursicon-Rk signaling and insulin signaling in the fat body. Regarding the connection between Nlaz and Rk, we note that this question represents a new inquiry not mentioned by the reviewer in the initial round of review. Indeed, another reviewer inquired about Nlaz in their first review, prompting us to include a substantial amount of data that fully addressed their concerns. Given that this is a new question, we have chosen not to elaborate further on this point.

The relationships between Burs/Rickets signaling, cAMP/PKA, and insulin sensitivity are indeed worthy of investigation, but we think that investigation lies beyond the scope of our work here. Our work demonstrates that the loss of Bursicon-Rk signaling in the fat body leads to impaired insulin signaling in that tissue, suggesting that in this instance, Bursicon-Rk signaling facilitates, rather than suppresses, insulin signaling. The reviewer correctly suggests that Rk activation could lead to cAMP/PKA signaling in the fat body, similar to its role in neurons. It is, however, important to acknowledge the complex effects of cAMP/PKA signaling in adipocytes. We note that this specific question was also not raised previously by the reviewer, and as such, we have not elaborated on it further.

Reviewer point 7: Gbb targeting fat body. Previous results have shown that the larval fat body expresses Tkv and Mad to respond to Gbb. It is likely that muscle-derived Gbb directly targets fat body to impair insulin signaling, independent of burs neurons. To exclude this, perform Tkv RNAi in the fat body. Additionally, Fig. 7O is not accurate. It appears to depict Gbb retrograde signaling, which might imply Burs neuron innervation of muscle. However, there's no current evidence to support this concept.

Response: Our research indeed supports the notion that Bursicon-positive neurons respond to Gbb signaling, and we have provided evidence that muscle tissue is a primary source of the relevant ligand affecting these neurons. Regarding the reviewer's comment that there is no current evidence of Bursicon neuron innervation of the muscle or of Gbb retrograde signaling, we respectfully disagree. Prior studies have clearly demonstrated that the neurons that express Burs and PBurs terminate on certain muscles

of the larval body wall and, based on the morphology of their terminals, likely release peptide both towards the muscle and into the hemolymph⁷⁻⁹. Furthermore, as we mention in our manuscript, retrograde Gbb signaling from these target muscles is necessary for the function of the Burs+PBurs neurons¹⁰. We show (Fig. 7g-n) that *gbb* expression in the larval muscle tissue responds to dietary sugar levels and insulin signaling and that muscle-derived Gbb or Gbb signaling in the Burs+ neurons affects insulin expression and pupation timing on high-sugar medium. These findings reinforce the notion that the larval body-wall muscles are a significant source of Gbb affecting these neurons, thereby aligning with and supporting earlier research. Regarding the depiction of retrograde Gbb signaling in Fig. 7o and the innervation of muscles by Bursicon neurons, we thus direct the reviewer to existing literature that confirms retrograde Gbb signaling from muscles plays a role in regulating Bursicon neurons. Making clear the retrograde nature of the muscle signal was indeed requested by another reviewer.

While we acknowledge that Gbb can be produced by tissues other than the musculature, including the fat body, and hormone from these other sources may exert metabolic effects, our study is primarily aimed at exploring the impact of muscle-derived Gbb on Bursicon neurons. This emphasis does not discount the potential metabolic roles of fat body-derived Gbb, which presents an interesting avenue for future research. However, our current findings highlight a critical regulatory pathway that originates from muscle tissue.

The suggestion to conduct RNAi against *tkv* in the fat body to investigate Gbb targeting of the fat body and its independent effects on insulin signaling is indeed an interesting one, but it introduces a new experimental direction that was not previously raised by the reviewer. Although we recognize the significance of investigating all possible Gbb pathways, such an experiment falls outside the current study's focus on the role of muscle-derived Gbb and its influence on Bursicon neurons. Given that this is a new inquiry and our belief that our findings already provide substantial insight into this particular signaling pathway, we have chosen not to pursue this additional line of inquiry.

References

- 1 Dewey, E. M. *et al.* Identification of the gene encoding bursicon, an insect neuropeptide responsible for cuticle sclerotization and wing spreading. *Curr Biol* **14**, 1208-1213 (2004). <https://doi.org:10.1016/j.cub.2004.06.051>
- 2 Williams, M. J. *et al.* CCAP regulates feeding behavior via the NPF pathway in *Drosophila* adults. *Proc Natl Acad Sci U S A* **117**, 7401-7408 (2020). <https://doi.org:10.1073/pnas.1914037117>
- 3 Kim, Y. J., Zitnan, D., Galizia, C. G., Cho, K. H. & Adams, M. E. A command chemical triggers an innate behavior by sequential activation of multiple peptidergic ensembles. *Curr Biol* **16**, 1395-1407 (2006). [https://doi.org:S0960-9822\(06\)01754-4](https://doi.org:S0960-9822(06)01754-4) [pii] 10.1016/j.cub.2006.06.027
- 4 Harwood, B. N. *et al.* Membrane tethered bursicon constructs as heterodimeric modulators of the *Drosophila* G protein-coupled receptor rickets. *Mol Pharmacol* **83**, 814-821 (2013). <https://doi.org:10.1124/mol.112.081570>
- 5 Galikova, M., Dirksen, H. & Nassel, D. R. The thirsty fly: Ion transport peptide (ITP) is a novel endocrine regulator of water homeostasis in *Drosophila*. *PLoS Genet* **14**, e1007618 (2018). <https://doi.org:10.1371/journal.pgen.1007618>
- 6 Lourido, F., Quenti, D., Salgado-Canales, D. & Tobar, N. Domeless receptor loss in fat body tissue reverts insulin resistance induced by a high-sugar diet in *Drosophila melanogaster*. *Sci Rep* **11**, 3263 (2021). <https://doi.org:10.1038/s41598-021-82944-4>

- 7 Prokop, A. Organization of the efferent system and structure of neuromuscular junctions in *Drosophila*. *Int Rev Neurobiol* **75**, 71-90 (2006). [https://doi.org:10.1016/S0074-7742\(06\)75004-8](https://doi.org:10.1016/S0074-7742(06)75004-8)
- 8 Vomel, M. & Wegener, C. Neurotransmitter-induced changes in the intracellular calcium concentration suggest a differential central modulation of CCAP neuron subsets in *Drosophila*. *Dev Neurobiol* **67**, 792-808 (2007). <https://doi.org:10.1002/dneu.20392>
- 9 Karsai, G. *et al.* Diverse in- and output polarities and high complexity of local synaptic and non-synaptic signaling within a chemically defined class of peptidergic *Drosophila* neurons. *Front Neural Circuits* **7**, 127 (2013). <https://doi.org:10.3389/fncir.2013.00127>
- 10 Veverytsa, L. & Allan, D. W. Retrograde BMP signaling controls *Drosophila* behavior through regulation of a peptide hormone battery. *Development* **138**, 3147-3157 (2011). <https://doi.org:10.1242/dev.064105>

REVIEWERS' COMMENTS

Reviewer #1 (Remarks to the Author):

It has been revised according to comments.

Reviewer #2 (Remarks to the Author):

The authors have adequately addressed all of my comments. The manuscript contains a substantial amount of work and a series of interesting findings that are an important contribution to the field. Thank you for the opportunity to participate in this review. Alisson Gontijo

Reviewer #3 (Remarks to the Author):

While I have carefully considered the situation, I will not recommend rejection of this manuscript. As the authors noted, it represents a significant amount of effort and appears to be a well-done piece of work. I encourage the authors to value their own efforts as well.

The authors have addressed most of the reviewer comments; however, the issue of rk expression in the larval fat body remains. This is a crucial point as it relates directly to mammalian physiology and strengthens the argument for conserved regulation. The authors already possess the necessary tools, including rk-Gal4>UAS-srcGFP (Fig. 4a) and knockdown+qPCR, to readily obtain direct evidence for larval rk expression. These assays are also standard practice for confirming gene expression in the larval fat body, a tissue with minimal heterogeneity and ease of dissection. Therefore, it is unclear why the authors REFUSED to perform these experiments and instead rely on indirect evidence (flyatlas, tetBurs and others) for their argument. Their response ("fat-body Rickets gene expression has been EFFICIENTLY addressed") is not entirely persuasive.

Alternatively, if the authors are unable to conduct the additional experiments, I would like to ask them to revise the manuscript and tone down their statement to acknowledge a relatively low level of rk expression in the larval fat body compared to neuronal expression.

We thank the reviewers for their help and insightful comments. Please find below a response to the final point by reviewer #3:

Final comment by reviewer #3: While I have carefully considered the situation, I will not recommend rejection of this manuscript. As the authors noted, it represents a significant amount of effort and appears to be a well-done piece of work. I encourage the authors to value their own efforts as well.

The authors have addressed most of the reviewer comments; however, the issue of *rk* expression in the larval fat body remains. This is a crucial point as it relates directly to mammalian physiology and strengthens the argument for conserved regulation. The authors already possess the necessary tools, including *rk-Gal4>UAS-srcGFP* (Fig. 4a) and knockdown+qPCR, to readily obtain direct evidence for larval *rk* expression. These assays are also standard practice for confirming gene expression in the larval fat body, a tissue with minimal heterogeneity and ease of dissection. Therefore, it is unclear why the authors REFUSED to perform these experiments and instead rely on indirect evidence (flyatlas, tetBurs and others) for their argument. Their response ("fat-body Rickets gene expression has been EFFICIENTLY addressed") is not entirely persuasive.

Alternatively, if the authors are unable to conduct the additional experiments, I would like to ask them to revise the manuscript and tone down their statement to acknowledge a relatively low level of *rk* expression in the larval fat body compared to neuronal expression.

Author response: We have revised the manuscript according to the alternative suggestion by the reviewer and toned down the statement about expression of *rk* in the larval fat body, which addresses this point. We changed "found **significant** expression in the fat body" to "found expression in the fat body". In addition to this, we have acknowledged the limitation of our study in the discussion by saying, "While our findings suggest *rk* gene expression in the larval fat body, based on indirect evidence from the FlyAtlas transcriptomic database¹ and corroborated by phenotypic changes observed in both knockdown and knockout models, we acknowledge a limitation of our study, which is the lack of direct demonstration of *rk* expression in the larval fat body"

Sincerely,

Kim Rewitz

Reference

- 1 Leader, D. P., Krause, S. A., Pandit, A., Davies, S. A. & Dow, J. A. T. FlyAtlas 2: a new version of the *Drosophila melanogaster* expression atlas with RNA-Seq, miRNA-Seq and sex-specific data. *Nucleic Acids Res* **46**, D809-D815 (2018). <https://doi.org/10.1093/nar/gkx976>